# Loss landscape Characterization of Neural Networks without Over-Parametrization

Rustem Islamov[1]   Niccoló Ajroldi[2]   Antonio Orvieto[2,3,4]   Aurelien Lucchi[1]

[1]University of Basel   [2]Max Planck Institute for Intelligent Systems   [3] ELLIS Institute Tübingen
[4]Tübingen AI Center

## Abstract

Optimization methods play a crucial role in modern machine learning, powering the remarkable empirical achievements of deep learning models. These successes are even more remarkable given the complex non-convex nature of the loss landscape of these models. Yet, ensuring the convergence of optimization methods requires specific structural conditions on the objective function that are rarely satisfied in practice. One prominent example is the widely recognized Polyak-Łojasiewicz (PL) inequality, which has gained considerable attention in recent years. However, validating such assumptions for deep neural networks entails substantial and often impractical levels of over-parametrization. In order to address this limitation, we propose a novel class of functions that can characterize the loss landscape of modern deep models without requiring extensive over-parametrization and can also include saddle points. Crucially, we prove that gradient-based optimizers possess theoretical guarantees of convergence under this assumption. Finally, we validate the soundness of our new function class through both theoretical analysis and empirical experimentation across a diverse range of deep learning models.

## 1 Introduction

The strides in empirical progress achieved by deep neural networks over the past decade have been truly remarkable. Central to the triumph of these techniques lies the effectiveness of optimization methods, which is particularly noteworthy given the non-convex nature of the objective functions under consideration. Worst-case theoretical results point to a pessimistic view since even a degree four polynomial can be NP-hard to optimize [30] and the loss landscape of some neural networks are known to include saddle points or bad local minima [6, 70, 88].

Yet, empirical evidence has shown that gradient-based optimizers – including SGD, AdaGrad [21] and Adam [39] among many others – can effectively optimize the loss of modern deep-learning-based models. While some have pointed to the ability of gradient-based optimizers to deal with potentially complex landscapes, e.g. escaping saddle points [36, 17], another potential explanation is that the loss landscape itself is less complex than previously assumed [27, 51].

Some key factors in this success include the choice of architecture [5, 48, 40, 18], as well as the over-parametrization [73, 12, 50, 51]. In the well-known infinite-width limit [86, 35, 32], neural networks are known to exhibit simple landscapes [49]. However, practical networks operate in a finite range, which still leaves a lot of uncertainty regarding the nature of the loss landscape. This is especially important given that the convergence guarantees of gradient-based optimizers are derived by assuming some specific structure on the objective function [37, 79, 71]. Consequently, an essential theoretical endeavor involves examining the class of functions that neural networks can represent.

In this work, we present a new class of functions that satisfy a newly proposed $\alpha$-$\beta$-condition (see Eq. (2)). We theoretically and empirically demonstrate that these functions effectively characterize

38th Conference on Neural Information Processing Systems (NeurIPS 2024).

Table 1: Summary of existing assumptions on the problem (1) and their limitations. Here $\mathcal{S}$ denotes the set of minimizers of $f$ and $f_i^* := \arg\min_x f_i(x)$. Unlike earlier conditions, the $\alpha$-$\beta$-condition is specifically designed to capture local minima and saddle points. NN = Neural Network.

| Condition | Definition | Comments |
|---|---|---|
| QCvx [28] | $\langle \nabla f(x), x - x^* \rangle \geq \theta(f(x) - f(x^*))$ for some fixed $x^* \in \mathcal{S}$ | - excludes saddle points and local minima |
| Aiming [51] | $\langle \nabla f(x), x - \mathrm{Proj}(x, \mathcal{S}) \rangle \geq \theta f(x)$ | - excludes saddle points and local minima
- theoretically holds for NN in the presence of impractical over-parameterization [51]
- does not always hold in practice [Fig. 1 a-b] |
| PL [a] [66] | $\|\nabla f(x)\|^2 \geq 2\mu(f(x) - f^*)$ | - excludes saddle points and local minima
- theoretically holds for NN in the presence of impractical over-parameterization [50]
- does not always hold in practice [Fig. 1 c-d] |
| $\alpha$-$\beta$-condition **[This work]** | $\langle \nabla f_i(x), x - \mathrm{Proj}(x, \mathcal{S}) \rangle \geq \alpha(f_i(x) - f_i(\mathrm{Proj}(x, \mathcal{S})))$ $-\beta(f_i(x) - f_i^*)$ | - might have saddles [Ex. 2] and local minima [Ex. 3]
- in practice does not require over-parameterization [Ex. 5] |

the loss landscape of neural networks. Furthermore, we derive theoretical convergence guarantees for commonly used gradient-based optimizers under the $\alpha$-$\beta$-condition.

In summary, we make the following contributions:

1. We introduce the $\alpha$-$\beta$-condition and theoretically demonstrate its applicability to a wide range of complex functions, notably those that include local saddle points and local minima.

2. We empirically validate that the $\alpha$-$\beta$-condition is a meaningful assumption that captures a wide range of practical functions, including matrix factorization and neural networks (ResNet, LSTM, GNN, Transformer, and other architectures).

3. We analyze the theoretical convergence of several optimizers under $\alpha$-$\beta$-condition, including vanilla SGD (Stochastic Gradient Descent), $\mathsf{SPS}_{\max}$ (Stochastic Polyak Stepsize) [53], and NGN [63] (Non-negative Gauss-Newton).

4. We provide empirical and theoretical counter-examples where the weakest assumptions, such as the PL and Aiming conditions, do not hold, but the $\alpha$-$\beta$-condition does.

## 2 Related work

### 2.1 Function classes in optimization

Studying the convergence properties of gradient-based optimizers has a long history in the field of optimization and machine learning. Notably, one of the fundamental observations is the linear and sub-linear convergence exhibited by GD for *strongly convex* (SCvx) and general *convex* (Cvx) functions [61]. However, most modern Machine Learning models have non-convex loss landscapes, for which the existing convex theory is not applicable. Without assumptions on the loss functions (other than smoothness), one can only obtain weak convergence guarantees to a first-order critical point. This situation has led to the derivation of assumptions that are weaker than convexity but that are sufficient to guarantee convergence of GD-based optimizers. The list includes *error bounds* (EB) [54], *essential strong convexity* (ESC) [52], weak strong convexity (WSC) [60], the restricted secant inequality (RSI) [87], and the quadratic growth (QG) condition [1]. In the neighborhood of the minimizer set $\mathcal{S}$, EB, PL, and QG are equivalent if the objective is twice differentiable [68]. All of them, except QG, are sufficient to guarantee a global linear convergence of GD. However, among these less stringent conditions, the Polyak-Łojasiewicz (PL) condition stands out as particularly renowned. Initially demonstrated by Polyak [66] to ensure linear convergence, it has recently experienced a resurgence of interest, in part because it accurately characterizes the loss landscape of heavily over-parametrized neural networks [50]. It was also shown to be one of the weakest assumptions among the other known conditions outlined so far [37]. A generalized form of the PL condition for non-smooth optimization is the Kurdyka-Łojasiewicz (KL) condition [43, 10] which is satisfied for a much larger class of functions [14, 77] than PL. The KL inequality has been employed to analyze the convergence of the classic proximal-point algorithm [3, 11, 47] and other optimization methods [4, 45].

More recently, some new convex-like conditions have appeared in the literature such as *star-convex* (StarCvx) [62], *quasar-convex* (QCvx) [28], and *Aiming* [51]. These conditions are relaxations of

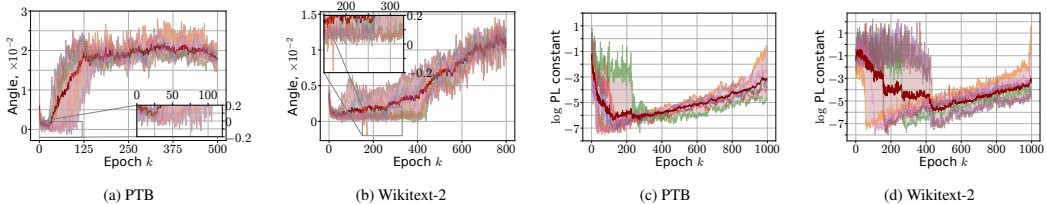

| (a) PTB | (b) Wikitext-2 | (c) PTB | (d) Wikitext-2 |

Figure 1: Training of 3 layer LSTM model that shows Aiming condition does not always hold. The term "Angle" in the figures refers to the angle $\angle(\nabla f(x^k), x^k - x^K)$, and it should be positive if Aiming holds, while in a-b we observe that it is negative during the first part of the training. Figures c-d demonstrate that possible constant $\mu$ in PL condition should be small which makes theoretical convergence slow.

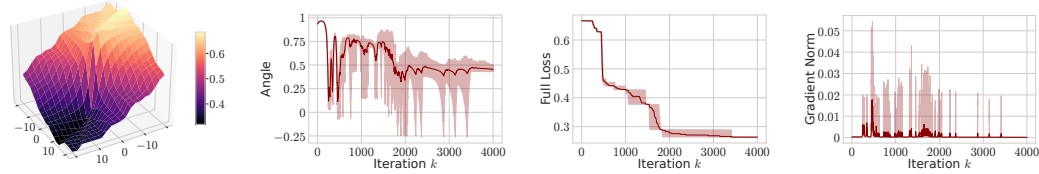

Figure 2: Training for half-space learning problem with SGD. The term "Angle" in the figures refers to the angle $\angle(\nabla f(x^k), x^k - x^K)$.

convexity and include non-convex functions. Within the domain of reinforcement learning, several works [84, 23] have also considered relaxations of the gradient domination condition, although these analyses are conducted specifically within the context of policy gradient methods, and therefore less relatable to StarCvx, QCvx or the Aiming condition.

We present a summary of some of these conditions in Table 1. There is no general implication between already existing assumptions such as QCvx, Aiming, PL, and the $\alpha$-$\beta$-condition. However, as we will later see, the $\alpha$-$\beta$-condition can more generally characterize the landscape of neural networks without requiring unpractical amounts of over-parametrization. Notably, the $\alpha$-$\beta$-condition is a condition that applies globally to the loss. However, we will demonstrate that convergence guarantees can still be established for commonly-used gradient-based optimizers, although these guarantees are weaker than those derived under the PL condition, which relies on much stronger assumptions.

## 2.2 Limitations of existing conditions

Next, we discuss the limitations of previous conditions to characterize the loss landscape of complex objective functions such as the ones encountered when training deep neural networks.

**Necessity of Over-parameterization.** When considering deep models, the theoretical justification of conditions such as Aiming [51] and PL [50] require a significant amount of over-parameterization. This implies that the neural network must be considerably large, often with the minimum layer's width scaling with the size of the dataset $n$. However, various studies suggest that this setup may not always accurately model real-world training dynamics [13, 2]. To the best of our knowledge, the weakest requirements on the width of a network are sub-quadratic $\widetilde{\Omega}(n^{3/2})$ [20, 74], where $n$ is the size of a dataset. This implies that, even for a small dataset such as MNIST, a network should have billions of parameters which is not a realistic setting in practice. In contrast, the $\alpha$-$\beta$-condition condition does not require such an over-parametrization condition to hold (e.g., see Example 5). In Section 5 we provide empirical results showing how our condition is affected by over-parameterization.

**Necessity of Invexity.** One limitation of prior assumptions is their inability to account for functions containing local minima or saddle points. Indeed, many of the weakest conditions, such as QCvx, PL, KL, and Aiming, require that any point where the gradient is zero must be deemed a global minimizer. However, such conditions are not consistent with the loss landscapes observed in practical neural networks. For example, finite-size MLPs can have spurious local points or saddle points [88, 80]. Another known example is the half-space learning problem which is known to have saddle points [17]. We refer the reader to Figure 2-a that illustrates this claim (it showcases the surface of the problem

fixing all parameters except first 2), and also demonstrates that the Aiming and PL conditions fail to hold in such a setting. We present the results in Figure 2[1] where we observe that $(i)$ the angle between the full gradient and direction to the minimizer $\angle(\nabla f(x^k), x^k - x^*)$ can be negative implying that the Aiming condition does not hold in this case (since the angle should remain positive); $(ii)$ the gradient norm can become zero while we did not reach minimum (loss is still large) implying that the PL condition does not hold as well (since the inequality $\|\nabla f(x^k)\|^2 \geq 2\mu(f(x^k) - f^*)$ is not true for any positive $\mu$). These observations suggest that the Aiming and PL conditions do not characterize well a landscape in the absence of invexity property. In contrast, we demonstrate in Example 2 and Figure 9 that our proposed assumption is preferable in this scenario.

**Lack of Theory.** As previously mentioned, most theoretical works apply to some infinite limit or neural networks of impractical sizes. In contrast, several works [89, 27, 75] have studied the empirical properties of the loss landscape of neural networks during training. They have shown that gradient-based optimization methods do not encounter significant obstacles that impede their progress. However, these studies fall short of providing theoretical explanations for this observed phenomenon.

**Lack of Empirical Evidence.** Several theoretical works [49, 51] prove results on the loss landscape of neural networks without supporting their claims using experimental validation on deep learning benchmarks. We demonstrate some practical counter-examples to these conditions proved in prior work. We train LSTM-based model[2] with standard initialization on Wikitext-2 [57] and Penn Treebank (PTB) [58] datasets. In Figure 1 (a-b), we show that the angle between the full gradient and direction to the minimizer $\angle(\nabla f(x^k), x^k - x^*)$ can be negative in the first part of the training. This result implies that the Aiming condition does not hold in this setting (either we do not have enough over-parameterization or the initialization does not lie in the locality region where Aiming holds). Moreover, for the same setting in Figure 1 (c-d) we plot $2\log(\|\nabla f(x^k)\|) - 2/\delta \log(f(x^k) - f(x^K))$[3] to measure the empirical value of PL constant $\log(2\mu)$ (see derivations in Appendix A). We observe that the value of $\mu$ that might satisfy PL condition should be of order $10^{-8} - 10^{-7}$ and leads to slow theoretical convergence [37]. These observations contradict with practical results. We defer to Appendix A for a more detailed discussion. In contrast, we demonstrate in Figure 7-(g-h) that the proposed $\alpha$-$\beta$-condition can be verified in this setting.

## 3 The proposed $\alpha$-$\beta$-condition

**Setting.** We consider the following Empirical Risk Minimization (ERM) problem that typically appears when training machine learning models:

$$\min_{x \in \mathbb{R}^d} \left[ f(x) := \tfrac{1}{n} \sum_{i=1}^n f_i(x) \right]. \tag{1}$$

Here $x \in \mathbb{R}^d$ denotes the parameters of a model we aim to train, $d$ is the number of parameters, and $n$ is the number of samples in the training dataset. Each $f_i(x)$ is the loss associated with the $i$-th data point. We denote the minimum of the problem (1) by $f^*$ and the minimum of each individual function by $f_i^* := \min_x f_i(x)$, which we assume to be finite. Besides, the set $\mathcal{S}$ denotes the set of all minimizers of $f$.

**A new class of functions.** Next, we present a new condition that characterizes the interplay between individual losses $f_i$ and the set of minimizers of the global loss function $f$.

**Definition 1** ($\alpha$-$\beta$-condition ). Let $\mathcal{X} \subseteq \mathbb{R}^d$ be a set and consider a function $f : \mathcal{X} \to \mathbb{R}$ as defined in (1). Then $f$ satisfies the $\alpha$-$\beta$-condition with positive parameters $\alpha$ and $\beta$ such that $\alpha > \beta$ if for any

---

[1]Our implementation is based on the open source repository https://github.com/archana95in/Escaping-saddles-using-Stochastic-Gradient-Descent from [17] with small changes to track necessary quantities.

[2]Our implementation is based on the open source repository https://github.com/fhueb/parameter-agnostic-lzlo from [33] with small changes to track necessary quantities. The detailed experiment description is given in Appendix D.2.

[3]We use $x^K$ (for a large value of $K$) to approximate the minimizer $x^*$.

$x \in \mathcal{X}$ there exists $x_p \in \mathrm{Proj}(x, \mathcal{S})$ such that for all $i$,

$$\langle \nabla f_i(x), x - x_p \rangle \geq \alpha(f_i(x) - f_i(x_p)) - \beta(f_i(x) - f_i^*). \tag{2}$$

The $\alpha$-$\beta$-condition recovers several existing assumptions as special cases. For example, the proposed assumption reduces to QCvx around $x^*$ if $\alpha > 0$, $\beta = 0$, and $\mathcal{S}$ is a singleton $\{x^*\}$. Importantly, the $\alpha$-$\beta$-condition is also applicable when the set $\mathcal{S}$ contains multiple elements.

One can easily check that the pair of parameters $(\alpha, \beta)$ in (2) can not be unique. Indeed, if the assumption is satisfied for some $\alpha$ and $\beta$, then due to inequality $f_i(x_p) \geq f_i^*$ it will be also satisfied for $(\alpha + \delta, \beta + \delta)$ for any $\delta \geq 0$.

### 3.1  Theoretical verification of the $\alpha$-$\beta$-condition

To demonstrate the significance of Definition 1 as a meaningful condition for describing structural non-convex functions, we provide several examples below that satisfy (2). We do not aim to provide the tightest possible choice of $\alpha$ and $\beta$ such that Definition 1 holds. Instead, this section aims to offer a variety of examples that demonstrate specific desired characteristics when $\alpha$-$\beta$-condition holds, encompassing a broad range of functions.

The initial example illustrates that $\mathcal{S}$ could potentially be infinite for the class of functions satisfying Definition 1.

**Example 1.** Let $f, f_1, f_2 \colon \mathbb{R}^2 \to \mathbb{R}$ be such that

$$f = \tfrac{1}{2}(f_1 + f_2) \quad \text{with} \quad f_1(x, y) = \tfrac{(x+y)^2}{(x+y)^2+1}, \quad f_2(x, y) = \tfrac{(x+y+1)^2}{(x+y+1)^2+1}, \tag{3}$$

then Definition 1 holds with $\alpha \in [5/2, +\infty)$ and $\beta \in [4\alpha/5, \alpha)$.

Next, we provide an example where $f$ satisfies Definition 1 even in the presence of saddle points.

**Example 2.** Let $f, f_1, f_2 \colon \mathbb{R}^2 \to \mathbb{R}$ be such that

$$f = \tfrac{1}{2}(f_1 + f_2) \quad \text{with} \quad f_1(x, y) = 1 - e^{-x^2 - y^2}, \quad f_2(x, y) = 1 - e^{-(x-2)^2 - (y-2)^2}, \tag{4}$$

then Definition 1 holds for some $\alpha$ and $\beta = \alpha - 8$.

**Remark 1.** Examples 1 and 2 can be generalized for any number of functions $n$ and dimension $d$ as follows

$$f_i(x) = \frac{\left(\sum_{j=1}^d x_j + a_i\right)^2}{\left(\sum_{j=1}^d x_j + a_i\right)^2 + 1}, \quad f_i(x) = \frac{\sum_{j=1}^d (x_j - b_{ij})^2}{1 + \sum_{j=1}^d (x_j - b_{ij})^2}, \tag{5}$$

for some properly chosen $\{a_i\}$ and $\{b_{ij}\}, i \in [n], j \in [d]$.

**Example 3.** Let $f, f_1, f_2 \colon \mathbb{R}^2 \to \mathbb{R}$ be such that

$$f = \tfrac{1}{2}(f_1 + f_2) \quad \text{with} \quad f_1(x, y) = \tfrac{1+x^2+y^2}{4+x^2+y^2}, \quad f_2(x, y) = \tfrac{(x-2.5)^2+(y-2.5)^2}{4+(y-2.5)^2+(y-2.5)^2}, \tag{6}$$

then Definition 1 holds for some $\alpha$ and $\beta = \alpha - 1$.

The three examples above demonstrate that functions satisfying Definition 1 can potentially be non-convex with an unbounded set of minimizers $\mathcal{S}$ (Example 1) and can have saddle points (Example 2) and local minima (Example 3). In contrast, the PL and Aiming conditions are not met in cases where a problem exhibits saddle points. For illustration purposes, we plot the loss landscapes of $f$ in Figure 3.

So far, we have presented simple examples to verify Definition 1. Next, we turn our attention to more practical examples in the field of machine learning. We start with the matrix factorization problem that is known to have saddle points [76] but can be shown to be PL after a sufficiently large number of iterations of alternating gradient descent and under a specific random initialization [81].

**Example 4.** Let $f_i, f_{ij}$ be such that

$$f(W, S) = \tfrac{1}{2nm} \|X - W^\top S\|_F^2 = \tfrac{1}{2nm} \sum_{i,j} (X_{ij} - w_i^\top s_j)^2, \quad f_{ij}(W, S) = \tfrac{1}{2}(X_{ij} - w_i^\top s_j)^2, \tag{7}$$

where $X \in \mathbb{R}^{n \times m}, W = (w_i)_{i=1}^n \in \mathbb{R}^{k \times n}, S = (s_j)_{j=1}^m \in \mathbb{R}^{k \times m}$, and $\mathrm{rank}(X) = r \geq k$. We assume that $X$ is generated using matrices $W^*$ and $S^*$ with non-zero additive noise that minimize empirical loss, namely, $X = (W^*)^\top S^* + (\varepsilon_{ij})_{i \in [n], j \in [m]}$ where $W^*, S^* = \mathrm{argmin}_{W,S} f(W, S)$. Let $\mathcal{X}$ be any bounded set that contains $\mathcal{S}$. Then Definition 1 is satisfied with $\alpha = \beta + 1$ and some $\beta > 0$.

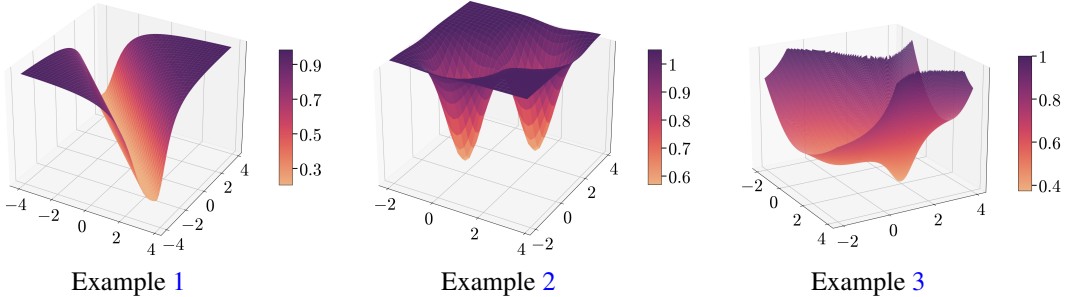

| Example 1 | Example 2 | Example 3 |

Figure 3: Loss landscape of $f$ that satisfy Definition 1. The analytical form of $f_i$ is given in Section 3.1. These examples demonstrate that the problem (1) that satisfies $\alpha$-$\beta$-condition might have an unbounded set of minimizers $\mathcal{S}$ (Example 1), a saddle point (Example 2), and local minima (Example 3) in contrast to the PL and Aiming conditions. The contour plots are presented in **??**.

**Example 5.** Consider training a two-layer neural network with a logistic loss

$$f = \tfrac{1}{n} \sum_{i=1}^n f_i, \quad f_i(W,v) = \phi(y_i \cdot v^\top \sigma(W x_i)) + \lambda_1 \|v\|^2 + \lambda_2 \|W\|_{\mathrm{F}}^2 \tag{8}$$

for a classification problem where $\phi(t) \coloneqq \log(1 + \exp(-t))$, $W \in \mathbb{R}^{k \times d}, v \in \mathbb{R}^k$, $\sigma$ is a ReLU function applied coordinate-wise, $y_i \in \{-1, +1\}$ is a label and $x_i \in \mathbb{R}^d$ is a feature vector. Let $\mathcal{X}$ be any bounded set that contains $\mathcal{S}$. Then the $\alpha$-$\beta$-condition holds in $\mathcal{X}$ for some $\alpha \geq 1$ and $\beta = \alpha - 1$.

**Remark 2.** The previous examples can be extended to any positive and convex function $\phi$ (e.g., square loss) with the additional assumption that each individual loss $f_i$ does not have minimizers in $\mathcal{S}$, i.e. $\nexists (W^*, v^*) \in \mathcal{S}$ such that $f_i(W^*, v^*) = f_i^*$ for some $i \in [n]$.

We highlight that Example 5 is applicable for a bounded set $\mathcal{X}$ of an *arbitrary* size. Moreover, in practice, we typically add $\ell_1$ or $\ell_2$ regularization which can be equivalently written as a constrained optimization problem, and therefore, Example 5 holds in this scenario. In comparison, the results from previous works do not hold for an arbitrary bounded set around $\mathcal{S}$ requiring initialization to be close enough to the solution set [51, 50]. The proofs for all examples are given in Appendix C.1.

## 4 Theoretical convergence of algorithms

We conduct our analysis under the following smoothness assumption that is standard in the optimization literature.

**Assumption 1.** *We assume that each $f_i$ is $L$-smooth, i.e. for all $x, y \in \mathbb{R}^d$ it holds $\|\nabla f_i(x) - \nabla f_i(y)\| \leq L\|x - y\|$.*

The next assumption, which is sometimes called functional dissimilarity [56], is standard in the analysis of SGD with adaptive stepsizes [53, 26, 24].

**Assumption 2.** *We assume that the interpolation error $\sigma_{\mathrm{int}}^2 \coloneqq \mathbb{E}_i[f^* - f_i^*]$ is finite, where the expectation is taken concerning the randomness of indices $i$ for a certain algorithm.*

### 4.1 Convergence under the $\alpha$-$\beta$-condition

Now we demonstrate that the $\alpha$-$\beta$-condition is sufficient for common optimizers to converge up to a neighbourhood of the set of minimizers $\mathcal{S}$. We provide convergence guarantees for SGD-based algorithms with fixed and adaptive stepsize (i.e., the update direction is of the form $-\gamma_k \nabla f_{i_k}(x^k)$). In this section, we only present the main statements about convergence while the algorithms' description and the proofs are deferred to Appendix C.2.

**Convergence of SGD.** We start with the results for vanilla SGD with constant stepsize.

**Theorem 1.** *Assume that Assumptions 1-2 hold. Then the iterates of SGD (Alg. 1) with stepsize $\gamma \leq \frac{\alpha - \beta}{2L}$ satisfy*

$$\min_{0 \leq k < K} \mathbb{E}\left[ f(x^k) - f^* \right] \leq \frac{\mathbb{E}\left[ \mathrm{dist}(x^0, \mathcal{S})^2 \right]}{K} \frac{1}{\gamma(\alpha - \beta)} + \frac{2L\gamma}{\alpha - \beta} \sigma_{\mathrm{int}}^2 + \frac{2\beta}{\alpha - \beta} \sigma_{\mathrm{int}}^2. \tag{9}$$

Table 2: Summary of how the non-vanishing term $\beta\sigma_{\text{int}}^2$ (as appearing e.g. in Eq. (9)) increases ($\nearrow$) or decreases ($\searrow$) as a function of specific quantities of interest.

| | Model's width $\nearrow$ | Model's depth $\nearrow$ | Batch-size $\nearrow$ |
|---|---|---|---|
| Change in $\beta\sigma_{\text{int}}^2$ | $\searrow$ | $\searrow$ | $\searrow$ |

Theorem 1 shows that under the $\alpha$-$\beta$-condition , SGD converges with a rate $\mathcal{O}(K^{-1/2})$ (the same rate obtained by SGD for convex functions [24]) up to a ball of size $\mathcal{O}(\beta\sigma_{\text{int}}^2)$. We argue that the non-vanishing term $\mathcal{O}(\beta\sigma_{\text{int}}^2)$ must appear in the convergence rate for non-convex optimization for several reasons: $(i)$ This term arises directly from the use of the $\alpha$-$\beta$ condition in the analysis, without resorting to additional upper bounds or approximations. It reflects the potential existence of local minima that the $\alpha$-$\beta$ condition is designed to model. In the worst-case scenario, if SGD is initialized near local minima and uses a sufficiently small step size, it may fail to converge to the exact minimizer and can become trapped in suboptimal minima. This sub-optimality is modeled in the upper bound by the stepsize-independent quantity $\mathcal{O}(\beta\sigma_{\text{int}}^2)$ since we provide convergence guarantees for the function value sub-optimality rather than the squared gradient norm, which is more typical in the non-convex setting. $(ii)$ We also observe that the last term in (9) shrinks as a model becomes more over-parameterized (which is consistent with prior works such as [50] that require large amounts of over-parametrization); see Sections 5.1, 5.2, and D.7 for experimental validation of this claim. Further empirical observations are summarized in Table 2 and will be discussed in Section 5. Theoretically, if a model is sufficiently over-parameterized such that the interpolation condition $f_i^* = f^*$ holds, then the non-vanishing term is not present in the bound. $(iii)$ The presence of a non-vanishing error term in the rate with the $\alpha$-$\beta$ condition is consistent with empirical observation as it is frequently observed when training neural networks (see Figure 8.1 in [25]). This is also observed during the training of language models where the loss is significantly larger than 0 (see Figure 19). This phenomenon suggests that reaching a critical point, which is a global minimizer, is not commonly observed practically. $(iv)$ Finally, we note a potential similarity with prior works that propose other conditions to describe the loss landscape of deep neural networks (e.g. gradient confusion [71]), and also obtain a non-vanishing term in the convergence rate (see Theorem 3.2 in [71]).

**Convergence of $\mathsf{SPS}_{\max}$.**   Next, we consider the $\mathsf{SPS}_{\max}$ algorithm. $\mathsf{SPS}_{\max}$ stepsize is given by $\gamma_k := \min\left\{ \frac{f_{i_k}(x^k) - f_{i_k}^*}{c\|\nabla f_{i_k}(x^k)\|^2}, \gamma_{\mathrm{b}} \right\}$ where $c$ and $\gamma_{\mathrm{b}}$ are the stepsize hyperparameters.

**Theorem 2.** *Assume that Assumptions 1-2 hold. Then the iterates of $\mathsf{SPS}_{\max}$ (Alg. 2) with a stepsize hyperparameter $c > \frac{1}{2(\alpha-\beta)}$ satisfy*

$$\min_{0 \le k < K} \mathbb{E}\left[ f(x^k) - f^* \right] \le \frac{c_1}{K} \mathbb{E}\left[ \text{dist}(x^0, \mathcal{S})^2 \right] + 2\alpha c_1 \gamma_{\mathrm{b}} \sigma_{\text{int}}^2, \tag{10}$$

*where $\gamma_{\min} := \min\{1/2cL, \gamma_b\}$ and $c_1 := \frac{c}{\gamma_{\min}(2(\alpha-\beta)c-1)}$.*

In the convex case, i.e. $\alpha$-$\beta$-condition holds with $\alpha = 1, \beta = 0$, we recover the rate of Loizou et al. [53].

**Convergence of NGN.**   Finally, we turn to the analysis of NGN. This algorithm is proposed for minimizing positive functions $f_i$ which is typically the case for many practical choices. Its stepsize $\gamma_k := \frac{\gamma}{1 + \frac{\gamma}{2f_{i_k}(x^k)}\|\nabla f_{i_k}(x^k)\|^2}$ where $\gamma$ is the stepsize hyperparameter. NGN stepsize differs from that of $\mathsf{SPS}_{\max}$ by replacing min operator by softer harmonic averaging of SPS stepsize and a constant $\gamma$. In addition to the already mentioned assumptions, we make a mild assumption that the positivity error $\sigma_{\text{pos}}^2 := \mathbb{E}\left[f_i^*\right]$ is finite.

**Theorem 3.** *Assume that Assumptions 1 with $\alpha \ge \beta + 1$ and 1-2 hold. Assume that each function $f_i$ is positive and $\sigma_{\text{pos}}^2 < \infty$. Then the iterates of NGN (Alg. 3) with a stepsize parameter $\gamma > 0$ satisfy*

$$\min_{0 \le k \le K-1} \mathbb{E}\left[ f(x^k) - f^* \right] \le \frac{\mathbb{E}\left[\text{dist}(x^0, \mathcal{S})^2\right]}{2\gamma K} \frac{(1 + 2\gamma L)^2}{c_2} + \frac{3L\gamma\alpha(1 + \gamma L)\sigma_{\text{int}}^2}{c_2}$$
$$+ \frac{\gamma L}{a} \max\{2\gamma L - 1, 0\}\sigma_{\text{pos}}^2 + \frac{2\beta\sigma_{\text{int}}^2}{c_2}, \tag{11}$$

*where $c_2 := 2\gamma L(\alpha - \beta - 1) + \alpha - \beta$.*

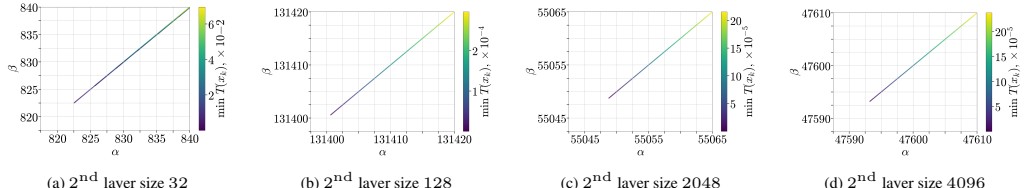

(a) $2^{\text{nd}}$ layer size 32     (b) $2^{\text{nd}}$ layer size 128     (c) $2^{\text{nd}}$ layer size 2048     (d) $2^{\text{nd}}$ layer size 4096

Figure 4: $\alpha$-$\beta$-condition in the training of 3 layer MLP model on Fashion-MNIST dataset varying the size of the second layer. Here $T(x_k) = \langle \nabla f_{i_k}(x^k), x^k - x^K \rangle - \alpha(f_{i_k}(x^k) - f_{i_k}(x^K)) - \beta f_{i_k}(x^k)$ assuming that $f_i^* = 0$. Minimum is taken across all runs and iterations for given pair of $(\alpha, \beta)$.

One of the main properties of NGN is its robustness to the choice of stepsize $\gamma$. Theorem 3 can be seen as an extension of this feature from the set of convex functions originally analyzed in [63] to the class of structured non-convex satisfying $\alpha$-$\beta$-condition.

Comparing the results of Theorems 2 and 3 we highlight several important differences. $(i)$ There is no restriction on the stepsize parameter $\gamma$ for NGN. Conversely, $\mathsf{SPS}_{\max}$ requires $c$ to be lower bounded. $(ii)$ Both algorithms converge to a neighborhood of the solution with a fixed stepsize hyperparameter. However, the neighborhood size of $\mathsf{SPS}_{\max}$ is not controllable by the stepsize hyperparameter and remains constant even in the convex setting when $\beta = 0$. In contrast, NGN converges to a ball whose size can be made smaller by choosing a small stepsize parameter, and the "non-vanishing" term disappears in the convex setting $\beta = 0$.

We note that our goal was not to achieve the tightest convergence guarantees for each algorithm, but rather to underscore the versatility of the $\alpha$-$\beta$-condition in deriving convergence guarantees for SGD-type algorithms, both for constant or adaptive stepsizes. In addition to the results of this section, we demonstrate the convergence guarantees for SGD, $\mathsf{SPS}_{\max}$, and NGN with decreasing with $k$ stepsizes in Appendix C.2. Besides, in Appendix C.2.4 we present a convergence of a slightly modified version of Adagrad-norm method [82] under $\alpha$-$\beta$-condition.

## 5 Experimental validation of the $\alpha$-$\beta$-condition

In this section, we provide extensive numerical results supporting that the $\alpha$-$\beta$-condition does hold in many practical applications for various tasks, model architectures, and datasets. The detailed experimental setting is described in Appendix D.

In all cases, we approximate $\mathrm{Proj}(x^k, \mathcal{S})$ as the last iterate $x^K$ in a run. After finding such an approximation, we start a second training run with the same random seed to measure all necessary quantities. To guarantee that the second training trajectory follows the same path as the first run, we disable non-deterministic CUDA operations while training on a GPU. For each task, we demonstrate possible values of pairs of $(\alpha, \beta)$ that work across all runs (might differ from one experiment to another) with different random seeds and satisfy $\alpha \geq \beta + 0.1$.

### 5.1 MLP architecture

First, we test MLP neural networks with 3 fully connected layers on Fashion-MNIST [83] dataset. We fix the second layer of the network to be a square matrix and vary its dimension layer to investigate the effect of over-parameterization on $\alpha$-$\beta$-condition. We test it for dimensions $\{32, 128, 2049, 4096\}$, and for each case, we run experiments for 4 different random seeds. In Figure 4 we demonstrate possible values of pairs of $(\alpha, \beta)$ that work across all 4 runs. We observe that minimum possible values of $\alpha$ and $\beta$ increase from small size to medium, and then tend to decrease again as the model becomes more over-parameterized. We defer more experimental results for MLP to Appendix D.3 to showcase this phenomenon. This observation leads to the fact that the neighborhood of convergence $\mathcal{O}(\beta \sigma_{\text{int}}^2)$ of SGD eventually becomes smaller with the size of the model as we expect (since it becomes more over-parameterized).

### 5.2 CNN architecture

In our next experiment, we test convolutional neural networks with 2 convolution layers and 1 fully connected layer on CIFAR10 dataset [41]. We vary the number of convolutions in the second convolution layer to investigate the effect of over-parameterization on $\alpha$-$\beta$-condition. We test it

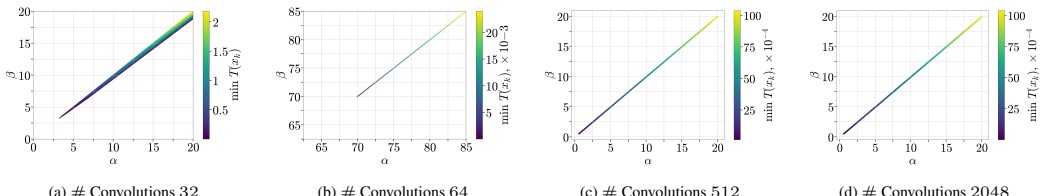

| (a) # Convolutions 32 | (b) # Convolutions 64 | (c) # Convolutions 512 | (d) # Convolutions 2048 |

Figure 5: $\alpha$-$\beta$-condition in the training of CNN model on CIFAR10 dataset varying the number of convolutions in the second layer. Here $T(x_k) = \langle \nabla f_{i_k}(x^k), x^k - x^K \rangle - \alpha(f_{i_k}(x^k) - f_{i_k}(x^K)) - \beta f_{i_k}(x^k)$ assuming that $f_i^* = 0$. Minimum is taken across all runs and iterations for a given pair of $(\alpha, \beta)$.

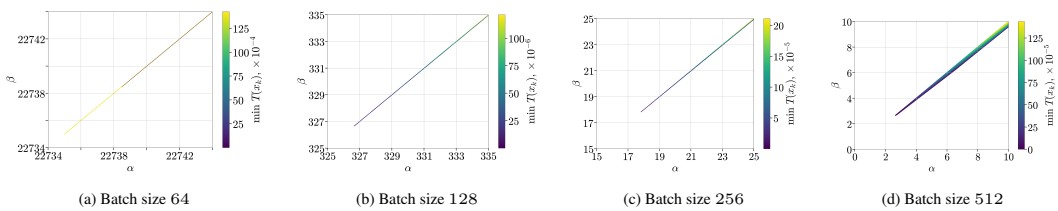

| (a) Batch size 64 | (b) Batch size 128 | (c) Batch size 256 | (d) Batch size 512 |

Figure 6: $\alpha$-$\beta$-condition in the training of Resnet9 model on CIFAR100 dataset varying the batch size. Here $T(x_k) = \langle \nabla f_{i_k}(x^k), x^k - x^K \rangle - \alpha(f_{i_k}(x^k) - f_{i_k}(x^K)) - \beta f_{i_k}(x^k)$ assuming that $f_i^* = 0$. Minimum is taken across all runs and iterations for a given pair of $(\alpha, \beta)$.

for $\{32, 128, 512, 2048\}$ number of convolutions in the second layer, and for each case, we run experiments for 4 different random seeds. In Figure 5, we observe that the smallest possible values of $(\alpha, \beta)$ increase till 64 convolutions, and then decrease back. Second, the difference $\alpha - \beta$ for possible choice of $\alpha$ and $\beta$ decreases from Figure 5-a to Figure 5-b, but then it increases again.

### 5.3 Resnet architecture

Next, we switch to the Resnet architecture [29] with batch sizes in $\{64, 128, 256, 512\}$ trained on CIFAR100 [41]. For each batch size, we run experiments for 4 different random seeds. In Figure 6, we plot the possible choice of pairs $(\alpha, \beta)$ that works across all runs. We observe that $\alpha$-$\beta$-condition holds in all cases. Besides, there is a tendency for the minimum possible choice of $\alpha$ and $\beta$ to decrease with batch size. Moreover, for larger batches, the difference between $\alpha$ and $\beta$ also increases. From Theorem 1, this result suggests that we can use bigger stepsizes with larger batches.

### 5.4 Training of AlgoPerf workloads and transformers for language modeling

We are now interested in assessing the validity of $\alpha$-$\beta$-condition on modern Deep Learning architectures. Thereby, we consider tasks from the AlgoPerf benchmarking suite [16]. We consider four workloads from the competition: (i) DLRMsmall model [59] on Criteo 1TB dataset [44]; (ii) U-Net model [69] on FastMRI dataset [85] (iii) GNN model [7] on OGBG dataset [31]; (iv) Transformer-big [78] on WMT dataset [9]. To train the models, we use NAdamW optimizer [19][4], which achieves state-of-the-art performances on the current version of the benchmark. The hyperparameters of the optimizer are chosen to reach the validation target threshold set by the original competition. Moreover, we consider the pretraining of a decoder-only transformer architecture [78] for causal language modeling. We conduct our evaluation on two publicly available Pythia models [8], of sizes 70M and 160M, trained on 1.25B and 2.5B tokens respectively. For this study, we use the SlimPajama [72] dataset. Following the original Pythia recipes, we fix a sequence length of 2048 and train the language model to predict the next token in a self-supervised fashion. We refer to section Appendices D and D.8 for additional details.

The results are presented in Figure 7. We observe that $\alpha$-$\beta$-condition holds for a wide range of values of $\alpha$ and $\beta$ which demonstrates that our condition can be seen as a good characterization of the training of modern large models as well. One can notice that the possible values of $\alpha$ and $\beta$ for AlgoPerf workloads are higher than those for smaller models discussed in previous sections.

---

[4]Our implementation is based on open source AlgoPerf code `https://github.com/mlcommons/algorithmic-efficiency` with minimal changes to track necessary quantities.

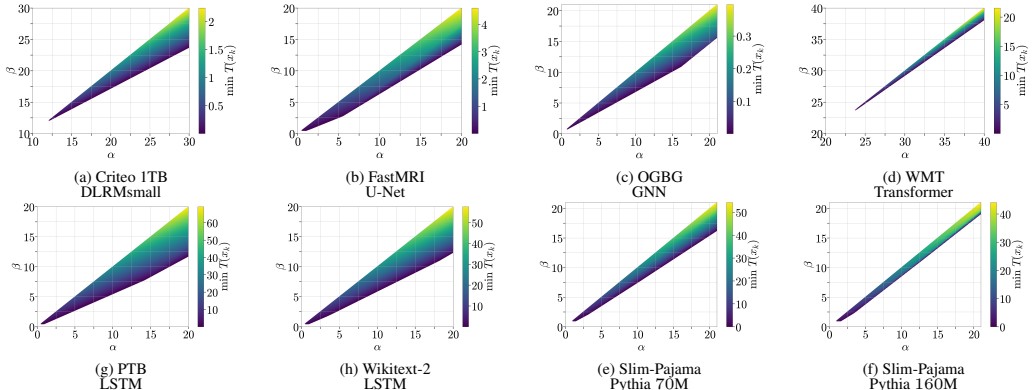

Figure 7: $\alpha$-$\beta$-condition in the training of some large models from AlgoPerf, 3-layer LSTM, and Transformers for language modeling. Here $T(x_k) = \langle \nabla f_{i_k}(x^k), x^k - x^K \rangle - \alpha(f_{i_k}(x^k) - f_{i_k}(x^K)) - \beta f_{i_k}(x^k)$ assuming that $f_i^* = 0$. Minimum is taken across all runs and iterations for a given pair of $(\alpha, \beta)$.

This difference is attributable to smaller models interpolating the training data more effectively, resulting in significantly lower training losses compared to those observed in AlgoPerf experiments (see Appendix D for detailed results). However, we highlight that the convergence guarantees of the optimizers depend on a term $\mathcal{O}(\beta \sigma_{\text{int}}^2)$ which is stable across all experiments we provide.

### 5.5 Additional experiments

We defer the verification of $\alpha$-$\beta$-condition by Adam and SGDM to Appendix D.6. The results in Figure 16 suggest that Adam explores the part of the landscape with smaller values of $\alpha$ and $\beta$ than those for SGDM. The same conclusions can be drawn when comparing SGDM and SGD. These observations might be of the reasons why diagonal preconditioning and momentum are helpful in the training of DL models. The verification of the proposed condition varying the depth of Resnet model can be found in Appendix D.7. The results from Figure 17 demonstrate that the values of $\alpha$ and $\beta$ decrease with the depth of Resnet model, i.e. the model becomes more over-parametrized.

## 6 Conclusion, potential extensions, and limitations.

In this work, we introduce a new class of functions that more accurately characterize loss landscapes of neural networks. In particular, we provide several examples that satisfy the proposed condition, including 2-layer ReLU-neural networks. Additionally, we prove that several optimization algorithms converge under our condition. Finally, we provide extensive empirical verification showing that the proposed $\alpha$-$\beta$-condition holds along the optimization trajectories of various large deep learning models.

It is also possible to further expand convergence guarantees upon the ones presented in Section 4, for instance, by considering momentum [67] which is widely used in practice. However, we defer the exploration of other possible extensions to future research endeavors.

One of the limitations of this work is the empirical validation of the $\alpha$-$\beta$-condition on neural networks. We are only able to verify this condition along the trajectories of specific optimizers. Even when performing checks with many random seeds, we cannot fully observe the entire loss landscape. Additionally, while our theoretical examples demonstrate that the proposed condition holds, we do not provide the most precise theoretical values of $\alpha$ and $\beta$ that satisfy Definition 1. Therefore, in future work, we aim to obtain stronger theoretical guarantees demonstrating that the $\alpha$-$\beta$-condition holds for neural networks with more precise values of $\alpha$ and $\beta$. We also intend to explore how the $\alpha$-$\beta$-condition varies when changing the architecture or the number of parameters (i.e., theoretical exploration of over-parameterization).

## Acknowledgement

Rustem Islamov and Aurelien Lucchi acknowledge the financial support of the Swiss National Foundation, SNF grant No 207392. Antonio Orvieto acknowledges the financial support of the Hector Foundation. The authors thank the anonymous reviewers for their valuable comments and suggestions on improving the paper.

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

# Contents

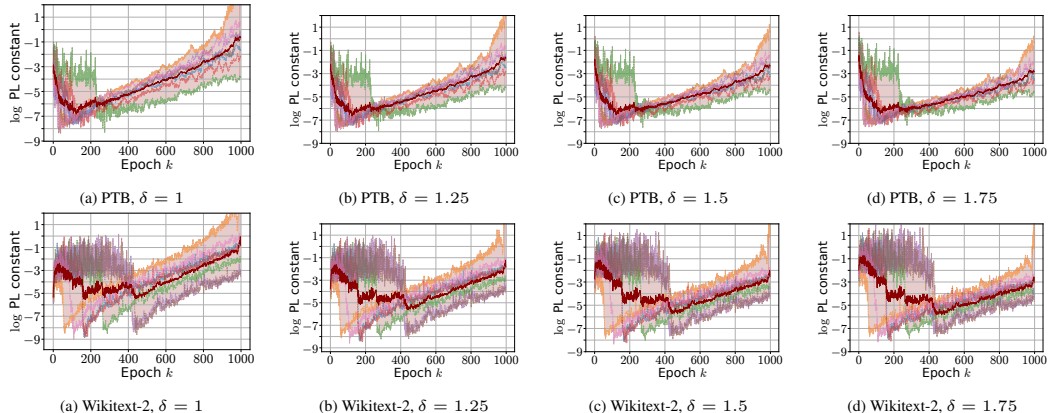

Figure 8: Training of 3 layer LSTM model that shows Aiming condition does not always hold. We plot possible values of PL constant for different powers $\delta$ in (12). We observe that possible values of $\mu$ are of order $10^{-9} - 10^{-7}$ which implies slow theoretical convergence and contradicts practical observations.

## A  Additional explanation on PL assumption

Let us consider the relaxation of PL condition for some $\delta \in [1, 2]$ and $\mu > 0$ (Assumption 3 in [22])

$$\|\nabla f(x)\|^{\delta} \geq (2\mu)^{\delta/2}(f(x) - f^*) \quad \text{for all } x \in \mathbb{R}^d. \tag{12}$$

Taking logarithms of both sides we continue

$$2\log(\|\nabla f(x)\|) - \frac{2}{\delta}\log(f(x) - f^*) \geq \log(2\mu). \tag{13}$$

To satisfy PL condition for some $\delta$, we need to find $\mu$ that satisfies (13) for all iterations. In Figure 8 we plot LHS of (13). We observe that the values of $\mu$ that satisfy (13) for all iterations should be of order $10^{-9} - 10^{-7}$ that leads to slow theoretical convergence [37]. Therefore, we claim that PL condition does not hold globally for neural networks. Nevertheless, it might hold locally around the solution (the value of $\mu$ closer to the end of the training are large enough).

## B  Additional examples

In this section, we list additional examples that satisfy Definition 1 and might have non-optimal stationary points. The functions of this form are typically used to simulate non-convex optimization problems and test the performance of algorithms on small or synthetic datasets [34, 55, 38].

**Example 6.** Let $f_1, f_2 \colon \mathbb{R}^2 \to \mathbb{R}$ be such that

$$f_1(x, y) = \frac{x^2 + y^2}{1 + x^2 + y^2}, \quad f_2(x, y) = \frac{(x-1)^2 + (y-1)^2}{1 + (x-1)^2 + (y-1)^2}, \tag{14}$$

then Definition 1 holds with $\alpha \geq \frac{2}{\min_i \min_{(z,t) \in \mathcal{S}} f_i(z,t)} \approx 41.369325$ and $\beta = \alpha - 1$. Besides, Definition 1 does not hold with $\beta = 0$ not satisfy Definition 1 with $\beta = 0$.

## C  Missing proofs

### C.1  Proofs of examples satisfying definition 1

We highlight that we only show that $\alpha$-$\beta$-condition holds for some $\alpha, \beta$ without giving all possible values of them.

### C.1.1 Proof of example 1

**Example 1.** Let $f, f_1, f_2 \colon \mathbb{R}^2 \to \mathbb{R}$ be such that

$$f = \tfrac{1}{2}(f_1 + f_2) \quad \text{with} \quad f_1(x,y) = \frac{(x+y)^2}{(x+y)^2+1}, \quad f_2(x,y) = \frac{(x+y+1)^2}{(x+y+1)^2+1}, \tag{3}$$

then Definition 1 holds with $\alpha \in [5/2, +\infty)$ and $\beta \in [4\alpha/5, \alpha)$.

*Proof.* Let $(z,t) = \mathrm{Proj}((x,y), \mathcal{S})$. One can show that the set of minimizers of $f$ in this example is $\mathcal{S} = \{(x,y) \colon y = -x - 1/2\}$. Moreover, $f_1^* = 0$ for $x = -y$. We will show that $\alpha$-$\beta$-condition holds for $i = 1$. For $i = 2$ it can be shown in similar way with change of variables. We have

$$\partial_x f_1(x,y) = \partial_y f_1(x,y) = \frac{2(x+y)}{(1+(x+y)^2)^2}.$$

Therefore, we need to show that

$$\frac{2(x+y)}{(1+(x+y)^2)}(x - z + y - t) \geq (\alpha - \beta)\frac{(x+y)^2}{1+(x+y)^2} - \alpha f_i(z,t).$$

Note that $f_i(z,t) = \frac{(z+t)^2}{1+(z+t)^2} = \frac{(-1/2)^2}{1+(-1/2)^2} = \frac{1}{5}$. Moreover, we can compute the projection operator onto $\mathcal{S}$. After simple derivations, the projection can be expressed as

$$\mathrm{Proj}((x,y), \mathcal{S}) = \frac{1}{2}\begin{pmatrix} x - y - 1/2 \\ -x + y - 1/2 \end{pmatrix}.$$

Therefore, we need to satisfy

$$\frac{2(x+y)}{(1+(x+y)^2)^2}(x + y + 1/2) \geq (\alpha - \beta)\frac{(x+y)^2}{1+(x+y)^2} - \frac{\alpha}{5}.$$

This is equivalent to

$$2(x+y)(x+y+1/2) \geq (\alpha - \beta)(x+y)^2(1+(x+y)^2) - \frac{\alpha}{5}(1+(x+y)^2)^2$$

$$\Leftrightarrow 2(x+y)^2 + \frac{\alpha}{5}(1 + 2(x+y)^2 + (x+y)^4) \geq (\alpha - \beta)((x+y)^2 + (x+y)^4) - (x+y). \tag{15}$$

We should satisfy the above for all values of $x + y$. First, we need the coefficient next to $(x+y)^4$ in LHS to be larger or equal of the corresponding coefficient in RHS. This gives us $\frac{\alpha}{5} \geq \alpha - \beta$. This shows that $\beta$ can not be zero. Using $2ab \leq a^2 + b^2$, (15) is satisfied as long as we have

$$2(x+y)^2 + \frac{\alpha}{5}(1 + 2(x+y)^2 + (x+y)^4) \geq (\alpha - \beta)((x+y)^2 + (x+y)^4) + \frac{1}{2}(x+y)^2 + \frac{1}{2}.$$

Therefore, we should satisfy

$$\begin{cases} \text{coefficient next to } (x+y)^4 : \frac{\alpha}{5} \geq \alpha - \beta \\ \text{coefficient next to } (x+y)^2 : 2 + \frac{2\alpha}{5} \geq \alpha - \beta + \frac{1}{2} \\ \text{coefficient next to } 1 : \frac{\alpha}{5} \geq \frac{1}{2} \end{cases}$$

Note that if the first inequality holds, then the second one is true as well. Therefore, from the first and third inequalities we get that the solution of the system of inequalities is $\alpha \geq \frac{5}{2}$ and $\beta \in [4\alpha/5, \alpha]$. $\square$

### C.1.2 Proof of example 2

**Example 2.** Let $f, f_1, f_2 \colon \mathbb{R}^2 \to \mathbb{R}$ be such that

$$f = \tfrac{1}{2}(f_1 + f_2) \quad \text{with} \quad f_1(x,y) = 1 - e^{-x^2 - y^2}, \quad f_2(x,y) = 1 - e^{-(x-2)^2 - (y-2)^2}, \tag{4}$$

then Definition 1 holds for some $\alpha$ and $\beta = \alpha - 8$.

*Proof.* Let $(z, t) = \text{Proj}((x, y), \mathcal{S})$. Again, we show that $\alpha$-$\beta$-condition holds for $f_1$; for $f_2$ it can be proved with the same arguments with change of variables. Note that $f_i^* = 0$ and $f^* > 0$. We also observe that $\mathcal{S}$ contains two points in total, and both of them of the form $(t, t)$ where for one $t$ is close to 0 (it is around $t \approx 0.00067$) and for second $t$ is close to 2 (it is around $t \approx 1.99932$). Besides, note that $c := \min_i \min_{(t,t) \in \mathcal{S}} f_i(t, t) \in (0, 1)$ and $c \approx 9 \cdot 10^{-7}$, which means that we are close to interpolation.

We have in this case

$$\partial_x f_1(x, y) = 2x \exp(-x^2 - y^2), \quad \partial_y f_1(x, y) = 2y \exp(-x^2 - y^2).$$

Therefore, we need to satisfy

$$\exp(-x^2 - y^2)(2x(x - t) + 2y(y - t)) \geq (\alpha - \beta)(1 - \exp(-x^2 - y^2)) - \alpha f_1(z, t). \quad (16)$$

Note that if $\beta = 0$ and $x, y \to +\infty$, then the inequality is obviously can not be satisfied for all $x, y \in \mathbb{R}$. Indeed, assume $\beta = 0$, then since $f_1(z, t) < 1$ and LHS goes to 0 while RHS to $\alpha - \alpha f_1(z, t) > 0$.

Rearranging terms in (16), we should satisfy

$$\exp(-x^2 - y^2)(2x^2 + 2y^2 + \alpha - \beta) + \alpha f_1(z, t) \geq \exp(-x^2 - y^2)(2xt + 2yt) + (\alpha - \beta).$$

Using $2ab \leq a^2 + b^2$, the above is satisfied if the following inequality holds

$$\exp(-x^2 - y^2)(2x^2 + 2y^2 + \alpha - \beta) + \alpha c \geq \exp(-x^2 - y^2)(x^2 + y^2 + 2t^2) + (\alpha - \beta)$$

$$\Leftrightarrow \quad \exp(-x^2 - y^2)(x^2 + y^2 + \alpha - \beta) + \alpha c \geq \exp(-x^2 - y^2) \cdot 2t^2 + (\alpha - \beta).$$

Since $t \in (0, 2)$ we can choose $\alpha - \beta = 8$, so that $x^2 + y^2 - 2t^2 + \alpha - \beta \geq x^2 + y^2 \geq 0$. Finally, we choose $\alpha \geq \frac{8}{c}$, so that $\alpha c \geq \alpha - \beta = 8$. We can estimate that $\alpha \gtrsim 72 \cdot 10^7$. □

### C.1.3 Proof of Example 3

**Example 3.** Let $f, f_1, f_2 : \mathbb{R}^2 \to \mathbb{R}$ be such that

$$f = \tfrac{1}{2}(f_1 + f_2) \quad \text{with} \quad f_1(x, y) = \tfrac{1 + x^2 + y^2}{4 + x^2 + y^2}, \quad f_2(x, y) = \tfrac{(x - 2.5)^2 + (y - 2.5)^2}{4 + (y - 2.5)^2 + (y - 2.5)^2}, \quad (6)$$

then Definition 1 holds for some $\alpha$ and $\beta = \alpha - 1$.

*Proof.* Let $(z, t) = \text{Proj}((x, y), \mathcal{S})$. For this example, we have $f_1^* = \frac{1}{4}$ (achieved at $(0, 0)$), $f_2^* = 0$ (achieved at $(2.5, 2.5)$), and $f^* \approx = 0.408$ (achieved at $(2.471, 2.471)$). Besides, we have $f(0, 0) = 0.587963$ and $f(2.5, 2.5) = 0.409091$. Besides, $c := \min_i \min_{(z,t) \in \mathcal{S}} f_i(z, t) = f_2(2.471, 2.471) = 0.00167918$. Let us show that $\alpha$-$\beta$-condition holds for $f_1$; for $f_2$ it can be shown similarly. We have

$$\partial_x f_1(x, y) = \frac{6x}{(4 + x^2 + y^2)^2}, \quad \partial_y f_1(x, y) = \frac{6y}{(4 + x^2 + y^2)^2}.$$

Therefore, we need to satisfy for some $\alpha$ and $\beta$

$$\frac{6}{(4 + x^2 + y^2)^2}(x(x - z) + y(y - t)) \geq (\alpha - \beta)\frac{1 + x^2 + y^2}{4 + x^2 + y^2} - f_1(z, t)$$

$$\Leftrightarrow \quad 6x^2 + 6y^2 - 6xz - 6yt \geq (\alpha - \beta)(1 + x^2 + y^2)(4 + x^2 + y^2) - \alpha f_1(z, t)(4 + x^2 + y^2)^2$$

$$\Leftrightarrow \quad 6(x^2 + y^2) + \alpha f_1(z, t)(16 + 8(x^2 + y^2) + (x^2 + y^2)^2) \geq 6xz + 6yt$$
$$+ (\alpha - \beta)(4 + 5(x^2 + y^2) + (x^2 + y^2)^2).$$

The above is satisfied for all $x, y \in \mathbb{R}$ and $\alpha - \beta = 1$ if we have

$$6(x^2 + y^2) + \alpha f_1(z, t)(16 + 8(x^2 + y^2) + (x^2 + y^2)^2) \geq 3(x^2 + y^2) + 3z^2 + 3t^2$$
$$+ (4 + 5(x^2 + y^2) + (x^2 + y^2)^2).$$

To satisfy the above for all $x, y \in \mathbb{R}$ we should have

$$\begin{cases} 16\alpha f_1(z, t) & \geq 4 + 3(z^2 + t^2) \\ 6 + 8\alpha f_1(z, t) & \geq 3 + 5 \\ \alpha f_1(z, t) & \geq 1, \end{cases}$$

where $z = t = 2.471$, and $f_1(z, t) = 0.0016718$. We can satisfy $\alpha$-$\beta$-condition with $\alpha \gtrsim 1512.4586$. □

### C.1.4 Proof of example 6

**Example 6.** Let $f_1, f_2 \colon \mathbb{R}^2 \to \mathbb{R}$ be such that

$$f_1(x,y) = \frac{x^2 + y^2}{1 + x^2 + y^2}, \quad f_2(x,y) = \frac{(x-1)^2 + (y-1)^2}{1 + (x-1)^2 + (y-1)^2}, \tag{14}$$

then Definition 1 holds with $\alpha \geq \frac{2}{\min_i \min_{(z,t) \in \mathcal{S}} f_i(z,t)} \approx 41.369325$ and $\beta = \alpha - 1$. Besides, Definition 1 does not hold with $\beta = 0$ not satisfy Definition 1 with $\beta = 0$.

*Proof.* Let $(z,t) = \mathrm{Proj}((x,y), \mathcal{S})$. We show that $\alpha$-$\beta$-condition holds for $f_1$; for $f_2$ the proof is similar with change of variables. In this case, $f_i^* = 0$, while $f^* > 0$ ($f^* \approx 0.316988$ at points $(0.159375, 0.159375)$ and $(0.840625, 0.840625)$). Besides, we have $c := \min_i \min_{(z,t) \in \mathcal{S}} f_i(z,t) \approx 0.048345 > 0$. Moreover, We have

$$\partial_x f_1(x,y) = \frac{2x}{(1 + x^2 + y^2)^2}, \quad \partial_y f_1(x,y) = \frac{2y}{(1 + x^2 + y^2)^2}.$$

Therefore, we need to satisfy for some $\alpha$ and $\beta$

$$\frac{1}{(1 + x^2 + y^2)^2}(2x(x - z) + 2y(y - t)) \geq (\alpha - \beta)\frac{x^2 + y^2}{1 + x^2 + y^2} - \alpha f_1(z,t)$$

$$\Leftrightarrow \quad 2x^2 + 2y^2 - 2xz - 2yt \geq (\alpha - \beta)(x^2 + y^2)(1 + x^2 + y^2) - \alpha f_1(z,t)(1 + x^2 + y^2)^2$$

$$\Leftrightarrow \quad 2x^2 + 2y^2 + \alpha f_1(z,t)(1 + 2(x^2 + y^2) + (x^2 + y^2)^2) \geq 2xz + 2yt +$$
$$(\alpha - \beta)(x^2 + y^2)(1 + x^2 + y^2).$$

Note that if we take $\beta = 0$ and $x, y \to +\infty$, then LHS grows as $\alpha f_1(z,t)(x^2 + y^2)^2$ and RHS grows as $\alpha(x^2 + y^2)^2$, therefore in the limit the RHS is larger. This implies that $\beta \neq 0$. Using $2ab \leq a^2 + b^2$ the above is satisfied if we have

$$(2 + 2\alpha f_1(z,t))(x^2 + y^2) + \alpha f_1(z,t)(x^2 + y^2)^2 + \alpha f_1(z,t) \geq (x^2 + y^2)(1 + \alpha - \beta) +$$
$$z^2 + t^2 + (\alpha - \beta)(x^2 + y^2)^2. \tag{17}$$

Let us take $\alpha - \beta = 1$ and $\alpha \geq \frac{2}{\min_i \min_{(z,t) \in \mathcal{S}} f_i(z,t)} \approx 41.369325$. With this choice, we have

$$z^2 + t^2 \leq 2 \leq \alpha f_1(z,t),$$
$$1 + \alpha - \beta = 2 \leq 2 + 2\alpha f_1(z,t),$$
$$\alpha - \beta = 1 < 2 \leq \alpha f_1(z,t).$$

Therefore, LHS is always larger than RHS in (17). $\qquad \square$

### C.1.5 Proof of example 4

**Example 4.** Let $f_i, f_{ij}$ be such that

$$f(W,S) = \frac{1}{2nm}\|X - W^\top S\|_F^2 = \frac{1}{2nm}\sum_{i,j}(X_{ij} - w_i^\top s_j)^2, \quad f_{ij}(W,S) = \frac{1}{2}(X_{ij} - w_i^\top s_j)^2, \tag{7}$$

where $X \in \mathbb{R}^{n \times m}, W = (w_i)_{i=1}^n \in \mathbb{R}^{k \times n}, S = (s_j)_{j=1}^m \in \mathbb{R}^{k \times m}$, and $\mathrm{rank}(X) = r \geq k$. We assume that $X$ is generated using matrices $W^*$ and $S^*$ with non-zero additive noise that minimize empirical loss, namely, $X = (W^*)^\top S^* + (\varepsilon_{ij})_{i \in [n], j \in [m]}$ where $W^*, S^* = \mathrm{argmin}_{W,S} f(W,S)$. Let $\mathcal{X}$ be any bounded set that contains $\mathcal{S}$. Then Definition 1 is satisfied with $\alpha = \beta + 1$ and some $\beta > 0$.

*Proof.* Since $k \leq r$, then the matrix factorization problem has a unique solution which can be found from SVD decomposition of $X$ [76].

We have

$$\nabla_{w_i} f_{ij}(W,S) = (w_i^\top s_j - X_{ij})s_j, \quad \nabla_{s_j} f_{ij}(W,S) = (w_i^\top s_j - X_{ij})w_i.$$

Therefore, we need to show that

$$\langle (w_i^\top s_j - X_{ij}) s_j, w_i - w_i^* \rangle + \langle (w_i^\top s_j - X_{ij}) w_i, s_j - s_j^* \rangle \geq \frac{\alpha - \beta}{2} (X_{ij} - w_i^\top s_j)^2$$
$$- \frac{\alpha}{2} (X_{ij} - (w_i^*)^\top s_j^*)^2,$$

since $f_{ij}^* = 0$. Since $f_{ij}$ is convex w.r.t. $w_i$, the above holds if we have

$$\frac{1}{2}(w_i^\top s_j - X_{ij})^2 - \frac{1}{2}(s_j^\top w_i^* - X_{ij})^2 + \frac{1}{2}(w_i^\top s_j - X_{ij})^2 - \frac{1}{2}(w_i^\top s_j^* - X_{ij})^2 \geq$$
$$\frac{\alpha - \beta}{2}(X_{ij} - w_i^\top s_j)^2 - \frac{\alpha}{2}(X_{ij} - (w_i^*)^\top s_j^*)^2.$$

Let us take $\alpha = \beta + 1$, then we can simplify the above as follows

$$\frac{1}{2}(w_i^\top s_j - X_{ij})^2 + \frac{\alpha}{2}(X_{ij} - (w_i^*)^\top s_j^*)^2 \geq \frac{1}{2}(s_j^\top w_i^* - X_{ij})^2 + \frac{1}{2}(w_i^\top s_j^* - X_{ij})^2 \quad (18)$$

Since we consider $\mathcal{X}$ to be bounded, then there exist $r \geq 0$ such that $\|s_j\|, \|w_i\| \leq r$ for all $i$ and $j$. Therefore, the RHS in (18) is bounded by some constant $C$. From the data generation, we have that $c = \min_{ij}(X_{ij} - (w_i^*)^\top s_j^*)^2 = \min_{ij} \varepsilon_{ij}^2 > 0$. Therefore, we can take $\alpha \geq \frac{2C}{c}$ to verify (18). $\qquad \square$

### C.1.6 Proof of example 5

**Example 5.** Consider training a two-layer neural network with a logistic loss
$$f = \frac{1}{n}\sum_{i=1}^n f_i, \quad f_i(W, v) = \phi(y_i \cdot v^\top \sigma(Wx_i)) + \lambda_1 \|v\|^2 + \lambda_2 \|W\|_F^2 \quad (8)$$
for a classification problem where $\phi(t) := \log(1 + \exp(-t))$, $W \in \mathbb{R}^{k \times d}, v \in \mathbb{R}^k$, $\sigma$ is a ReLU function applied coordinate-wise, $y_i \in \{-1, +1\}$ is a label and $x_i \in \mathbb{R}^d$ is a feature vector. Let $\mathcal{X}$ be any bounded set that contains $\mathcal{S}$. Then the $\alpha$-$\beta$-condition holds in $\mathcal{X}$ for some $\alpha \geq 1$ and $\beta = \alpha - 1$.

*Proof.* Let $(Z, z) := \mathrm{Proj}((W, v), \mathcal{S})$. We have the following derivations for gradients
$$\mathbb{R}^{k \times d} \ni \nabla_W f_i(W, v) = \phi'(y_i v^\top \sigma(Wx_i)) \cdot y_i (v \circ \mathbb{1}_{Wx_i \geq 0}) x_i^\top + 2\lambda_2 W,$$
$$\mathbb{R}^k \ni \nabla_v f_i(W, v) = \phi'(y_i v^\top \sigma(Wx_i)) \cdot y_i \sigma(Wx_i) = \phi'(y_i v^\top \sigma(Wx_i)) \cdot y_i (Wx_i) \circ \mathbb{1}_{Wx_i \geq 0}$$
$$+ 2\lambda_1 v$$
$$= \phi'(y_i v^\top \sigma(Wx_i)) \cdot y_i (Wx_i) \circ e_w + 2\lambda_1 v,$$
where we denote $e_w := \mathbb{1}_{Wx_i \geq 0} \in \mathbb{R}^k$. Besides, we denote $e_z := \mathbb{1}_{Zx_i \geq 0} \in \mathbb{R}^k$. Note that the optimal value of $f_i^* > 0$ because of the L2 regularization.

Note that we have the following relations
$$v^\top \sigma(Wx_i) = \langle v, (Wx_i) \circ \mathbb{1}_{Wx_i \geq 0} \rangle = \langle v \circ \mathbb{1}_{Wx_i \geq 0}, Wx_i \rangle = \langle v \circ e_w, Wx_i \rangle. \quad (19)$$

Note that $\mathcal{S}$ is bounded because of the L2 regularization. Since we assume that the interpolation does not hold, then $f$ is always strictly larger than $f_i^*$ in $\mathcal{X}$, and due to continuity there exists $c := \min_{i \in [n]} \min_{(Z,z) \in \mathcal{S}} f_i(Z, z) > 0$.

We need to show that there exist some $\alpha$ and $\beta$ such that
$$\langle \nabla_W f_i(W, v), W - Z \rangle_F + \langle \nabla_v f_i(W, v), v - z \rangle$$
$$\geq \alpha(f_i(W, v) - f_i(Z, z)) - \beta(f_i(W, v) - f_i^*)$$
$$\Leftrightarrow \phi'(y_i(v \circ e_w)^\top Wx_i)y_i \left( \langle (v \circ e_w)x_i^\top, W - Z \rangle + \langle (Wx_i) \circ e_w, v - z \rangle \right) + 2\lambda_1 \langle v, v - z \rangle$$
$$+ 2\lambda_2 \langle W, W - Z \rangle$$
$$\geq \alpha \left[ \phi(y_i(v \circ e_w)^\top Wx_i) + \lambda_1 \|v\|^2 + \lambda_2 \|W\|_F^2 - \phi(y_i(z \circ e_z)^\top Zx_i) - \lambda_1 \|z\|^2 - \lambda_2 \|Z\|_F^2) \right]$$
$$- \beta \left[ \phi(y_i(v \circ e_w)^\top Wx_i) + \lambda_1 \|v\|^2 + \lambda_2 \|W\|_F^2 - f_i^* \right]$$
$$\Leftrightarrow \phi'(y_i(v \circ e_w)^\top Wx_i) \left( [y_i(v \circ e_w)^\top Wx_i - y_i(v \circ e_w)^\top Zx_i] + [y_i(v \circ e_w)^\top Wx_i - y_i(z \circ e_w)^\top Wx_i] \right)$$
$$+ 2\lambda_1 \langle v, v - z \rangle + 2\lambda_2 \langle W, W - Z \rangle$$
$$\geq \alpha \left[ \phi(y_i(v \circ e_w)^\top Wx_i) + \lambda_1 \|v\|^2 + \lambda_2 \|W\|_F^2 - \phi(y_i(z \circ e_z)^\top Zx_i) - \lambda_1 \|z\|^2 - \lambda_2 \|Z\|_F^2 \right]$$
$$- \beta \left[ \phi(y_i(v \circ e_w)^\top Wx_i) + \lambda_1 \|v\|^2 + \lambda_2 \|W\|^2 - f_i^* \right], \quad (20)$$

---

**Algorithm 1** SGD with constant stepsize

---
1: **Input:** Stepsize $\gamma$
2: **for** $k = 0, 1, 2, \ldots, K - 1$ **do**
3:     Sample $i_k \sim \text{Unif}[n]$ and compute $\nabla f_{i_k}(x^k)$
4:     Update model
$$x^{k+1} = x^k - \gamma \nabla f_{i_k}(x^k)$$
5: **end for**

---

where we use (19). Since $\phi$ is convex, then we have $\phi'(x)(x - y) \geq \phi(x) - \phi(y)$. Therefore, using (19) again, we get that (20) is satisfied if we have

$$2\phi(y_i(v \circ e_w)^\top W x_i) - \phi(y_i(v \circ e_w)^\top Z x_i) - \phi(y_i(z \circ e_w)^\top W x_i) + \lambda_1 \langle v, v - z \rangle + \lambda_2 \langle W, W - Z \rangle$$
$$\geq \alpha \left[ \phi(y_i(v \circ e_w)^\top W x_i) + \lambda_1 \|v\|^2 + \lambda_2 \|W\|_{\text{F}}^2 - \phi(y_i(z \circ e_z)^\top Z x_i) - \lambda_1 \|z\|^2 - \lambda_2 \|Z\|_{\text{F}}^2 \right]$$
$$- \beta \left[ \phi(y_i(v \circ e_w)^\top W x_i) + \lambda_1 \|v\|^2 + \lambda_2 \|W\|_{\text{F}}^2 - f_i^* \right].$$

Now we take $\alpha = \beta + 1$ and simplify the above as follows

$$\phi(y_i(v \circ e_w)^\top W x_i) + \alpha \phi(y_i(z \circ e_w)^\top Z x_i)) + \lambda_1(2\langle v, v - z \rangle - \|v\|^2 - \|z\|^2)$$
$$+ \lambda_2(2\langle W, W - Z \rangle - \|W\|_{\text{F}}^2 - \|Z\|_{\text{F}}^2) + (\alpha - 1)(\lambda_1 \|z\|^2 + \lambda_2 \|Z\|_{\text{F}}^2)$$
$$\geq \phi(y_i(v \circ e_w)^\top Z x_i) + \phi(y_i(z \circ e_w)^\top W x_i) + (\alpha - 1)f_i^*. \tag{21}$$

The above is satisfied if we have

$$\phi(y_i(v \circ e_w)^\top W x_i) + \alpha c + \lambda_1 \|v - z\|^2 + \lambda_2 \|W - Z\|_{\text{F}}^2 + (\alpha - 1)(\|z\|^2 + \|Z\|_{\text{F}}^2)$$
$$\geq \phi(y_i(v \circ e_w)^\top Z x_i) + \phi(y_i(z \circ e_w)^\top W x_i) + (\alpha - 1)f_i^*, \tag{22}$$

because we also have $\phi(y_i(z \circ e_z)^\top Z x_i) \geq c$ for all $i$ and $(Z, z) \in \mathcal{S}$. Since $(W, v) \in \mathcal{X}$, there exist constants $R, r \geq 0$ such that $\|v\| \leq r$ and $\|W\| \leq R$, and RHS in (22) is bounded by

$$2 \max_{i \in [n]} \log(1 + \exp(Rr\|x_i\|)) \geq \phi(y_i(v \circ e_w)^\top Z x_i) + \phi(y_i(z \circ e_w)^\top W x_i).$$

Therefore, we can take $\alpha \geq \left\{ \frac{2 \max_{i \in [n]} \log(1 + \exp(Rr\|x_i\|))}{c - f_i^*}, 1 \right\}$ and $\beta = \alpha - 1$.  □

**Remark 3.** The result suggest that the choice $\beta = 0$ is possible only if constants $r$ and $R$ are small, i.e. locally around $\mathcal{S}$ only. In order to satisfy $\alpha$-$\beta$-condition in larger set $\mathcal{X}$, one needs to choose $\beta > 0$.

## C.2 Convergence of optimization algorithms under $\alpha$-$\beta$-condition

### C.2.1 Convergence of SGD

**Constant stepsize.** In this section, we present the proof of convergence of SGD with constant stepsize under $\alpha$-$\beta$-condition for completeness of the presentation.

**Theorem 1.** *Assume that Assumptions 1-2 hold. Then the iterates of SGD (Alg. 1) with stepsize $\gamma \leq \frac{\alpha - \beta}{2L}$ satisfy*

$$\min_{0 \leq k < K} \mathbb{E}\left[ f(x^k) - f^* \right] \leq \frac{\mathbb{E}\left[ \text{dist}(x^0, \mathcal{S})^2 \right]}{K} \frac{1}{\gamma(\alpha - \beta)} + \frac{2L\gamma}{\alpha - \beta}\sigma_{\text{int}}^2 + \frac{2\beta}{\alpha - \beta}\sigma_{\text{int}}^2. \tag{9}$$

*Proof.* Let $x_p^k \in \text{Proj}(x^k, \mathcal{S})$ satisfies Definition 1. Using smoothness we have

$$
\begin{aligned}
\mathbb{E}_k \left[ \text{dist}(x^{k+1}, \mathcal{S})^2 \right] &\leq \mathbb{E}_k \left[ \|x^{k+1} - x_p^k\|^2 \right] \\
&= \text{dist}(x^k, \mathcal{S})^2 - 2\gamma \mathbb{E}_k \left[ \langle \nabla f_{i_k}(x^k), x^k - x_p^k \rangle \right] + \gamma^2 \mathbb{E}_k \left[ \|\nabla f_{i_k}(x^k)\|^2 \right] \\
&\overset{(i)}{\leq} \text{dist}(x^k, \mathcal{S})^2 - 2\alpha\gamma \mathbb{E}_k \left[ f_{i_k}(x^k) - f_{i_k}(x_p^k) \right] + 2\beta\gamma \mathbb{E}_k \left[ f_{i_k}(x^k) - f_{i_k}^* \right] \\
&\quad + 2L\gamma^2 \mathbb{E}_k \left[ f_{i_k}(x^k) - f_{i_k}^* \right] \\
&= \text{dist}(x^k, \mathcal{S})^2 - 2\alpha\gamma \mathbb{E}_k \left[ f_{i_k}(x^k) - f_{i_k}(x_p^k) \right] \\
&\quad + 2\gamma(\beta + L\gamma) \mathbb{E}_k \left[ (f_{i_k}(x^k) - f_{i_k}(x_p^k)) + (f_{i_k}(x_p^k) - f_{i_k}^*) \right],
\end{aligned}
$$

where $(i)$ holds because of the $\alpha$-$\beta$-condition and smoothness. Now we need to choose a stepsize $\gamma \leq \frac{\alpha - \beta}{2L}$ to get

$$
\begin{aligned}
\mathbb{E}_k \left[ \text{dist}(x^{k+1}, \mathcal{S})^2 \right] \leq\ & \text{dist}(x^k, \mathcal{S})^2 - \gamma(\alpha - \beta) \mathbb{E}_k \left[ f_{i_k}(x^k) - f_{i_k}(x_p^k) \right] \\
& + 2\gamma(\beta + L\gamma) \mathbb{E}_k \left[ f_{i_k}(x_p^k) - f_{i_k}^* \right].
\end{aligned}
$$

Taking full expectation, noting that $x_p^k$ is independent of the randomness of $i_k$, and performing simple derivations, we get

$$
\min_{0 \leq k < K} \mathbb{E}\left[ f(x^k) - f^* \right] \leq \frac{\mathbb{E}\left[ \text{dist}(x^0, \mathcal{S})^2 \right]}{K} \frac{1}{\gamma(\alpha - \beta)} + \frac{2L\gamma}{\alpha - \beta} \sigma_{\text{int}}^2 + \frac{2\beta}{\alpha - \beta} \sigma_{\text{int}}^2. \tag{23}
$$

$\square$

**Decreasing stepsize.** Now we present the results with decreasing stepsize.

**Theorem 4.** *Assume that Assumptions 1-2 hold. Then the iterates of* SGD *with decreasing stepsize* $\gamma_k = \frac{\gamma_0}{\sqrt{k+1}}$ *where* $\gamma_0 \leq \frac{\alpha - \beta}{2L}$ *satisfy*

$$
\begin{aligned}
\min_{0 \leq k < K} \mathbb{E}\left[ f(x^k) - f^* \right] \leq\ & \frac{5\mathbb{E}\left[ \text{dist}(x^0, \mathcal{S})^2 \right]}{4(\alpha - \beta)\gamma_0 \sqrt{K}} + \frac{5\gamma_0 L \sigma_{\text{int}}^2}{\alpha - \beta} \frac{\log(K+1)}{\sqrt{K}} + \frac{2\beta}{\alpha - \beta} \sigma_{\text{int}}^2 \quad (24) \\
=\ & \widetilde{\mathcal{O}} \left( \frac{1}{\sqrt{K}} + \beta \sigma_{\text{int}}^2 \right). \tag{25}
\end{aligned}
$$

*Proof.* Similarly to the proof of constant stepsize SGD we can obtain

$$
\begin{aligned}
\mathbb{E}_k \left[ \text{dist}(x^{k+1}, \mathcal{S})^2 \right] \leq\ & \text{dist}(x^k, \mathcal{S})^2 - \gamma_k(\alpha - \beta) \mathbb{E}_k \left[ f_{i_k}(x^k) - f_{i_k}(x_p^k) \right] \\
& + 2\gamma_k(\beta + L\gamma_k) \mathbb{E}_k \left[ f_{i_k}(x_p^k) - f_{i_k}^* \right],
\end{aligned}
$$

since $\gamma_k \leq \gamma_0 \leq \frac{\alpha - \beta}{2L}$ for all $k$. Now we follow standard proof techniques [24] for decreasing stepsize. Taking full expectation and dividing both sides by $\alpha - \beta$ we get

$$
\gamma_k \mathbb{E}\left[ f(x^k) - f^* \right] \leq \frac{\mathbb{E}\left[ \text{dist}(x^k, \mathcal{S})^2 \right] - \mathbb{E}\left[ \text{dist}(x^{k+1}, \mathcal{S})^2 \right]}{\alpha - \beta} + \frac{2\beta}{\alpha - \beta} \gamma_k \sigma_{\text{int}}^2 + \frac{2L}{\alpha - \beta} \gamma_k^2 \sigma_{\text{int}}^2.
$$

Summing the above from iteration $0$ to $K-1$ leads to

$$
\sum_{k=0}^{K-1} \gamma_k \mathbb{E}\left[ f(x^k) - f^* \right] \leq \frac{\mathbb{E}\left[ \text{dist}(x^0, \mathcal{S})^2 \right]}{\alpha - \beta} + \frac{2\beta}{\alpha - \beta} \sigma_{\text{int}}^2 \sum_{k=0}^{K-1} \gamma_k + \frac{2L}{\alpha - \beta} \sigma_{\text{int}}^2 \sum_{k=0}^{K-1} \gamma_k^2.
$$

Therefore, we get

$$
\min_{0 \geq k < K} \mathbb{E}\left[ f(x^k) - f^* \right] \leq \frac{\mathbb{E}\left[ \text{dist}(x^0, \mathcal{S})^2 \right]}{(\alpha - \beta) \sum_{k=0}^{K-1} \gamma_k} + \frac{2\beta}{\alpha - \beta} \sigma_{\text{int}}^2 + \frac{2L}{\alpha - \beta} \sigma_{\text{int}}^2 \frac{\sum_{k=0}^{K-1} \gamma_k^2}{\sum_{k=0}^{K_1} \gamma_k}.
$$

Using the results of Theorem 5.7 [24] we get

$$
\sum_{k=0}^{K-1} \gamma_k^2 \leq 2\gamma_0^2 \log(K+1), \quad \sum_{k=0}^{K-1} \gamma_k \geq \frac{4\gamma_0}{5} \sqrt{K}.
$$

Therefore, the final rate we get is

$$
\begin{aligned}
\min_{0 \geq k < K} \mathbb{E}\left[ f(x^k) - f^* \right] \leq\ & \frac{\mathbb{E}\left[ \text{dist}(x^0, \mathcal{S})^2 \right]}{(\alpha - \beta)\frac{4}{5}\gamma_0 \sqrt{K}} + \frac{2\beta}{\alpha - \beta} \sigma_{\text{int}}^2 + \frac{2L}{\alpha - \beta} \sigma_{\text{int}}^2 \frac{2\gamma_0^2 \log(K+1)}{\frac{4\gamma_0}{5} \sqrt{K}} \\
=\ & \frac{5\mathbb{E}\left[ \text{dist}(x^0, \mathcal{S})^2 \right]}{4(\alpha - \beta)\gamma_0 \sqrt{K}} + \frac{5\gamma_0 L \sigma_{\text{int}}^2}{\alpha - \beta} \frac{\log(K+1)}{\sqrt{K}} + \frac{2\beta}{\alpha - \beta} \sigma_{\text{int}}^2.
\end{aligned}
$$

$\square$

---

**Algorithm 2** $\text{SPS}_{\max}$: Stochastic Polyak Stepsize

---

1: **Input:** Stepsize parameter $c$ and stepsize upper bound $\gamma_b$
2: **for** $k = 0, 1, 2, \ldots, K - 1$ **do**
3:     Sample $i_k \sim \text{Unif}[n]$ and compute $\nabla f_{i_k}(x^k)$
4:     Compute Polyak stepsize

$$\gamma_k := \min \left\{ \frac{f_{i_k}(x^k) - f_{i_k}^*}{c\|\nabla f_{i_k}(x^k)\|^2}, \gamma_b \right\}$$

5:     Update model

$$x^{k+1} = x^k - \gamma_k \nabla f_{i_k}(x^k)$$

6: **end for**

---

### C.2.2   Convergence of SGD with Polyak Stepsize

**Constant stepsize parameter.** In this section, we present the proof of convergence of SGD with Polyak stepsize (with constant stepsize parameters) under $\alpha$-$\beta$-condition for completeness of the presentation.

**Lemma 1** (Lemma from [53]). The $\text{SPS}_{\max}$ stepsize satisfy

$$\gamma_k^2 \|\nabla f_{i_k}(x^k)\|^2 \leq \frac{\gamma_k}{c}(f_{i_k}(x^k) - f_{i_k}^*). \tag{26}$$

**Lemma 2** (Lemma from [53]). Assume each $f_i$ is $L$-smooth, then $\text{SPS}_{\max}$ stepsize satisfy

$$\gamma_{\min} := \min \left\{ \frac{1}{2cL}, \gamma_b \right\} \leq \gamma_k \leq \gamma_b. \tag{27}$$

Now we present the proof of Theorem 2 with constant stepsize parameters.

**Theorem 2.** *Assume that Assumptions 1-2 hold. Then the iterates of $\text{SPS}_{\max}$ (Alg. 2) with a stepsize hyperparameter $c > \frac{1}{2(\alpha-\beta)}$ satisfy*

$$\min_{0 \leq k < K} \mathbb{E}\left[f(x^k) - f^*\right] \leq \frac{c_1}{K}\mathbb{E}\left[\text{dist}(x^0, \mathcal{S})^2\right] + 2\alpha c_1 \gamma_b \sigma_{\text{int}}^2, \tag{10}$$

*where $\gamma_{\min} := \min\{1/2cL, \gamma_b\}$ and $c_1 := \frac{c}{\gamma_{\min}(2(\alpha-\beta)c-1)}$.*

*Proof.* Let $x_p^k \in \text{Proj}(x^k, \mathcal{S})$ that satisfies Definition 1, then we have

$$
\begin{aligned}
\mathbb{E}_k\left[\text{dist}(x^{k+1}, \mathcal{S})^2\right] &\leq \mathbb{E}_k\left[\|x^{k+1} - x_p^k\|^2\right] \\
&= \|x^k - x_p^k\|^2 - 2\mathbb{E}_k\left[\gamma_k \left\langle \nabla f_{i_k}(x^k), x^k - x_p^k \right\rangle\right] + \mathbb{E}_k\left[\gamma_k^2 \|\nabla f_{i_k}(x^k)\|^2\right] \\
&\stackrel{(i)}{\leq} \text{dist}(x^k, \mathcal{S})^2 - 2\alpha\mathbb{E}_k\left[\gamma_k(f_{i_k}(x^k) - f_{i_k}(x_p^k))\right] \\
&\quad + 2\beta\mathbb{E}_k\left[\gamma_k(f_{i_k}(x^k) - f_{i_k}^*)\right] + \frac{1}{c}\mathbb{E}_k\left[\gamma_k(f_{i_k}(x^k) - f_{i_k}^*)\right] \\
&= \text{dist}(x^k, \mathcal{S})^2 - 2\alpha\mathbb{E}_k\left[\gamma_k([f_{i_k}(x^k) - f_{i_k}^*] - [f_{i_k}(x_p^k) - f_{i_k}^*])\right] \\
&\quad + 2\beta\mathbb{E}_k\left[\gamma_k(f_{i_k}(x^k) - f_{i_k}^*)\right] + \frac{1}{c}\mathbb{E}_k\left[\gamma_k(f_{i_k}(x^k) - f_{i_k}^*)\right] \\
&= \text{dist}(x^k, \mathcal{S})^2 - \mathbb{E}_k\left[\gamma_k\left(2\alpha - 2\beta - \frac{1}{c}\right)(f_{i_k}(x^k) - f_{i_k}^*)\right] \\
&\quad + 2\alpha\mathbb{E}_k\left[\gamma_k(f_{i_k}(x_p^k) - f_{i_k}^*)\right],
\end{aligned}
$$

where $(i)$ follows from Lemma 1 and $\alpha$-$\beta$-condition . Now, since $c > \frac{1}{2(\alpha-\beta)}$, we get that $2\alpha - 2\beta - 1/c > 0$. Therefore, using Lemma 2 and the fact that $f_{i_k}^* \leq f_{i_k}(x_p^k)$ and $f_{i_k}^* \leq f_{i_k}(x^k)$ we get

$$
\begin{aligned}
\mathbb{E}_k \left[\mathrm{dist}(x^{k+1}, \mathcal{S})^2\right] &\leq \mathrm{dist}(x^k, \mathcal{S})^2 - \gamma_{\min}\left(2\alpha - 2\beta - \frac{1}{c}\right)\mathbb{E}_k\left[f_{i_k}(x^k) - f_{i_k}^*\right] \\
&\quad + 2\alpha\gamma_{\mathrm{b}}\mathbb{E}_k\left[(f_{i_k}(x_p^k) - f_{i_k}^*)\right] \\
&= \mathrm{dist}(x^k, \mathcal{S})^2 - \gamma_{\min}\left(2\alpha - 2\beta - \frac{1}{c}\right)\mathbb{E}_k\left[f_{i_k}(x^k) - f_{i_k}(x_p^k)\right] \\
&\quad - \gamma_{\min}\left(2\alpha - 2\beta - \frac{1}{c}\right)\mathbb{E}_k\left[f_{i_k}(x_p^k) - f_{i_k}^*\right] + 2\alpha\gamma_{\mathrm{b}}\mathbb{E}_k\left[(f_{i_k}(x_p^k) - f_{i_k}^*)\right] \\
&\leq \mathrm{dist}(x^k, \mathcal{S})^2 - \gamma_{\min}\left(2\alpha - 2\beta - \frac{1}{c}\right)\mathbb{E}_k\left[f_{i_k}(x^k) - f_{i_k}(x_p^k)\right] \\
&\quad + 2\alpha\gamma_{\mathrm{b}}\mathbb{E}_k\left[(f_{i_k}(x_p^k) - f_{i_k}^*)\right].
\end{aligned}
$$

Therefore, noticing that $\mathbb{E}_k\left[f_{i_k}(x_p^k)\right] = f^*$ since $x_p^k$ is independent of $i_k$, we have

$$
\gamma_{\min}\left(2\alpha - 2\beta - \frac{1}{c}\right)\mathbb{E}_k\left[f(x^k) - f^*\right] \leq \mathrm{dist}(x^k, \mathcal{S})^2 - \mathbb{E}_k\left[\mathrm{dist}(x^{k+1}, \mathcal{S})^2\right] + 2\alpha\gamma_b\sigma_{\mathrm{int}}^2.
$$

Dividing both sides by $\gamma_{\min}(2\alpha - 2\beta - 1/c)$ and taking full expectation, we get

$$
\begin{aligned}
\mathbb{E}\left[f(x^k) - f^*\right] &\leq \frac{c}{\gamma_{\min}(2(\alpha-\beta)c - 1)}\left(\mathbb{E}\left[\mathrm{dist}(x^k, \mathcal{S})^2\right] - \mathbb{E}\left[\mathrm{dist}(x^{k+1}, \mathcal{S})^2\right]\right) \\
&\quad + \frac{2\alpha c\gamma_{\mathrm{b}}}{\gamma_{\min}(2(\alpha-\beta)c - 1)}\sigma_{\mathrm{int}}^2.
\end{aligned}
$$

Averaging for $k \in \{0, \ldots, K-1\}$ we get

$$
\begin{aligned}
\min_{0 \leq k < K}\mathbb{E}\left[f(x^k) - f^*\right] &\leq \frac{c}{\gamma_{\min}(2(\alpha-\beta)c - 1)K}\mathbb{E}\left[\mathrm{dist}(x^0, \mathcal{S})^2\right] \\
&\quad + \frac{2\alpha c\gamma_{\mathrm{b}}}{\gamma_{\min}(2(\alpha-\beta)c - 1)}\sigma_{\mathrm{int}}^2,
\end{aligned}
$$

that finalizes the proof. $\qquad\square$

**Decreasing stepsize parameter.** Now we switch to the analysis of $\mathsf{SPS}_{\max}$ with decreasing stepsize parameters. We consider the stepsize $\mathsf{DecSPS}$ introduced in [64]

$$
\gamma_k = \frac{1}{c_k}\min\left\{\frac{f_{i_k}(x^k) - f_{i_k}^*}{\|\nabla f_{i_k}(x^k)\|^2}, c_{k-1}\gamma_{k-1}\right\}
$$

with $c_{-1} = c_0$ and $\gamma_{-1} = \gamma_b > 0$ to get

$$
\gamma_0 = \frac{1}{c_0}\min\left\{\frac{f_{i_0}(x^0) - f_{i_0}^*}{\|\nabla f_{i_0}(x^0)\|^2}, c_0\gamma_b\right\}.
$$

**Lemma 3** (Lemma 1 from [64]). Let Assumption 1 holds. Let $\{c_k\}_{k \geq 0}$ be any non-decreasing positive sequence of real numbers. Then we have

$$
\min\left\{\frac{1}{2c_k L}, \frac{c_0\gamma_b}{c_k}\right\} \leq \gamma_k \leq \frac{c_0\gamma_b}{c_k}, \quad \text{and} \quad \gamma_k \leq \gamma_{k-1}. \tag{28}
$$

**Theorem 5.** *Assume that Assumptions 1-2 hold. Let $\{c_k\}_{k \geq 0}$ be any positive non-decreasing sequence such that $c_k \geq \frac{1}{\alpha-\beta}$. Let Then iterates of* $\mathsf{DecSPS}$ *satisfy*

$$
\min_{0 \leq k < K}\mathbb{E}\left[f(x^k) - f^*\right] \leq \frac{2\widetilde{L}D^2 c_{K-1}}{K(\alpha-\beta)} + \frac{2\beta}{\alpha-\beta}\sigma_{\mathrm{int}}^2 + \frac{1}{K}\sum_{k=0}^{K-1}\frac{\sigma_{\mathrm{int}}^2}{(\alpha-\beta)c_k}, \tag{29}
$$

*where $D^2 := \max_{0 \leq k \leq K}\mathrm{dist}(x^k, \mathcal{S})^2$, $\widetilde{L} := \max\left\{L, \frac{1}{2c_0\gamma_b}\right\}$.*

*Proof.* From the definition of the stepsize we have

$$\gamma_k \leq \frac{1}{c_k} \frac{f_{i_k}(x^k) - f_{i_k}^*}{\|\nabla f_{i_k}(x^k)\|^2}.$$

Therefore, we have

$$\gamma_k^2 \|\nabla f_{i_k}(x^k)\|^2 \leq \frac{\gamma_k}{c_k}(f_{i_k}(x^k) - f_{i_k}^*).$$

Let $x_p^k = \mathrm{Proj}(x^k, \mathcal{S})$ which satisfies Definition 1. Now we have

$$
\begin{aligned}
\mathrm{dist}(x^{k+1}, \mathcal{S})^2 \quad &\leq \quad \|x^{k+1} - x_p^k\|^2 \\
&\leq \quad \mathrm{dist}(x^k, \mathcal{S})^2 - 2\gamma_k \left\langle \nabla f_{i_k}(x^k), x^k - x_p^k \right\rangle + \frac{\gamma_k}{c_k}(f_{i_k}(x^k) - f_{i_k}^*).
\end{aligned}
$$

Using $\alpha$-$\beta$-condition we get

$$
\begin{aligned}
\mathrm{dist}(x^{k+1}, \mathcal{S})^2 \quad &\leq \quad \|x^{k+1} - x_p^k\|^2 \\
&\leq \quad \mathrm{dist}(x^k, \mathcal{S})^2 - 2\alpha\gamma_k(f_{i_k}(x^k) - f_{i_k}(x_p^k)) + 2\beta\gamma_k(f_{i_k}(x^k) - f_{i_k}^*) \\
&\quad + \frac{\gamma_k}{c_k}(f_{i_k}(x^k) - f_{i_k}^*) \\
&= \quad \mathrm{dist}(x^k, \mathcal{S})^2 - \gamma_k\left(2\alpha - 2\beta - \frac{1}{c_k}\right)(f_{i_k}(x^k) - f_{i_k}(x_p^k)) \\
&\quad + \gamma_k(2\beta + \frac{1}{c_k})(f_{i_k}(x_p^k) - f_{i_k}^*).
\end{aligned}
$$

Let us divide both sides by $\gamma_k > 0$

$$
\begin{aligned}
\frac{\mathrm{dist}(x^{k+1}, \mathcal{S})^2}{\gamma_k} \quad &\leq \quad \frac{\mathrm{dist}(x^k, \mathcal{S})^2}{\gamma_k} - \left(2\alpha - 2\beta - \frac{1}{c_k}\right)(f_{i_k}(x^k) - f_{i_k}(x_p^k)) \\
&\quad + (2\beta + \frac{1}{c_k})(f_{i_k}(x_p^k) - f_{i_k}^*).
\end{aligned}
$$

Since by hypothesis $c_k \geq \frac{1}{\alpha - \beta}$, we get $2\alpha - 2\beta - \frac{1}{c_k} \geq \alpha - \beta$. Therefore, we get

$$f_{i_k}(x^k) - f_{i_k}(x_p^k) \quad \leq \quad \frac{\mathrm{dist}(x^k, \mathcal{S})^2}{(\alpha - \beta)\gamma_k} - \frac{\mathrm{dist}(x^{k+1}, \mathcal{S})^2}{(\alpha - \beta)\gamma_k} + \frac{(2\beta + \frac{1}{c_k})}{\alpha - \beta}(f_{i_k}(x_p^k) - f_{i_k}^*).$$

Summing from $k = 0$ to $K - 1$ we get

$$
\begin{aligned}
\sum_{k=0}^{K-1} f_{i_k}(x^k) - f_{i_k}(x_p^k) \quad &\leq \quad \sum_{k=0}^{K-1} \frac{\mathrm{dist}(x^k, \mathcal{S})^2}{(\alpha - \beta)\gamma_k} - \sum_{k=0}^{K-1} \frac{\mathrm{dist}(x^{k+1}, \mathcal{S})^2}{(\alpha - \beta)\gamma_k} \\
&\quad + \sum_{k=0}^{K-1} \frac{2\beta}{\alpha - \beta}(f_{i_k}(x_p^k) - f_{i_k}^*) + \sum_{k=0}^{K-1} \frac{1}{(\alpha - \beta)c_k}(f_{i_k}(x_p^k) - f_{i_k}^*) \\
&\leq \quad \frac{\mathrm{dist}(x^0, \mathcal{S})^2}{(\alpha - \beta)\gamma_0} + \sum_{k=1}^{K-1} \frac{\mathrm{dist}(x^k, \mathcal{S})^2}{(\alpha - \beta)\gamma_k} - \sum_{k=0}^{K-2} \frac{\mathrm{dist}(x^{k+1}, \mathcal{S})^2}{(\alpha - \beta)\gamma_k} \\
&\quad - \frac{\mathrm{dist}(x^K, \mathcal{S})^2}{(\alpha - \beta)\gamma_{K-2}} + \sum_{k=0}^{K-1} \frac{2\beta}{\alpha - \beta}(f_{i_k}(x_p^k) - f_{i_k}^*) \\
&\quad + \sum_{k=0}^{K-1} \frac{1}{(\alpha - \beta)c_k}(f_{i_k}(x_p^k) - f_{i_k}^*).
\end{aligned}
$$

Here we use the fact that $\gamma_k$ is a non-increasing sequence with $k$. We continue as follows

$$
\begin{aligned}
\sum_{k=0}^{K-1} f_{i_k}(x^k) - f_{i_k}(x_p^k) &\leq \frac{\operatorname{dist}(x^0, \mathcal{S})^2}{(\alpha - \beta)\gamma_0} + \sum_{k=0}^{K-2} \left( \frac{1}{\gamma_{k+1}} - \frac{1}{\gamma_k} \right) \frac{\operatorname{dist}(x^{k+1}, \mathcal{S})^2}{\alpha - \beta} \\
&\quad + \sum_{k=0}^{K-1} \frac{2\beta}{\alpha - \beta}(f_{i_k}(x_p^k) - f_{i_k}^*) + \sum_{k=0}^{K-1} \frac{1}{(\alpha - \beta)c_k}(f_{i_k}(x_p^k) - f_{i_k}^*) \\
&\leq \frac{D^2}{\alpha - \beta} \left( \frac{1}{\gamma_0} + \sum_{k=0}^{K-2} \left( \frac{1}{\gamma_{k+1}} - \frac{1}{\gamma_k} \right) \right) \\
&\quad + \sum_{k=0}^{K-1} \frac{2\beta}{\alpha - \beta}(f_{i_k}(x_p^k) - f_{i_k}^*) + \sum_{k=0}^{K-1} \frac{1}{(\alpha - \beta)c_k}(f_{i_k}(x_p^k) - f_{i_k}^*) \\
&\leq \frac{D^2}{(\alpha - \beta)\gamma_{K-1}} + \sum_{k=0}^{K-1} \frac{2\beta}{\alpha - \beta}(f_{i_k}(x_p^k) - f_{i_k}^*) \\
&\quad + \sum_{k=0}^{K-1} \frac{1}{(\alpha - \beta)c_k}(f_{i_k}(x_p^k) - f_{i_k}^*).
\end{aligned}
$$

Here we use the fact that $1/\gamma_{k+1} - 1/\gamma_k \geq 0$. From Lemma 3 we have

$$
\frac{1}{\gamma_k} \leq c_k \underbrace{\max \left\{ 2L, \frac{1}{c_0 \gamma_b} \right\}}_{:= \widetilde{L}}.
$$

Therefore, we continue as follows

$$
\begin{aligned}
\frac{1}{K} \sum_{k=0}^{K-1} f_{i_k}(x^k) - f_{i_k}(x_p^k) &\leq \frac{2\widetilde{L}D^2 c_{K-1}}{K(\alpha - \beta)} + \frac{1}{K} \sum_{k=0}^{K-1} \frac{2\beta}{\alpha - \beta}(f_{i_k}(x_p^k) - f_{i_k}^*) \\
&\quad + \frac{1}{K} \sum_{k=0}^{K-1} \frac{1}{(\alpha - \beta)c_k}(f_{i_k}(x_p^k) - f_{i_k}^*).
\end{aligned}
$$

Taking expectation we get

$$
\min_{0 \leq k < K} \mathbb{E}\left[ f(x^k) - f^* \right] \leq \frac{2\widetilde{L}D^2 c_{K-1}}{K(\alpha - \beta)} + \frac{2\beta}{\alpha - \beta}\sigma_{\text{int}}^2 + \frac{1}{K} \sum_{k=0}^{K-1} \frac{\sigma_{\text{int}}^2}{(\alpha - \beta)c_k}.
$$

$\square$

**Corollary 1.** Let $c_k = \sqrt{k+1}$ with $c_{-1} = c_0$, then iterates of DecSPS satisfy

$$
\min_{0 \leq k < K} \mathbb{E}\left[ f(x^k) - f^* \right] \leq \frac{2\widetilde{L}D^2 + 2\sigma_{\text{int}}^2}{\sqrt{K}(\alpha - \beta)} + \frac{2\beta}{\alpha - \beta}\sigma_{\text{int}}^2 \tag{30}
$$

$$
= \widetilde{\mathcal{O}} \left( \frac{1}{\sqrt{K}} + \beta\sigma_{\text{int}}^2 \right). \tag{31}
$$

*Proof.* The proof directly follows from Theorem 4 using the choice $c_k = \sqrt{k+1}$ and the fact that $\sum_{k=0}^{K-1} k^{-1/2} \leq 2\sqrt{K}$. $\square$

**Remark 4.** It turned out that removing the bounded iterates assumption for DecSPS is a challenging task. Nevertheless, we believe that this is rather the technicalities of Polyak stepsize, but not of our assumption. Moreover, we highlight that [64] removed bounded iterates assumption in restrictive strongly convex setting.

**Algorithm 3** NGN: Non-negative Gauss Newton
___
1: **Input:** Stepsize parameter $\gamma$
2: **for** $k = 0, 1, 2, \ldots, K - 1$ **do**
3:     Sample $i_k \sim \mathrm{Unif}[n]$ and compute $\nabla f_{i_k}(x^k)$
4:     Compute NGN stepsize

$$\gamma_k := \frac{\gamma}{1 + \frac{\gamma}{2f_{i_k}(x^k)} \|\nabla f_{i_k}(x^k)\|^2}$$

5:     Update model

$$x^{k+1} = x^k - \gamma_k \nabla f_{i_k}(x^k)$$

6: **end for**
___

### C.2.3   Convergence of NGN

**Constant stepsize parameter.**   In this section we present the proof of convergence of NGN with constant stepsize parameter $\gamma$ under $\alpha$-$\beta$-condition for completeness of the presentation.

**Lemma 4** (Lemma from [63])**.** Let $f_i$ be $L$-smooth for all $i$, then the stepsize of NGN satisfies

$$\gamma_k \in \left[ \frac{\gamma}{1 + \gamma L}, \gamma \right]. \tag{32}$$

**Lemma 5** (Lemma from [63])**.** Let $f_i$ be $L$-smooth for all $i$, then the iterates of NGN satisfy

$$\gamma_k^2 \|\nabla f_{i_k}(x^k)\|^2 \leq \frac{4\gamma L}{1 + 2\gamma L} \gamma_k (f_{i_k}(x^k) - f_{i_k}^*) + \frac{2\gamma^2 L}{1 + \gamma L} \max\left\{ \frac{2\gamma L - 1}{2\gamma L + 1}, 0 \right\} f_{i_k}^*. \tag{33}$$

**Theorem 3.** *Assume that Assumptions 1 with $\alpha \geq \beta + 1$ and 1-2 hold. Assume that each function $f_i$ is positive and $\sigma_{\mathrm{pos}}^2 < \infty$. Then the iterates of NGN (Alg. 3) with a stepsize parameter $\gamma > 0$ satisfy*

$$
\begin{aligned}
\min_{0 \leq k \leq K-1} \mathbb{E}\left[ f(x^k) - f^* \right] \quad \leq \quad & \frac{\mathbb{E}\left[ \mathrm{dist}(x^0, \mathcal{S})^2 \right]}{2\gamma K} \frac{(1 + 2\gamma L)^2}{c_2} + \frac{3L\gamma\alpha(1 + \gamma L)\sigma_{\mathrm{int}}^2}{c_2} \\
& + \frac{\gamma L}{a} \max\left\{ 2\gamma L - 1, 0 \right\} \sigma_{\mathrm{pos}}^2 + \frac{2\beta\sigma_{\mathrm{int}}^2}{c_2},
\end{aligned}
\tag{11}
$$

*where $c_2 := 2\gamma L(\alpha - \beta - 1) + \alpha - \beta$.*

*Proof.* Let $x_p^k \in \mathrm{Proj}(x^k, \mathcal{S})$ satisfying Definition 1. Then we have

$$
\begin{aligned}
\mathbb{E}_k\left[ \mathrm{dist}(x^{k+1}, \mathcal{S})^2 \right] \quad \leq \quad & \mathbb{E}_k\left[ \|x^{k+1} - x_p^k\|^2 \right] \\
= \quad & \|x^k - x_p^k\|^2 - 2\mathbb{E}_k\left[ \gamma_k \langle \nabla f_{i_k}(x^k), x^k - x_p^k \rangle \right] + \mathbb{E}_k\left[ \gamma_k^2 \|\nabla f_{i_k}(x^k)\|^2 \right] \\
\leq \quad & \mathrm{dist}(x^k, \mathcal{S})^2 - 2\alpha\mathbb{E}_k\left[ \gamma_k (f_{i_k}(x^k) - f_{i_k}(x_p^k)) \right] \\
& + 2\beta\mathbb{E}_k\left[ \gamma_k (f_{i_k}(x^k) - f_{i_k}^*) \right] + \mathbb{E}_k\left[ \gamma_k^2 \|\nabla f_{i_k}(x^k)\|^2 \right].
\end{aligned}
\tag{34}
$$

From Lemma 5

$$\gamma_k^2 \|\nabla f_{i_k}(x^k)\|^2 \leq \frac{4\gamma L}{1 + 2\gamma L} \gamma_k (f_{i_k}(x^k) - f_{i_k}^*) + \frac{2\gamma^2 L}{1 + \gamma L} \max\left\{ \frac{2\gamma L - 1}{2\gamma L + 1}, 0 \right\} f_{i_k}^*. \tag{35}$$

Plugging (35) in (34) we get

$$
\begin{aligned}
\mathbb{E}_k\left[ \mathrm{dist}(x^{k+1}, \mathcal{S})^2 \right] \quad \leq \quad & \mathrm{dist}(x^k, \mathcal{S})^2 - 2\alpha\mathbb{E}_k\left[ \gamma_k (f_{i_k}(x^k) - f_{i_k}(x_p^k)) \right] \\
& + 2\beta\mathbb{E}_k\left[ \gamma_k (f_{i_k}(x^k) - f_{i_k}^*) \right] \frac{4\gamma L}{1 + 2\gamma L} + \mathbb{E}_k\left[ \gamma_k (f_{i_k}(x^k) - f_{i_k}^*) \right] \\
& + \frac{2\gamma^2 L}{1 + \gamma L} \max\left\{ \frac{2\gamma L - 1}{2\gamma L + 1}, 0 \right\} \mathbb{E}_k\left[ f_{i_k}^* \right].
\end{aligned}
$$

We have $f_{i_k}(x^k) - f_{i_k}(x_p^k) = (f_{i_k}(x^k) - f_{i_k}^*) - (f_{i_k}(x_p^k) - f_{i_k}^*)$. Now we write $\gamma_k = \rho + \epsilon_k$ where $\rho$ is a constant independent of $i_k$. Therefore, we have

$$
\begin{aligned}
\mathbb{E}_k\left[\text{dist}(x^{k+1}, \mathcal{S})^2\right] \leq\ & \text{dist}(x^k, \mathcal{S})^2 - 2\alpha\rho\mathbb{E}_k\left[f_{i_k}(x^k) - f_{i_k}(x_p^k)\right] \\
& - 2\alpha\mathbb{E}_k\left[\epsilon_k(f_{i_k}(x^k) - f_{i_k}^*)\right] + 2\alpha\mathbb{E}_k\left[\epsilon_k(f_{i_k}(x_p^k) - f_{i_k}^*)\right] \\
& + 2\beta\mathbb{E}_k\left[\gamma_k(f_{i_k}(x^k) - f_{i_k}^*)\right] + \frac{4\gamma L}{1 + 2\gamma L}\mathbb{E}_k\left[\gamma_k(f_{i_k}(x^k) - f_{i_k}^*)\right] \\
& + \frac{2\gamma^2 L}{1 + \gamma L}\max\left\{\frac{2\gamma L - 1}{2\gamma L + 1}, 0\right\}\mathbb{E}_k\left[f_{i_k}^*\right] \\
=\ & \text{dist}(x^k, \mathcal{S})^2 - 2\alpha\rho\mathbb{E}_k\left[f_{i_k}(x^k) - f_{i_k}(x_p^k)\right] \\
& - 2\mathbb{E}_k\left[\left(\alpha\epsilon_k - \left(\beta + \frac{2\gamma L}{1 + 2\gamma L}\right)\gamma_k\right)(f_{i_k}(x^k) - f_{i_k}^*)\right] \\
& + 2\alpha\mathbb{E}_k\left[\epsilon_k(f_{i_k}(x_p^k) - f_{i_k}^*)\right] \\
& + \frac{2\gamma^2 L}{1 + \gamma L}\max\left\{\frac{2\gamma L - 1}{2\gamma L + 1}, 0\right\}\mathbb{E}_k\left[f_{i_k}^*\right]. \quad (36)
\end{aligned}
$$

We need to find $\rho$ such that

$$
\begin{aligned}
\alpha\epsilon_k - \left(\beta + \frac{2\gamma L}{1 + 2\gamma L}\right)\gamma_k &\geq 0 \\
\alpha(\gamma_k - \rho) - \left(\beta + \frac{2\gamma L}{1 + 2\gamma L}\right)\gamma_k &\geq 0 \\
\gamma_k\left(\alpha - \beta - \frac{2\gamma L}{1 + 2\gamma L}\right) &\geq \alpha\rho.
\end{aligned}
$$

Note that since $\gamma_k \geq \frac{\gamma}{1+\gamma L}$ from Lemma 4, then it is enough if $\rho$ satisfies

$$
\begin{aligned}
\frac{\gamma}{1 + \gamma L}\left(\alpha - \beta - \frac{2\gamma L}{1 + 2\gamma L}\right) &\geq \alpha\rho \\
\frac{\gamma(2\gamma L(\alpha - \beta - 1) + \alpha - \beta)}{\alpha(1 + \gamma L)(1 + 2\gamma L)} &\geq \rho.
\end{aligned}
$$

Let us take this bound as a value of $\rho$. Note that since $\alpha \geq \beta + 1$, then $\rho \geq 0$. Therefore, the bound for $\epsilon_k$ is the following

$$
\begin{aligned}
\epsilon_k =\ & \gamma_k - \rho \\
\leq\ & \gamma - \frac{\gamma(2\gamma L(\alpha - \beta - 1) + \alpha - \beta)}{\alpha(1 + \gamma L)(1 + 2\gamma L)} \\
=\ & \gamma\left(\frac{\alpha(1 + \gamma L)(1 + 2\gamma L) - 2\gamma L(\alpha - \beta - 1) - (\alpha - \beta)}{\alpha(1 + \gamma L)(1 + 2\gamma L)}\right) \\
=\ & L\gamma^2\frac{\alpha + 2\alpha\gamma L + 2(\beta + 1)}{\alpha(1 + \gamma L)(1 + 2\gamma L)} + \gamma\frac{\beta}{\alpha(1 + \gamma L)(1 + 2\gamma L)} \\
\leq\ & \underbrace{\frac{3L\gamma^2}{1 + 2\gamma L}}_{:=\frac{1}{2}T_2(\gamma^2)} + \underbrace{\frac{\beta\gamma}{\alpha(1 + \gamma L)(1 + 2\gamma L)}}_{:=\frac{1}{2}\beta T_1(\gamma)}.
\end{aligned}
$$

Therefore, we get the following descent inequality

$$
\begin{aligned}
\mathbb{E}_k \left[ \mathrm{dist}(x^{k+1}, \mathcal{S})^2 \right] \;\leq\;\; & \mathrm{dist}(x^k, \mathcal{S})^2 - \underbrace{2\alpha\rho}_{:=T_0(\gamma)} \mathbb{E}_k \left[ f_{i_k}(x^k) - f_{i_k}(x_p^k) \right] \\
& + 2\alpha \mathbb{E}_k \left[ \epsilon_k (f_{i_k}(x_p^k) - f_{i_k}^*) \right] + \underbrace{\frac{2\gamma^2 L}{1 + \gamma L} \max\left\{ \frac{2\gamma L - 1}{2\gamma L + 1}, 0 \right\} \mathbb{E}_k \left[ f_{i_k}^* \right]}_{:=T_3(\gamma^2)} \\
\leq\;\; & \mathrm{dist}(x^k, \mathcal{S})^2 - T_0(\gamma) \mathbb{E}_k \left[ f_{i_k}(x^k) - f_{i_k}(x_p^k) \right] \\
& + T_2(\gamma^2)\alpha \mathbb{E}_k \left[ f_{i_k}(x_p^k) - f_{i_k}^* \right] + \beta T_1(\gamma)\alpha \mathbb{E}_k \left[ f_{i_k}(x_p^k) - f_{i_k}^* \right] \\
& + T_3(\gamma^2) \mathbb{E}_k \left[ f_{i_k}^* \right] \\
=\;\; & \mathrm{dist}(x^k, \mathcal{S})^2 - T_0(\gamma)(f(x^k) - f^*) + T_2(\gamma^2)\alpha \mathbb{E}_k \left[ f^* - f_{i_k}^* \right] \\
& + \beta T_1(\gamma)\alpha \mathbb{E}_k \left[ f^* - f_{i_k}^* \right] + T_3(\gamma^2) \mathbb{E}_k \left[ f_{i_k}^* \right]. \tag{37}
\end{aligned}
$$

Here we use the fact that $x_p^k$ is a minimizer of $f$ and independent of $i_k$. Unrolling the recursion, we get

$$
\frac{1}{K} \left( \mathbb{E} \left[ \mathrm{dist}(x^K, \mathcal{S})^2 \right] - \mathbb{E} \left[ \mathrm{dist}(x^0, \mathcal{S})^2 \right] \right) \;\leq\; -\frac{T_0(\gamma)}{K} \sum_{k=0}^{K-1} \mathbb{E} \left[ f(x_k) - f^* \right]
$$
$$
+ E_1(\gamma) + E_2(\gamma^2), \tag{38}
$$

where

$$
E_1(\gamma) := \beta T_1(\gamma)\alpha \underbrace{\mathbb{E} \left[ f^* - f_i^* \right]}_{\sigma_{\mathrm{int}}^2}, \quad E_2(\gamma^2) := T_2(\gamma^2)\alpha \underbrace{\mathbb{E} \left[ f^* - f_i^* \right]}_{\sigma_{\mathrm{int}}^2} + T_3(\gamma^2) \underbrace{\mathbb{E} \left[ f_i^* \right]}_{\sigma_{\mathrm{pos}}^2}. \tag{39}
$$

Therefore, we get

$$
\begin{aligned}
\min_{0 \leq k \leq K-1} \mathbb{E} \left[ f(x^k) - f^* \right] \;\leq\;\; & \frac{\mathbb{E} \left[ \mathrm{dist}(x^0, \mathcal{S})^2 \right]}{K T_0(\gamma)} + \frac{E_1(\gamma)}{T_0(\gamma)} + \frac{E_2(\gamma^2)}{T_0(\gamma)} \\
\leq\;\; & \frac{\mathbb{E} \left[ \mathrm{dist}(x^0, \mathcal{S})^2 \right]}{2\gamma K} \frac{(1 + 2\gamma L)^2}{2\gamma L(\alpha - \beta - 1) + \alpha - \beta} \\
& + \frac{3L\gamma\alpha}{2\gamma L(\alpha - \beta - 1) + \alpha - \beta}(1 + \gamma L)\sigma_{\mathrm{int}}^2 \\
& + \frac{\gamma L}{2\gamma L(\alpha - \beta - 1) + \alpha - \beta} \max\{2\gamma L - 1, 0\}\, \sigma_{\mathrm{pos}}^2 \\
& + \frac{\beta\sigma_{\mathrm{int}}^2}{2\gamma L(\alpha - \beta - 1) + \alpha - \beta}. \tag{40}
\end{aligned}
$$

This finished the proof. $\qquad\square$

**Remark 5.** If all $f_i$ are convex, i.e. we can take $\alpha = 1, \beta = 0$ in $\alpha$-$\beta$-condition, then we get

$$
\begin{aligned}
\min_{0 \leq k \leq K-1} \mathbb{E} \left[ f(x^k) - f^* \right] \;\leq\;\; & \frac{\mathbb{E} \left[ \mathrm{dist}(x^0, \mathcal{S})^2 \right]}{2\gamma K}(1 + 2\gamma L)^2 + 3L\gamma(1 + \gamma L)\sigma_{\mathrm{int}}^2 \\
& + \gamma L \max\{2\gamma L - 1, 0\}\, \sigma_{\mathrm{pos}}^2,
\end{aligned}
$$

that coincides with the results in [63].

**Decreasing stepsize parameter.** Now we present the results with decreasing stepsize parameter.

**Theorem 6.** *Assume that Assumptions 1 with $\alpha \geq \beta + 1$ and 1-2 hold. Assume that each function $f_i$ is positive and $\sigma_{\mathrm{pos}}^2 < \infty$. Then the iterates of* NGN *with decreasing stepsize of the form*

$$
\gamma_k = \frac{\widetilde{\gamma}_k}{1 + \frac{\widetilde{\gamma}_k}{2 f_{i_k}(x^k)} \|\nabla f_{i_k}(x^k)\|^2}, \quad \widetilde{\gamma}_k := \frac{\gamma}{\sqrt{k+1}}
$$

*satisfy*

$$\min_{0\leq k<K} \mathbb{E}\left[f(x^k)-f^*\right] \leq \frac{C_1}{\sqrt{K}}\mathbb{E}\left[\text{dist}(x^0,\mathcal{S})^2\right] + \frac{C_2\log(K+1)}{\sqrt{K}}\alpha\sigma_{\text{int}}^2 + \frac{C_3}{\alpha-\beta}\beta\sigma_{\text{int}}^2$$
$$+ \frac{C_4\log(K+1)}{\sqrt{K}}\sigma_{\text{pos}}^2 \tag{41}$$
$$= \widetilde{\mathcal{O}}\left(\frac{1}{\sqrt{K}}+\beta\sigma_{\text{int}}^2\right). \tag{42}$$

*where $C_1, C_2, C_3,$ and $C_4$ are defined in (43).*

*Proof.* Similarly to the proof with constant stepsize parameter we can obtain (similar to (37))

$$\mathbb{E}_k\left[\text{dist}(x^{k+1},\mathcal{S})^2\right] \leq \text{dist}(x^k,\mathcal{S})^2 - T_0(\widetilde{\gamma}_k)(f(x^k)-f^*) + T_2(\widetilde{\gamma}_k^2)\alpha\sigma_{\text{int}}^2$$
$$+ \beta T_1(\widetilde{\gamma}_k)\alpha\sigma_{\text{int}}^2 + T_3(\widetilde{\gamma}_k^2)\sigma_{\text{pos}}^2.$$

Note that we have

$$T_0(\widetilde{\gamma}_k) = \frac{2\widetilde{\gamma}_k(2\widetilde{\gamma}_k L(\alpha-\beta-1)+\alpha-\beta)}{(1+\widetilde{\gamma}_k L)(1+2\widetilde{\gamma}_k L)}$$
$$\geq \frac{2\widetilde{\gamma}_k(\alpha-\beta)}{(1+\widetilde{\gamma}_0 L)(1+2\widetilde{\gamma}_0 L)} =: \widetilde{T}_0(\widetilde{\gamma}_k),$$
$$T_1(\widetilde{\gamma}_k) = \frac{2\widetilde{\gamma}_k}{\alpha(1+\widetilde{\gamma}_k L)(1+2\widetilde{\gamma}_k L)}$$
$$\leq \frac{2\widetilde{\gamma}_k}{\alpha} =: \widetilde{T}_1(\widetilde{\gamma}_k),$$
$$T_2(\widetilde{\gamma}_k^2) = \frac{6L\widetilde{\gamma}_k^2}{1+2\widetilde{\gamma}_k L}$$
$$\leq 6L\widetilde{\gamma}_k^2 =: \widetilde{T}_2(\widetilde{\gamma}_k^2),$$
$$T_3(\widetilde{\gamma}_k^2) = \frac{2\widetilde{\gamma}_k^2 L}{1+\widetilde{\gamma}_k L}\max\left\{\frac{2\widetilde{\gamma}_k L-1}{2\widetilde{\gamma}_k L+1},0\right\}$$
$$\leq 2\widetilde{\gamma}_k^2 L\max\{2\gamma L-1,0\} =: \widetilde{T}_3(\widetilde{\gamma}_k^2).$$

Therefore, we can continue as follows

$$\mathbb{E}\left[\text{dist}(x^{k+1},\mathcal{S})^2\right] \leq \mathbb{E}\left[\text{dist}(x^k,\mathcal{S})^2\right] - \widetilde{T}_0(\widetilde{\gamma}_k)\mathbb{E}\left[f(x^k)-f^*\right] + \widetilde{T}_2(\widetilde{\gamma}_k^2)\alpha\sigma_{\text{int}}^2$$
$$+ \beta\widetilde{T}_1(\widetilde{\gamma}_k)\alpha\sigma_{\text{int}}^2 + \widetilde{T}_3(\widetilde{\gamma}_k^2)\sigma_{\text{pos}}^2.$$

This leads to

$$\sum_{k=0}^{K-1}\widetilde{T}_0(\widetilde{\gamma}_k)\mathbb{E}\left[f(x^k)-f^*\right] \leq \mathbb{E}\left[\text{dist}(x^0,\mathcal{S})^2\right] + \sum_{k=0}^{K-1}\widetilde{T}_2(\widetilde{\gamma}_k^2)\alpha\sigma_{\text{int}}^2 + \sum_{k=0}^{K-1}\beta\widetilde{T}_1(\widetilde{\gamma}_k)\alpha\sigma_{\text{int}}^2$$
$$+ \sum_{k=0}^{K-1}\widetilde{T}_3(\widetilde{\gamma}_k^2)\sigma_{\text{pos}}^2.$$

Therefore, we have

$$\min_{0\leq k<K}\mathbb{E}\left[f(x^k)-f^*\right] \leq \frac{1}{\sum_{k=0}^{K-1}\widetilde{T}_0(\widetilde{\gamma}_k)}\mathbb{E}\left[\text{dist}(x^0,\mathcal{S})^2\right] + \frac{\sum_{k=0}^{K-1}\widetilde{T}_2(\widetilde{\gamma}_k^2)}{\sum_{k=0}^{K-1}\widetilde{T}_0(\widetilde{\gamma}_k)}\alpha\sigma_{\text{int}}^2$$
$$+ \frac{\beta\sum_{k=0}^{K-1}\widetilde{T}_1(\widetilde{\gamma}_k)}{\sum_{k=0}^{K-1}\widetilde{T}_0(\widetilde{\gamma}_k)}\alpha\sigma_{\text{int}}^2 + \frac{\sum_{k=0}^{K-1}\widetilde{T}_3(\widetilde{\gamma}_k^2)}{\sum_{k=0}^{K-1}\widetilde{T}_0(\widetilde{\gamma}_k)}\sigma_{\text{pos}}^2.$$

Following the results of [64] and [24] we get

$$
\begin{aligned}
\sum_{k=0}^{K-1} \widetilde{T}_0(\widetilde{\gamma}_k) &= \sum_{k=0}^{K-1} \frac{2\widetilde{\gamma}_k}{(1+\gamma L)(1+2\gamma L)} \\
&\geq \frac{8\gamma\sqrt{K}}{5(1+\gamma L)(1+2\gamma L)}, \\
\sum_{k=0}^{K-1} \widetilde{T}_2(\widetilde{\gamma}_k^2) &= \sum_{k=0}^{K-1} 6L\widetilde{\gamma}_k^2 \\
&\leq 12L\gamma^2 \log(K+1), \\
\sum_{k=0}^{K-1} \widetilde{T}_3(\widetilde{\gamma}_k^2) &= \sum_{k=0}^{K-1} 2\widetilde{\gamma}_k^2 L \max\{2\gamma L - 1, 0\} \\
&\leq 4\gamma^2 L \max\{2\gamma L - 1, 0\} \log(K+1), \\
\frac{\sum_{k=0}^{K-1} \widetilde{T}_1(\widetilde{\gamma}_k)}{\sum_{k=0}^{K-1} \widetilde{T}_0(\widetilde{\gamma}_k)} &= \frac{\sum_{k=0}^{K-1} \frac{2\widetilde{\gamma}_k}{\alpha}}{\sum_{k=0}^{K-1} \frac{2\widetilde{\gamma}_k(\alpha-\beta)}{(1+\gamma L)(1+2\gamma L)}} \\
&= \frac{(1+\gamma L)(1+2\gamma L)}{\alpha(\alpha-\beta)}.
\end{aligned}
$$

Thus, the final result can be written as follows

$$
\begin{aligned}
\min_{0 \leq k < K} \mathbb{E}\left[f(x^k) - f^*\right] &\leq \frac{5(1+\gamma L)(1+2\gamma L)}{8\gamma\sqrt{K}} \mathbb{E}\left[\operatorname{dist}(x^0, \mathcal{S})^2\right] + \frac{12L\gamma^2 \log(K+1)}{\frac{8\gamma\sqrt{K}}{5(1+\gamma L)(1+2\gamma L)}} \alpha\sigma_{\text{int}}^2 \\
&\quad + \frac{\sum_{k=0}^{K-1} \widetilde{T}_1(\widetilde{\gamma}_k)}{\sum_{k=0}^{K-1} \widetilde{T}_0(\widetilde{\gamma}_k)} \alpha\sigma_{\text{int}}^2 + \frac{\sum_{k=0}^{K-1} \widetilde{T}_3(\widetilde{\gamma}_k^2)}{\sum_{k=0}^{K-1} \widetilde{T}_0(\widetilde{\gamma}_k)} \sigma_{\text{pos}}^2 \\
&= \frac{5(1+\gamma L)(1+2\gamma L)}{8\gamma\sqrt{K}} \mathbb{E}\left[\operatorname{dist}(x^0, \mathcal{S})^2\right] \\
&\quad + \frac{15L\gamma(1+\gamma L)(1+2\gamma L) \log(K+1)}{2\sqrt{K}} \alpha\sigma_{\text{int}}^2 \\
&\quad + \frac{(1+\gamma L)(1+2\gamma L)}{(\alpha-\beta)} \beta\sigma_{\text{int}}^2 \\
&\quad + \frac{5(1+\gamma L)(1+2\gamma L)\gamma L \max\{2\gamma L - 1, 0\} \log(K+1)}{2\sqrt{K}} \sigma_{\text{pos}}^2.
\end{aligned}
$$

Now it is left to use the definitions of constants $C_1, C_2,$ and $C_3$

$$
\begin{aligned}
C_1 &:= \frac{5(1+\gamma L)(1+2\gamma L)}{8\gamma}, \\
C_2 &:= \frac{15L\gamma(1+\gamma L)(1+2\gamma L)}{2}, \\
C_3 &:= (1+\gamma L)(1+2\gamma L), \\
C_4 &:= \frac{5(1+\gamma L)(1+2\gamma L)\gamma L \max\{2\gamma L - 1, 0\}}{2}.
\end{aligned}
\tag{43}
$$

$\square$

### C.2.4 Convergence of AdaGrad-norm-max

**Theorem 7.** *Assume that Assumptions 1-2 hold. Assume that for all $k \geq 0$ stochastic gradients satisfy $\|\nabla f_{i_k}(x^k)\|^2 \leq G^2$ for some $G > 0$ and $b_{-1} \geq \frac{2L\gamma}{\alpha-\beta}$. Let $D^2 = \max_i \|\nabla f_i(x^0)\|^2$. Then the*

---

**Algorithm 4** AdaGrad-norm-max

---

1: **Input:** Stepsize parameter $\gamma > 0, c_{-1} = 0, b_{-1} \geq \frac{2L\gamma}{\alpha-\beta}$
2: **for** $k = 0, 1, 2, \ldots, K - 1$ **do**
3:     Sample $i_k \sim \text{Unif}[n]$ and compute $\nabla f_{i_k}(x^k)$
4:     Compute AdaGrad-norm-max stepsize

$$
\begin{aligned}
c_k^2 &:= \max \left\{ c_{k-1}^2, \|\nabla f_{i_k}(x^k)\|^2 \right\} \\
b_k^2 &:= b_{k-1}^2 + c_k^2, \quad \gamma_k = \frac{\gamma}{b_k}
\end{aligned}
$$

5:     Update model

$$
x^{k+1} = x^k - \gamma_k \nabla f_{i_k}(x^k)
$$

6: **end for**

---

*iterates of* AdaGrad-norm-max *(Alg. 4) satisfy*

$$
\begin{aligned}
\min_{0 \leq k < K} \mathbb{E}\left[ f(x^k) - f^* \right] &\leq \frac{\text{dist}(x^k, \mathcal{S})^2}{\gamma K(\alpha - \beta)} \sqrt{b_{-1}^2 + G^2 K} \\
&\quad + \frac{2\alpha}{(\alpha - \beta)KD^2} \sigma_{\text{int}}^2 \sqrt{b_{-1}^2 + G^2 K} \sqrt{b_{-1}^2 + D^2(K+1)} \\
&= \mathcal{O}\left( \frac{1}{\sqrt{K}} + \alpha\sigma_{\text{int}}^2 \right).
\end{aligned}
\tag{44}
$$

We see that if $K$ is large enough, then we recover the standard convex rate of Adagrad of order $\widetilde{\mathcal{O}}(K^{-1/2})$ [46].

*Proof.* Let $x_p^k = \text{Proj}(x^k, \mathcal{S})$ satysfying Definition 1. Then we have

$$
\begin{aligned}
\text{dist}(x^{k+1}, \mathcal{S})^2 &\leq \|x^{k+1} - x_p^k\|^2 \\
&= \|x^k - x_p^k\|^2 - 2\gamma_k \langle \nabla f_{i_k}(x^k), x^k - x_p^k \rangle + \gamma_k^2 \|\nabla f_{i_k}(x^k)\|^2 \\
&\leq \text{dist}(x^k, \mathcal{S})^2 - 2\alpha\gamma_k(f_{i_k}(x^k) - f_{i_k}(x_p^k)) + 2\beta\gamma_k(f_{i_k}(x^k) - f_{i_k}^*) \\
&\quad + \gamma_k^2 \|\nabla f_{i_k}(x^k)\|^2 \\
&\leq \text{dist}(x^k, \mathcal{S})^2 - 2\alpha\gamma_k(f_{i_k}(x^k) - f_{i_k}(x_p^k)) + 2\beta\gamma_k(f_{i_k}(x^k) - f_{i_k}^*) \\
&\quad + 2\gamma_k^2 L(f_{i_k}(x^k) - f_{i_k}^*) \\
&= \text{dist}(x^k, \mathcal{S})^2 - 2\alpha\gamma_k(f_{i_k}(x^k) - f_{i_k}^* + f_{i_k}^* - f_{i_k}(x_p^k)) + 2\beta\gamma_k(f_{i_k}(x^k) - f_{i_k}^*) \\
&\quad + 2\gamma_k^2 L(f_{i_k}(x^k) - f_{i_k}^*) \\
&= \text{dist}(x^k, \mathcal{S})^2 - 2\gamma_k(\alpha - \beta - L\gamma_k)(f_{i_k}(x^k) - f_{i_k}^*) \\
&\quad + 2\alpha\gamma_k(f_{i_k}(x_p^k) - f_{i_k}^*).
\end{aligned}
\tag{45}
$$

Note that we choose $b_{-1} \geq \frac{2L\gamma}{\alpha-\beta}$. Since $b_k$ is inreasing sequence, then we have for any $k$ that $b_k \geq \frac{2L\gamma}{\alpha-\beta}$. Therefore,

$$
L\gamma_k = L\frac{\gamma}{b_k} \leq L\frac{\gamma}{b_{-1}} \leq L\frac{\gamma}{2L\gamma/\alpha-\beta} = \frac{\alpha - \beta}{2}.
$$

This means that

$$
-2\gamma_k(\alpha - \beta - L\gamma_k) \leq -2\gamma_k(\alpha - \beta - \alpha-\beta/2) = -\gamma_k(\alpha - \beta).
$$

Thus, we can continue (45) as follows

$$
\text{dist}(x^{k+1}, \mathcal{S})^2 \leq \text{dist}(x^k, \mathcal{S})^2 - \frac{\gamma}{b_k}(\alpha - \beta)(f_{i_k}(x^k) - f_{i_k}^*) + 2\alpha\frac{\gamma}{b_k}(f_{i_k}(x_p^k) - f_{i_k}^*).
$$

We know that $b_k$ is increasing sequence that satisfy

$$\sqrt{b_{-1}^2 + D^2(k+1)} \leq b_k \leq b_{K-1} \leq \sqrt{b_{-1}^2 + G^2 K}.$$

This leads together with the fact that both $f_{i_k}(x^k) - f_{i_k}^*$ and $f_{i_k}(x_p^k) - f_{i_k}^*$ are non-negative to

$$
\begin{aligned}
\text{dist}(x^{k+1}, \mathcal{S})^2 &\leq \text{dist}(x^k, \mathcal{S})^2 - \frac{\gamma}{\sqrt{b_{-1}^2 + G^2 K}}(\alpha - \beta)(f_{i_k}(x^k) - f_{i_k}^*) \\
&\quad + 2\alpha \frac{\gamma}{\sqrt{b_{-1}^2 + D^2(k+1)}}(f_{i_k}(x_p^k) - f_{i_k}^*) \\
&= \text{dist}(x^k, \mathcal{S})^2 - \frac{\gamma(\alpha - \beta)}{\sqrt{b_{-1}^2 + G^2 K}}(f_{i_k}(x^k) - f_{i_k}(x_p^k)) \\
&\quad + \frac{2\alpha\gamma}{\sqrt{b_{-1}^2 + D^2(k+1)}}(f_{i_k}(x_p^k) - f_{i_k}^*).
\end{aligned}
$$

Taking the conditional expectation we get

$$
\frac{\mathbb{E}_k\left[f(x^k) - f^*\right]}{\sqrt{b_{-1}^2 + G^2 K}} \leq \frac{\text{dist}(x^k, \mathcal{S})^2}{\gamma(\alpha - \beta)} - \frac{\mathbb{E}_k\left[\text{dist}(x^{k+1}, \mathcal{S})^2\right]}{\gamma(\alpha - \beta)} + \frac{2\alpha}{\sqrt{b_{-1}^2 + D^2(k+1)}(\alpha - \beta)}\sigma_{\text{int}}^2,
$$

which leads to the following bound

$$
\begin{aligned}
\mathbb{E}_k\left[f(x^k) - f^*\right] &\leq \left(\frac{\text{dist}(x^k, \mathcal{S})^2}{\gamma(\alpha - \beta)} - \frac{\mathbb{E}_k\left[\text{dist}(x^{k+1}, \mathcal{S})^2\right]}{\gamma(\alpha - \beta)}\right)\sqrt{b_{-1}^2 + G^2 K} \\
&\quad + \frac{2\alpha}{\sqrt{b_{-1}^2 + D^2(k+1)}(\alpha - \beta)}\sigma_{\text{int}}^2\sqrt{b_{-1}^2 + G^2 K}.
\end{aligned}
$$

Averaging over $K$ iterations we get

$$
\begin{aligned}
\min_{0 \leq k < K} \mathbb{E}\left[f(x^k) - f^*\right] &\leq \frac{1}{K}\sum_{k=0}^{K-1}\mathbb{E}\left[f(x^k) - f^*\right] \\
&\leq \frac{\text{dist}(x^k, \mathcal{S})^2}{\gamma K(\alpha - \beta)}\sqrt{b_{-1}^2 + G^2 K} \\
&\quad + \frac{2\alpha}{(\alpha - \beta)K}\sigma_{\text{int}}^2\sqrt{b_{-1}^2 + G^2 K}\sum_{k=0}^{K-1}\frac{1}{\sqrt{b_{-1}^2 + D^2(k+1)}} \\
&\leq \frac{\text{dist}(x^k, \mathcal{S})^2}{\gamma K(\alpha - \beta)}\sqrt{b_{-1}^2 + G^2 K} \\
&\quad + \frac{2\alpha}{(\alpha - \beta)KD^2}\sigma_{\text{int}}^2\sqrt{b_{-1}^2 + G^2 K}\sqrt{b_{-1}^2 + D^2(K+1)} \\
&= \mathcal{O}\left(\frac{1}{\sqrt{K}} + \alpha\sigma_{\text{int}}^2\right).
\end{aligned}
$$

$\square$

# D   Additional experiments

For all the experiments, we make use of PyTorch [65] package. LSTM, MLP, CNN and Resnet experiments are performed using one NVIDIA GeForce RTX 3090 GPU with a memory of 24 GB. For training Algoperf and Pythia language models, we resort instead to 4xA100-SXM4 GPUs, with a memory of 40 GB each, and employ data parallelism for efficient distributed training. When necessary, we disable CUDA non-deterministic operations, to allow consistency between runs with the same random seed.

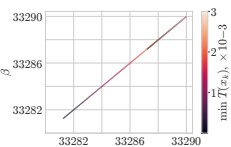 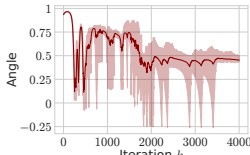 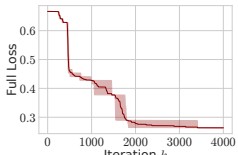 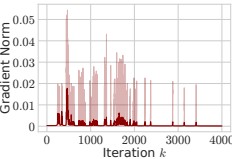

Figure 9: Training for half-space learning problem with SGD. Here $T(x_k) = \langle \nabla f_{i_k}(x^k), x^k - x^K \rangle - \alpha(f_{i_k}(x^k) - f_{i_k}(x^K)) - \beta f_{i_k}(x^k)$ assuming that $f_i^* \approx 0.000523853$; angle denotes $\angle(\nabla f(x^k), x^k - x^K)$.

## D.1 Half space learning

Half Space Learning problem corresponds to the following optimization problem

$$f(x) = \frac{1}{n} \sum_{i=1}^{n} \sigma(-b_i x^\top a_i) + \frac{\lambda}{2} \|x\|^2,$$

where $\{a_i, b_i\}_{i=1}^{n}, a_i \in \mathbb{R}^d, y_i \in \{0, 1\}$ is a given dataset, $\lambda = 10^{-5}$, and $\sigma$ is a sigmoid function. For the test, we create a synthetic dataset that contains 20 samples for both classes. We sample data points from normal distribution where each class has its own mean and variance value of 2. We use SGD with learning rate $\frac{1}{4}$ and batch size 1 for minimization task.[5]

We observe that the gradient norm becomes zero quite which means that SGD trajectory goes through saddle point. This leads to possible negative angle between full gradient and direction to minimizer. Nevertheless, we demonstrate that even for such highly non-convex landscape with many saddle points our $\alpha$-$\beta$-condition holds for large enough values of $\alpha$ and $\beta$.

## D.2 Experiment setup from section 2.2

We use 3 layer LSTM based model from Hübler et al. [33][6]. The model for PTB dataset has 35441600 parameters while for Wikitext-2 dataset the model has 44798534 parameters. To train the model, we choose NSGD with momentum [15] with decaying stepsize and momentum parameters according to the experiment section from [33]. We train the model for 1000 epochs with initial stepsize values 158 and 900 for PTB and Wikitext-2 datasets respectively. We switch off dropout during measuring stochastic gradients and losses for $\alpha$-$\beta$-condition . We run the experiments for 7 different random seeds and plot the mean with maximum and minimum fluctuations.

In Figure 10, we plot empirical aiming coefficient $\frac{\langle \nabla f(x^k), x^k - x^K \rangle}{f(x^k)}$ for full loss $f$; mean of stochastic losses across 7 runs along with maximum and minimum fluctuations from the mean; pairs of $(\alpha, \beta)$ that satisfy $\alpha$-$\beta$-condition across all 7 runs. We observe that for both datasets, there is a plateau at the beginning of the training. This part of the training corresponds to possible negative values of the empirical coefficient Aiming condition. After this, it becomes stable and positive.

Besides, we demonstrate that possible values of pairs of $(\alpha, \beta)$ that satisfy $\alpha$-$\beta$-condition are large. We believe, this happens because the full loss $f$ has a minimum value of around 3.5 while individual losses have $f_i^*$, i.e. the model is far from the interpolation regime (when $f^* = f_i^*$).

## D.3 MLP architecture

We use MLP model with 3 fully connected layers. We fix the dimensions of the second layer to be equal, i.e. the parameter matrix of this layer is square. After the first and second fully connected layers we use ReLU activation function. We train the model in all cases with fixed learning rate 0.09 for 1500 epochs and batch size 64 on Fashion-MNIST [83] dataset. In Figure 12, we plot the

---

[5]We use the implementation from [17] that can be found https://github.com/archana95in/Escaping-saddles-using-Stochastic-Gradient-Descent.

[6]The implementation can be found following the link https://github.com/fhueb/parameter-agnostic-lzlo

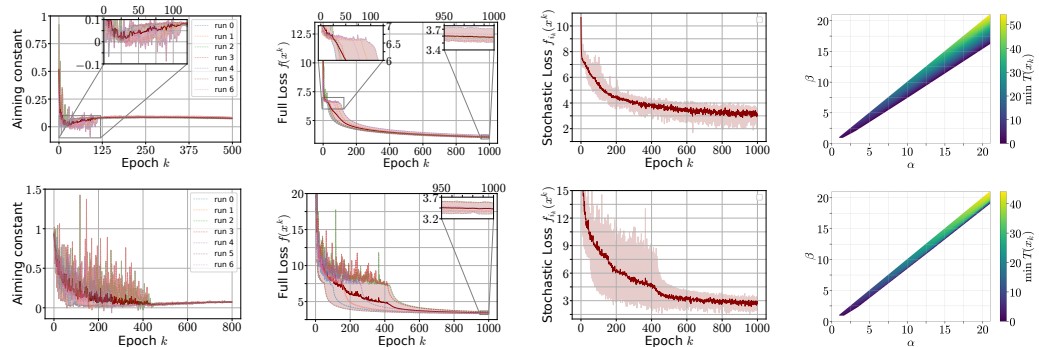

Figure 10: Training of 3 layer LSTM model on PTB (first row) and Wikitext-2 (second row) datasets. Here $T(x_k) = \langle \nabla f_{i_k}(x^k), x^k - x^K \rangle - \alpha(f_{i_k}(x^k) - f_{i_k}(x^K)) - \beta f_{i_k}(x^k)$ assuming that $f_i^* = 0$.

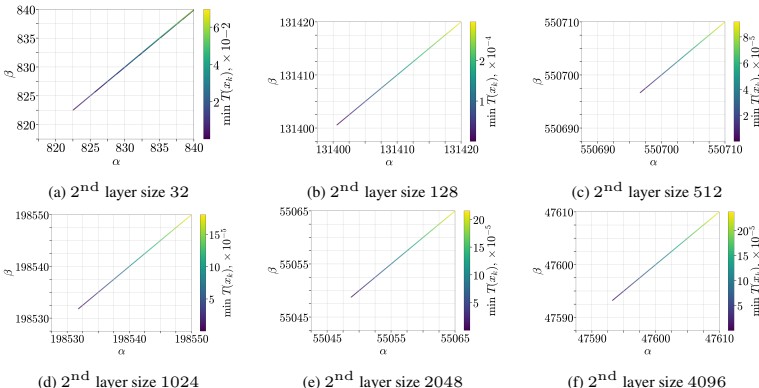

Figure 11: Values of $\alpha$ and $\beta$ during the training of 3 layer MLP model on Fashion-MNIST dataset varying the dimension of the second layer. Here $T(x_k) = \langle \nabla f_{i_k}(x^k), x^k - x^K \rangle - \alpha(f_{i_k}(x^k) - f_{i_k}(x^K)) - \beta f_{i_k}(x^k)$ assuming that $f_i^* = 0$.

mean stochastic loss across 4 runs along with maximum and minimum fluctuations, and in Figure 11 possible values of $\alpha$ and $\beta$ that work over all random seeds and iterations satisfying $\alpha \geq \beta + 0.1$.

We observe that the magnitude of the smallest possible values of $\alpha$ and $\beta$ increase till up to the second layer size 512, but then it starts decreasing as the model becomes more over-parameterized. This leads to smaller values of neighborhood $\mathcal{O}(\beta \sigma_{\text{int}}^2)$ as we expect in this setting.

## D.4 CNN architecture

We use CNN model with 2 convolution layers followed by a fully connected one. After each convolution layer, we use max-pooling and ReLU activation functions. We train the model with a cosine annealing learning rate scheduler with a maximum value $0.01$ and batch size $64$. We train the model on CIFAR10 dataset [41] for $1500$ epochs. We run the experiments for $4$ different random seeds. In Figure 13 we plot possible values of $\alpha$ and $\beta$ satisfying $\alpha \geq \beta + 0.1$ that work across all runs and iterations.

We observe that increasing the dimension of the second layer of the model makes the model closer to over-parameterization: values of stochastic losses decrease. We observe the same phenomenon as in Appendix D.3: minimum possible values of $\alpha$ and $\beta$ increase up to 128 number of convolutions, but then it decreases for a larger number of convolutions. This happens because the model becomes more over-parameterized. Moreover, the possible difference between $\alpha$ and $\beta$ tends to increase with number of convolutions starting from 128 convolutions.

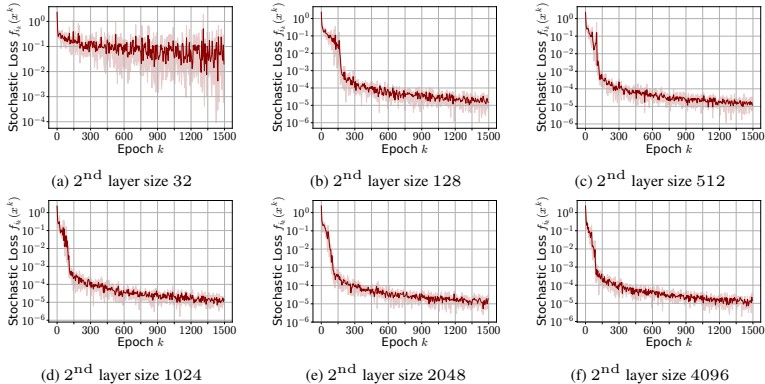

Figure 12: Values of stochastic loss during the training of 3 layer MLP model on Fashion-MNIST dataset varying the dimension of the second layer.

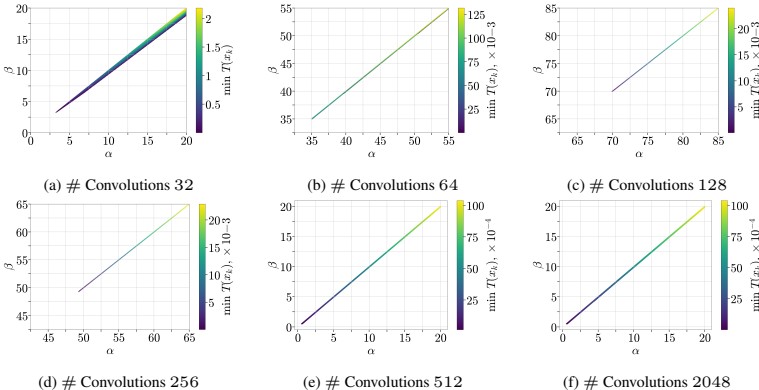

Figure 13: Values of $\alpha$ and $\beta$ during the training of CNN model on CIFAR10 dataset varying the number of convolutions in the second layer. Here $T(x_k) = \langle \nabla f_{i_k}(x^k), x^k - x^K \rangle - \alpha(f_{i_k}(x^k) - f_{i_k}(x^K)) - \beta f_{i_k}(x^k)$ assuming that $f_i^* = 0$.

## D.5 Resnet architecture

We use the implementation from Kumar [42]. We train the model on CIFAR100 dataset [41] for 1000 epochs. We use one-cycle scheduler with a maximum learning rate $0.01$. To compute dense stochastic gradients, we switch off dropout during evaluations of stochastic gradients and losses for $\alpha$-$\beta$-condition. We run the experiments for $4$ different random seeds and plot the mean along with maximum and minimum fluctuations.

In Figure 15 we observe that the minimum value of stochastic loss increases with batch size which means that the model becomes further from over-parameterization.

## D.6 Verification of $\alpha$-$\beta$-condition by different optimizers

Now we turn to another interesting question: how does the choice of an optimizer affect the practical verification of the $\alpha$-$\beta$-condition? To explore this question, we train Resnet9 model with SGD, SGDM, and Adam. We report the results in Figure 16 varying the batch size used in the training. Comparing the values of $\alpha$ and $\beta$ for SGD (from Figure 6), SGDM, and Adam, we observe that the loss landscape explored by the Adam optimizer achieves smaller values of $\alpha$ and $\beta$. Moreover, the values of $\alpha$ and $\beta$ found by SGDM are typically smaller than those found by SGD. This result may shed light on why momentum (from a comparison of SGDM against SGD) and adaptive stepsize (from a comparison of SGDM against Adam) are typically beneficial in practice: these more advanced algorithms explore better part of a loss landscape from the $\alpha$-$\beta$-condition point of view.

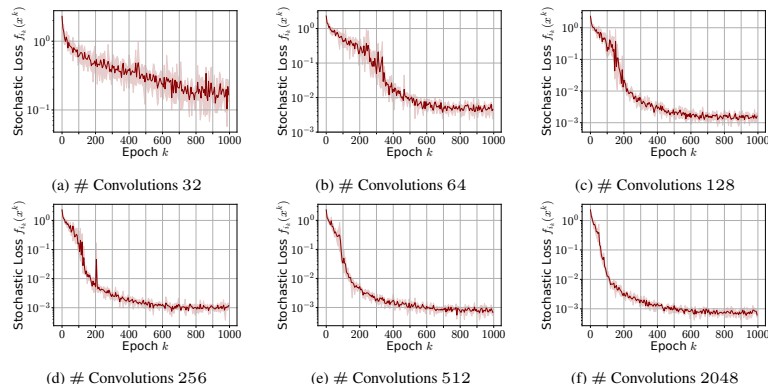

Figure 14: Values of stochastic loss during the training of CNN model on CIFAR10 dataset varying the number of convolutions in the second layer.

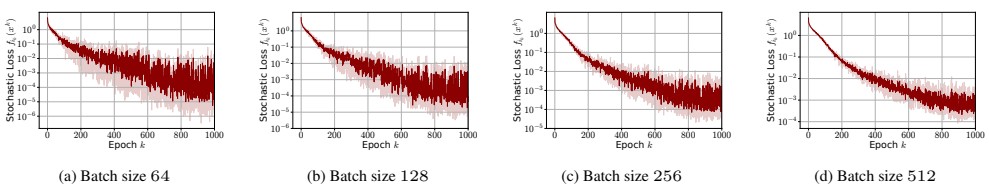

Figure 15: Training of Resnet9 model on CIFAR100 dataset varying the batch size.

Table 3: Training details of large models from Appendix D.8 and Appendix D.9

| Dataset | Model | Batch Size | LR | $\beta_1$ | $\beta_2$ | Weight Decay | Warmup |
|---------|-------|-----------|-----|-----------|-----------|--------------|--------|
| Criteo 1TB | DLRMsmall | 262144 | 0.0017 | 0.93 | 0.995 | 0.08 | 0.02 |
| Fastmri | U-Net | 32 | 0.001 | 0.9 | 0.998 | 0.15 | 0.1 |
| OBGB | GNN | 512 | 0.0017 | 0.93 | 0.995 | 0.08 | 0.02 |
| WMT | Transformer | 128 | 0.001 | 0.97 | 0.999 | 0.15 | 0.1 |
| Slim-Pajama-627B | Pythia-70M | 256 | 0.01 | 0.9 | 0.95 | 0.1 | 0.1 |
| Slim-Pajama-627B | Pythia-160M | 256 | 0.006 | 0.9 | 0.95 | 0.1 | 0.1 |

## D.7 Increasing the depth of Resnet architecture

In our next experiment, we aim to investigate how the $\alpha$-$\beta$-condition behaves increasing the depth of a model. To do so, in addition to training Resnet9 model, we test Resnet18 and Resnet34 models on CIFAR100 dataset. The results in Figure 17 suggest that the values of $\alpha$ and $\beta$ tend to decrease with the depth of a model. These observations align with those from Section 5.1 and Section 5.2 as the constants $\alpha$ and $\beta$ decrease as the depth of the model increases.

## D.8 AlgoPerf experiments

For each of all aforementioned tasks, we repeat the training with 3 random seeds to create more stable results. The detailed model architectures are given in [16]. In Table 3 we provide the parameters of optimizers we use for each task. The loss curves are presented in Figure 18. For each workload, we run the experiments for 3 different random seeds to obtain more stable results. The hyperparameters of optimizer NadamW are chosen such that we can reach the validation threshold set by the organizers[7]. We employ a cosine annealing learning rate schedule that reduces the learning to $1e-10$, with an initial linear warm-up. For each workload, we run the experiments sufficiently enough so that we reach the validation target threshold and the stochastic loss becomes sufficiently stable.

---

[7]The quality of performance is measured differently from one task to another; we defer to the [16] for a more detailed description of the competition.

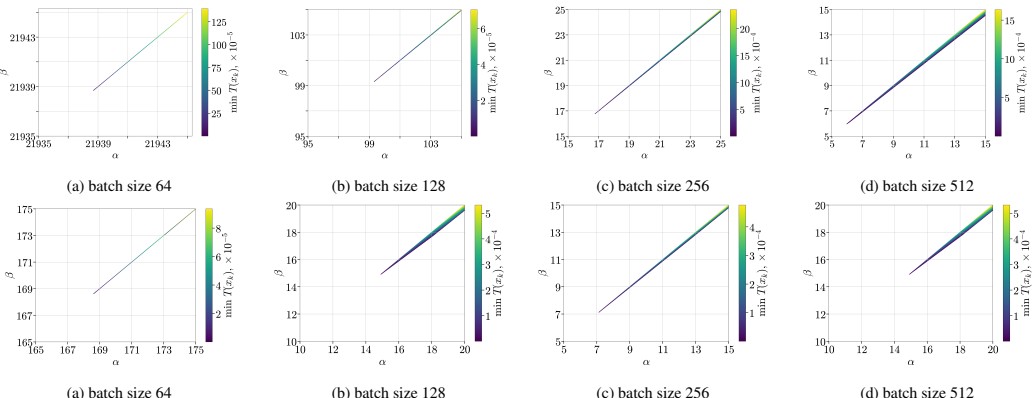

Figure 16: Training of ResNet9 model on CIFAR100 dataset varying the batch size with SGDM (stepsize 0.01 and momentum 0.9 with OneCycle learning rate scheduler) and Adam (stepsize 0.0001 with default momentum parameters with OneCycle learning rate scheduler) optimizers. Here $T(x_k) = \langle \nabla f_{i_k}(x^k), x^k - x^K \rangle - \alpha(f_{i_k}(x^k) - f_{i_k}(x^K)) - \beta f_{i_k}(x^k)$ assuming that $f_i^* = 0$. Minimum is taken across all runs and iterations for a given pair of $(\alpha, \beta)$. We plot values of $\alpha, \beta$ in $\alpha$-$\beta$-condition for SGDM (**first row**) and Adam (**second row**).

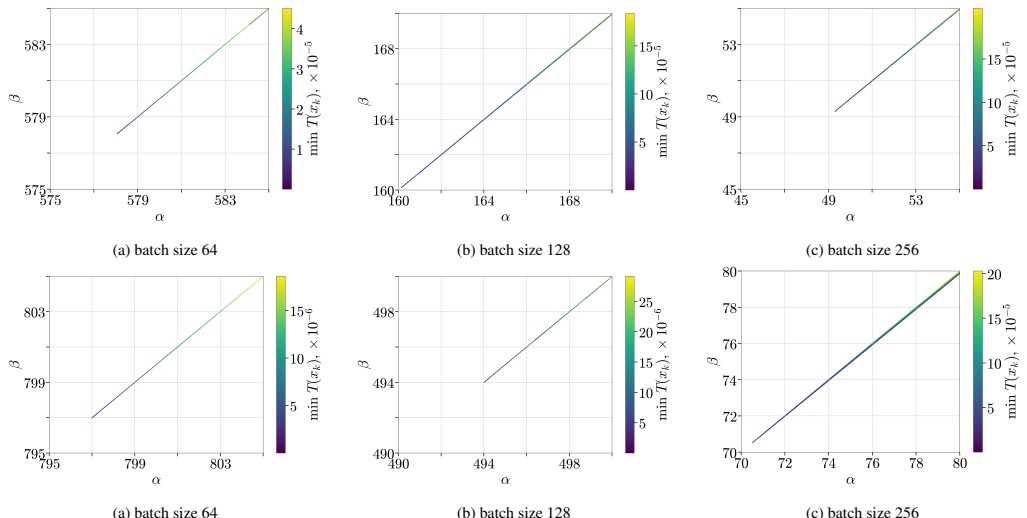

Figure 17: Training of ResNet18 and Resnet34 model on CIFAR100 dataset varying the batch size SGD (stepsize 0.01 and momentum 0.9 with OneCycle learning rate scheduler). Here $T(x_k) = \langle \nabla f_{i_k}(x^k), x^k - x^K \rangle - \alpha(f_{i_k}(x^k) - f_{i_k}(x^K)) - \beta f_{i_k}(x^k)$ assuming that $f_i^* = 0$. Minimum is taken across all runs and iterations for a given pair of $(\alpha, \beta)$.

### D.9    Pythia experiments

For each of all aforementioned tasks, we repeat the training with 3 random seeds to create more stable results. We train Pythia 70M and Pythia 160M [8] on publicly available Slim-Pajama-627B dataset [72]. Both models are trained on sequences of length 2048, and makes use of a batch size of 0.5M tokens, which amounts to a batch size of 256 samples. We use AdamW optimizer and a cosine annealing with linear warmup, with hyperparameters specified in Table 3. The stochastic loss and training perplexity are reported in Figure 19.

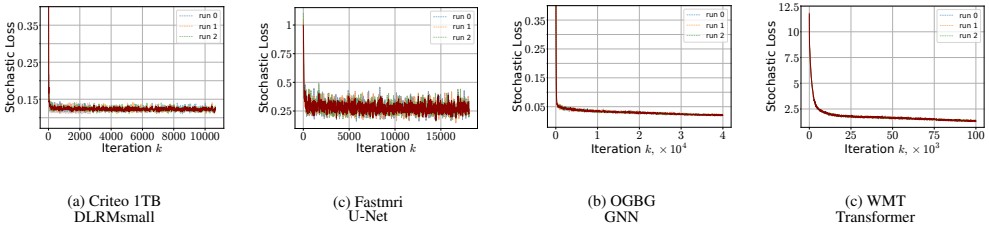

Figure 18: Training of large models from AlgoPerf benchmark.

(a) Criteo 1TB DLRMsmall
(c) Fastmri U-Net
(b) OGBG GNN
(c) WMT Transformer

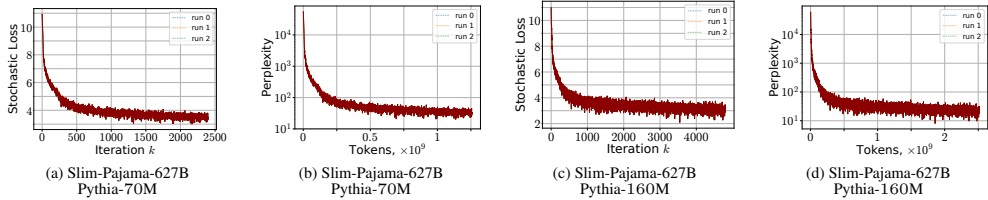

(a) Slim-Pajama-627B Pythia-70M
(b) Slim-Pajama-627B Pythia-70M
(c) Slim-Pajama-627B Pythia-160M
(d) Slim-Pajama-627B Pythia-160M

Figure 19: Training statistics for Pythia language models.

