# OpenReview forum: "Loss Landscape Characterization of Neural Networks without  Over-Parametrization"
_NeurIPS.cc/2024/Conference — NeurIPS 2024 poster_

### Official Review · Reviewer_MYcC · 2024-06-13

**Soundness:** 3
**Presentation:** 4
**Contribution:** 4
**Rating:** 7
**Confidence:** 4

**Summary:**

This paper proposes a new condition to describe the optimization landscape of deep neural networks. This condition alleviates restrictive consequences of alternative conditions such as PL, in particular the overparameterisation and absence of saddle points. A convergence for SGD (and other variants of first-order stochastic methods) is proven for the class of functions that satisfy this condition. The paper showcases examples, both in simple cases and in deep learning, where other conditions do not hold while theirs does. Numerical experiments show that the condition holds in practice.

**Strengths:**

The paper is very clear and well-written. It tackles the crucial question of understanding which structure in the loss landscape of deep neural networks enables efficient optimization with first-order methods. The arguments in favour of the proposed condition are fairly compelling, with both theoretical results and experimental insights. I verified the proof of one of the main results (Theorem 1), which is correct.

**Weaknesses:**

The convergence for SGD is shown up to a non-vanishing error term $O(\beta \sigma^2)$. While the authors give some explanation on this term, I would have liked more insights: is the error term appearing for SPS and NGN the same (this is not directly obvious from the formulas)? Do authors think this term is an artefact of the proof technique, or intimately connected to the $\alpha-\beta$ condition? The authors argue that overparameterization leads to vanishing of this term. As far as I understand, both $\beta$ and $\sigma^2$ are affected by overparameterization, and both terms should decay with overparameterization. Is this correct? Does this give an insight of the "best" level of overparameterization?

**Questions:**

Questions:

See Weaknesses for main questions.

Other questions:
- “However, we highlight that the convergence guarantees of the optimizers depend on a term $O(\beta \sigma^2)$ which is stable across all experiments we provide.” What does ”stable” mean here?
- Remark 3 line 687: what is the connection between r and R small, and the fact that X is situated locally around S? If S is far away from 0, then r and R would be large even around S?
- Line 696: several nabla signs missing. I also believe there is a factor 2 missing from the use of the smoothness condition, see for instance equation 2.1.10 in [1].
- Several existing function class (Polyak--Łojasiewicz inequality, Quadratic Growth and Error Bound) that are mentioned in the overview of Section 2.1 are actually equivalent, see [2]. This would benefit from being mentioned.
- Line 84-85: there does exist works showing a PL inequality with only linear overparameterization in width, see [3].

Minor remarks:
- The acronyms SPS and NGN are not explained in the main text, and are not that well-known. Spelling out the acronyms once would be beneficial.
- $\gamma_b$ in Theorem 2 is not defined (in the main text).

[1] Nesterov, Lectures on Convex Optimization, Second Edition, 2018.

[2] Rebjock, Boumal, Fast convergence to non-isolated minima: four equivalent conditions for C2 functions, arXiv:2303.00096

[3] Marion, Wu, Sander, Biau, Implicit regularization of deep residual networks towards neural ODEs, ICLR 2024.

**Limitations:**

The limitations are adequately addressed.

---

> ### Author Rebuttal · Authors · 2024-08-06
>
> **A to W (part 1):** For NGN the first three terms that appear in our Theorem 3 also appear in the convex setting; see Theorem 4.5 in [1]. The fourth (and last) term appears because of the $\alpha$-$\beta$-condition. We highlight that the first three terms shrink with a decreasing stepsize, but the last term persists, and it is proportional to $\beta\sigma^2_{\mathrm{int}}$ similarly to SGD. However, if we consider a convex regime, i.e. $\alpha=1, \beta=0$ in the $\alpha$-$\beta$-condition, then the last term disappears. For SPS, the error term is different. First, it is proportional to $\alpha\sigma^2_{\mathrm{int}}$, i.e. even in the convex regime it remains in the bound  (this result aligns with convergence guarantees in [2]). However, in the interpolation regime, for both NGN and SPS, the error terms disappear. In conclusion, the error terms for NGN and SPS are different in the general case ($\alpha,\beta >0$), but both disappear in the interpolation regime.
>
> [1] Orvieto, Antonio and Xiao, Lin, An Adaptive Stochastic Gradient Method with Non-negative Gauss-Newton Stepsizes, arXiv preprint arXiv:2407.04358, 2024.
>
> [2] Loizou, Nicolas and Vaswani, Sharan and Laradji, Issam Hadj and Lacoste-Julien, Simon, Stochastic polyak step-size for sgd: An adaptive learning rate for fast convergence, AISTATS 2021.
>
> **A to W (part 2):** We refer to the general answer. Regarding the "best" level of over-parameterization, could the reviewer please clarify his questions? From Theorem 1, the best rate of SGD is given when a model is over-parametrized, and then the second and third terms in the rate disappear; only the classic optimization term decaying as $\mathcal{O}(1/K)$ remains in this setting.
>
> **A to Q1:** Thank you for this comment. We can very roughly estimate the value of $\beta\sigma^2_{\mathrm{int}}$ based on experiments. The value of $\sigma^2_{\mathrm{int}}$ can be approximated as the value of the stochastic loss at the end of the training (since we use $x^K \approx x_p$, then $f_i(x^K) \approx f_i(x_p)$). Therefore, assuming that $f_i^*=0$ we get that $\sigma^2_{\mathrm{int}} \approx f_i(x^K).$ Using such approximations, we compute the value $\beta\sigma^2_{\mathrm{int}}$ and show following the experimental setup of the paper that $\beta\sigma^2_{\mathrm{int}}$ remains stable and decreases as a model becomes closer to over-parameterization. See the table in the separate pdf file for concrete values.
>
> **A to Q2:** We need $\mathcal{S}$ to be inside $\mathcal{X}$, so that the projection operation $\mathrm{proj}_{\mathcal{S}}$ in the definition of $\alpha$-$\beta$-condition is correctly defined for any $(v,W)\in\mathcal{X}.$  Note that if $r$ and $R$ are small, then $\mathcal{S}$ cannot be arbitrarily far away from $0$ as it lies fully inside $\mathcal{X}$. If $R$ and $r$ are small enough, then the revised value of $\alpha$ we take is
> $$\max\left\\{\frac{2\max\_{i\in[n]} \log(1+\exp(Rr\\|x_i\\|))}{c-f_i^*}, 1\right\\} = 1.$$
> This implies that $\beta=\alpha-1=0.$ We provide Remark 3 to show that in most of the cases, we might need $\beta > 0,$ which implies that quasi-convexity (i.e., $\beta=0$) does not hold in this case. However, we highlight that this is only our intuition, and a more careful study is needed in this case.
>
> **A to Q3:** We thank the reviewer for pointing to the typos. We added the missing nabla signs in the proof of the convergence of SGD. Regarding the missing factor 2, there was a typo in the first equality (it should be $\gamma^2\mathbb{E}_k[\\|\nabla f\_{i_k}(x^k)\\|^2]$ instead of  $2\gamma^2\mathbb{E}_k[\\|\nabla f\_{i_k}(x^k)\\|^2]$). Therefore, the final result does not change.
>
> **A to Q4:** We thank the reviewer for providing a reference [2]. This is indeed an interesting result that PL, EB, and QG are equivalent in the neighborhood of the $\mathcal{S}$ if $f$ is sufficiently differentiable. We will add this reference to highlight that the discussion on the limitations of PL condition is also transferable to EB and QG.
>
> **A to Q5:** We thank the reviewer for providing a reference [3]. After checking the paper, we would like to highlight that their Definition 1 is stricter than the standard PL condition. They ask for the PL condition to hold in the bounded set around some fixed set of parameters. The radius of the bound is vanishing with the number of data points $n$ as $M = \mathcal{O}(1/\sqrt{n})$. We believe this could be one of the reasons to make over-parameterization to be linear in $n$ in their work. However, we will add a citation of [3] to the main body to highlight the possible improvement of necessary over-parameterization with additional restrictions (i.e., in a bounded set around some point).
>
> **A to Q6:** We thank the reviewer for this suggestion. We will spell out the acronyms in the list of contributions where we mention algorithms for the first time.
>
> **A to Q7:** $\gamma_b$ is a stepsize upper bound of SPS${}\_{\max}$ algorithm $\gamma_k = \min\left\\{\frac{f\_{i_k}(x^k) - f\_{i_k}^*}{c\\|\nabla f\_{i_k}(x^k)\\|^2}, \gamma\_{\rm b}\right\\}$. We will add the definition in the statement of SPS convergence theorem.

---

> > ### Comment · Reviewer_MYcC · 2024-08-07
> > **Thank you for your rebuttal.**
> >
> > I thank the authors for their detailed rebuttal. Regarding the "best" level of over-parameterization, the authors adequately answered my question (the term "best" referred to the fact that I thought that there was some sort of tradeoff). I keep my score.

---

### Official Review · Reviewer_qMQA · 2024-07-04

**Soundness:** 4
**Presentation:** 4
**Contribution:** 4
**Rating:** 8
**Confidence:** 4

**Summary:**

A major challenge in Deep Learning optimization has been identifying structural conditions on the loss objective that ensure convergence of SGD and variants. As in practice despite the non-convexity, stochastic gradient algorithms have been tremendously successful in training neural networks. Many conditions have been proposed that purport to explain this phenomena, including the PL-condition and Aiming. Unfortunately, such conditions may fail to hold unless the network is extremely over-parametrized, which does not align with practice. Moreover, no comprehensive empirical study has been performed verifying these conditions hold on networks of realistic sizes.

 In this paper, the authors address these issues by performing an extensive empirical analysis, showing, in general, that many prior conditions fail to hold for realistic networks. They also provide theoretical counter-examples where these conditions fail to hold. This shows convergence analyses of SGD-like algorithms performed under these assumptions may not hold in practice. To rectify this unfortunate situation, the authors introduce a new condition, which they refer to as the $\alpha-\beta$ condition. They show under this assumption that SGD and variants like SPS converge to a ball of noise under this condition. Perhaps most importantly, the paper provides extensive empirical evidence and several theoretical examples showing the $\alpha-\beta$ condition holds.

**Strengths:**

The paper has several major strengths that make it a clear accept in my view:

1. Showing the PL-condition and Aiming do not necessarily hold for networks of realistic size encountered in practice.

To the best of my knowledge, such an extensive empirical investigation has not been done before. This is of prime importance in my view, as much of the recent DL optimization literature takes these conditions for granted despite lacking a strong empirical foundation. I've long harbored doubts about how relevant the analysis in the over-parametrized regime (where said conditions hold with high probability) is to practice. Especially considering analyses over the past couple years showing networks that are well-approximated by their first-order Taylor expansion do not perform as well as networks not in this regime. So, I think the extensive empirics done in this paper showing these conditions don't hold for networks commonly used in practice is a valuable contribution to the literature

2. Introduction of the $\alpha-\beta$ condition and convergence analysis.

The authors have gone beyond just identifying failure modes of previous conditions. They introduce a new condition, the $\alpha-\beta$ condition, under which optimizers like SGD (and several variants) converge to a ball-of-noise (or the minimum in the interpolation setting). Moreover, the condition allows for more realistic loss landscapes, i.e., saddle-points can occur.

3. Empirical verification of $\alpha-\beta$ condition.

This is one of the most significant contributions of the paper. The authors provide strong empirical evidence that $\alpha-\beta$ holds for networks of practical interest.

4. The paper is very well-written. I found the paper easily to follow and enjoyable to read.

**Weaknesses:**

Overall, the paper has no major weaknesses, but it can improve on a few minor points.

1.) For the MLP and ResNet experiments I would have liked to have seen experiments with SGD+momentum and Adam (or AdamW), as these tend to be the most popular optimizers employed in practice. Given that you observed $\alpha-\beta$ holds for NAdamW for language tasks, I'd be surprised if you found the condition doesn't hold in the MLP and ResNet settings for optimizers other than SGD.

2.) Continuing off the preceding point, it would be insightful if, besides verifying $\alpha-\beta$ for several sets of optimizers, the authors also plotted the corresponding loss curves alongside and reported the estimated $\alpha, \beta$ values for each optimizer. This would give insight into how values of $\alpha, \beta$ differ across optimizers, and could help explain why an optimizer converges faster than another, as it may exhibit more favorable values of $\alpha, \beta$. Though admittedly, this requires some extrapolation as the current theory doesn't apply to a method like Adam, so it is unclear how the values of $\alpha$ and $\beta$ would affect its convergence. But it could lead to interesting questions that provide more avenues for future research.

3.) The authors provide no idealized setting, i.e. wide-networks, where $\alpha-\beta$ holds. It would be interesting to see if the level of overparametrization required to ensure $\alpha-\beta$ is weaker than what is needed to ensure the Aiming condition. Intuitively I'd expect the answer to be yes, given that it is a weaker condition than Aiming and the strong empirical evidence provided in the paper. Nevertheless, it seems non-trivial to verify and somewhat tangential to the goals of the paper, so its understandable the authors chose to leave this as a direction for future work.

**Questions:**

My main suggestion/request is that the authors address the first point under the Weaknesses sections. If the authors promise to include additional experiments along this line, I'm willing to raise my score. The results themselves can appear in the supplement, but there should be a clear reference to them in the main paper. In my view, this would significantly increase the value of the paper, as it would show that $\alpha-\beta$ empirically holds for three of the most popular optimizers in DL: SGD, SGD+Polyak momentum, and Adam. Even if the condition doesn't hold for one of them, say, Adam, it's still valuable as it raises interesting questions for future research.

**Limitations:**

I think the authors have adequately addressed limitations of the work for the most part. But I think two more items should be added:

1) The assumption of globally Lipschitz continuous gradients. It is known empirically that models like transformers do not satisfy this property, see Zhang et al. (2019). Thus, the analysis here may not hold in such settings. It is known empirically, however, that transformers satisfy a generalized smoothness condition of the form [Zhang et al. (2019)]:

     $\||H(x)\|| \leq L_0+L_1\||\nabla f(x)\||$.

The authors should state their assumption as a limitation and point to extending the convergence analysis under the $\alpha-\beta$ condition to the generalized smoothness setting as an interesting direction for future work.

2) The analysis does not cover methods like Adam, which is arguably the most popular optimization algorithm in deep learning. Moreover, much progress has been made in recent years on analysis of its performance; see, for instance Li et al. (2024). So, this should be stated as a limitation and an interesting direction for future work.

References:
1.) Zhang, Jingzhao, Tianxing He, Suvrit Sra, and Ali Jadbabaie. "Why Gradient Clipping Accelerates Training: A Theoretical Justification for Adaptivity." In International Conference on Learning Representations. 2019.

2.) Li, Haochuan, Alexander Rakhlin, and Ali Jadbabaie. "Convergence of adam under relaxed assumptions." In Advances in Neural Information Processing Systems 36 (2024).

---

> ### Author Rebuttal · Authors · 2024-08-06
>
> We sincerely thank the reviewer for their thorough reviews, insightful comments, and valuable
> questions regarding our paper.
>
> **A to W1:** The main reason why we used SGD for MLP, CNN, and ResNet experiments is the fact that SGD is known to be able to train those models, i.e. the last iterate of $x^K$ is a good approximation of $x^*$ [1]. In contrast, for larger networks, especially for language modeling tasks, SGD is not able to train a model well, see e.g. [2]. Therefore, we use NAdamW to obtain a better last iterate $x^K$ as an approximation of $x^*.$
>
> [1] Choi, Dami and Shallue, Christopher J and Nado, Zachary and Lee, Jaehoon and Maddison, Chris J and Dahl, George E, On empirical comparisons of optimizers for deep learning, arXiv preprint arXiv:1910.05446, 2019
>
> [2] Noci, Lorenzo, et al. "Signal propagation in transformers: Theoretical perspectives and the role of rank collapse." Advances in Neural Information Processing Systems 35 (2022): 27198-27211.
>
>
> **A to W2:** Following the experimental setup that the reviewer suggested, we present the experiments on ResNet9 model on CIFAR100 dataset trained with three different optimizers in a separate pdf file. We use SGD (stepsize $0.01$ with OneCycle learning rate scheduler), SGDM (stepsize $0.01$, momentum $0.9$ with OneCycle learning rate scheduler), and Adam (stepsize $0.0001$, default momentum and epsilon parameters with OneCycle learning rate scheduler). For each optimizer, we run experiments with $3$ random seeds to obtain more stable results. We observe that Adam converges slower than SGD and SGDM. This is because the learning rate for Adam is chosen to be slightly smaller so that the convergence behaviour in the end of the training is stable (with learning rates $0.001$ or higher Adam converges faster than SGD and SGDM, but the fluctuations at the end of the training are too high. This most likely happens because, at the end of the training, the Adam stepsize involves a division by a small number as the gradient becomes close to zero. Therefore, we decided to choose a smaller stepsize which still gives the same stochastic loss at the end but with smaller fluctuations). First, we observe that momentum decreases the $\alpha$, $\beta$ constants in the proposed condition which suggests that the trajectory of SGD with momentum is better under the $\alpha$-$\beta$-condition. Next, we see that Adam optimizer achieves much better $\alpha$ and $\beta$ constants than both SGD and SGDM. The provided experimental results open new interesting questions as practically both stepsize adaptivity in Adam and momentum improve convergence under the $\alpha$-$\beta$-condition.
>
> **A to W3:** We agree with the reviewer that understanding how over-parametrization affects the $\alpha$-$\beta$-condition is an important direction to explore in follow-up works, although as you pointed out, it is a complex and non-trivial question. We will acknowledge this as a limitation of our work that requires further investigation. Thank you for highlighting this point.
>
> **A to Q:** We observe that the condition still holds using all three mentioned algorithms, which we agree is an interesting addition to the paper. We present the discussion in **A to Q2** together with the plots in a separate file.
>
> **A to L1:** We thank the reviewer for these comments. Relaxed smoothness is indeed an interesting approach for future work. We will add a discussion on this to the limitations at the end of the paper.
>
> **A to L2:** We agree with the reviewer that Adam is a widely used optimizer in practice, and including convergence guarantees for Adam would be a nice addition to the paper. However, we emphasize that deriving formal convergence guarantees for Adam requires an involved analysis, which seemed out of scope for the paper where we already derived convergence guarantees for three optimizers. Nonetheless, we concur with the reviewer that this topic should be noted for future research endeavors.

---

> > ### Comment · Reviewer_qMQA · 2024-08-07
> > **Thank you for your rebuttal**
> >
> > I would like to sincerely thank the authors for their detailed reply. In particular, I appreciate the authors adding the experiments I've requested. I think they will enhance the quality of the submission. The response has addressed all my concerns.
> > As I said in my initial review, I think this paper is interesting and provides a strong contribution. Therefore, I am happy to raise my score from 7 to 8.

---

### Official Review · Reviewer_W37Q · 2024-07-12

**Soundness:** 2
**Presentation:** 3
**Contribution:** 2
**Rating:** 4
**Confidence:** 4

**Summary:**

This paper introduces a new regulatory condition named $\alpha$-$\beta$ condition. To motivate the necessity of such condition, it first show empirically that the aiming condition is not always satisfied in neural newtork training. The paper then support the $\alpha$-$\beta$ condition with $i).$ examples (including a shallow neural network) that satisfies such condition; $ii).$ convergence guarantee of SGD, SPS, and NGN under such condition combined with smoothness; and $iii).$ Empirical evidence that the $\alpha$-$\beta$ condition holds during neural network training.

**Strengths:**

1. The newly introduced $\alpha$-$\beta$ condition covers the case where the function has saddle points.
2. The paper tries to give examples for which the condition holds
3. The paper empiricallly indentified cases where the aiminig condition does not hold

**Weaknesses:**

1. The proof for Example 4 (shallow neural network) is problematic. In particular, based on the definition and assumption of $S$, the paper derived $f_i^* = 0$, and that points in $S$ have bounded norm (which is necessary to obtain the equation below line 684). However, these two conditions are contradictory as the logistic loss only achieves zero training loss when the output of the neural networks has magnitude infinity. This breaks the proof of Example 4.
2. The theoretical implications of Theorem 1-3 are week. The minimum loss during training is upper bounded by $O(\beta\sigma_{\text{int}}^2)$ which can be extremely large. First, in real world applications, it is very hard to control $\sigma_{\text{int}}^2$ as the model output can vary a lot based on different inputs (this is not a big issue, as many previous works assumes the boundedness of $\sigma^2$). Moreover, $\beta$ can be as large as $10^6$ in order for the condition to hold, as shown in Figure 4. Multiplying the two gives an error region that does not seems to be small enough.
3. There is no strong theoretical evidence that the proposed condition holds for neural network training in general. In particular, a major significance of developping the regulatory condition for neural network training is to theoretically show the convergence of neural network training based on such conditions. The proposed conditions loses such significance because it misses the theoretical connection with neural network training.
4. The empirical justification that neural network training satisfies the proposed condition is limited. Figure 4 only shows the condition holds for large $\beta$ (weaker condition), which does not imply great applicability of the condition in showing the convergence property. Indeed, Figure 5-7 shows that the condition holds for smaller $\beta$ for harder tasks, which is quite conter-intuitive since harder tasks should have less flavorable loss landscape. However, it should be noticed that the results is obtained by using $x^K$ to approximate $x^*$. Therefore, the seemingly flavorable result may come from the fact that in harder tasks $x^K$ is a worse estimator of $x^*$ than in simpler tasks, which implies that the results reported in Figure 5-7 may not be showing the actual $\alpha$-$\beta$ condition in practice.

**Questions:**

Is it possible to establish the convergence rate of the non-stochastic gradient descent based on the proposed condition and the smoothness condition?

**Limitations:**

The author mentioned some limitations of the paper. However, I believe that the biggest limitation is the missing theoretical connection between the proposed condition and neural network training, which is not mentioned in the paper.

---

> ### Author Rebuttal · Authors · 2024-08-06
>
> We sincerely thank the reviewer for their thorough reviews, insightful comments, and valuable
> questions regarding our paper.
>
> **A to W1:** Thank you for pointing this out. To ensure boundedness of $\mathcal{S}$, one can simply add L2 regularization. We provide a sketch of the proof and will change Example 4 in the revised version of the paper.
>
> **Example.** Consider training a two-layer neural network with a logistic loss $f = n^{-1}\sum_{i=1}^nf_i$, $f_i(W,v)=\phi(y_i\cdot v^\top\sigma(Wx_i))+\lambda_1||v||^2+\lambda_2||W||^2_F$ for a classification problem where $\phi(t):=\log(1+\exp(-t)),$ $W\in\mathbb{R}^{k\times d}, v\in\mathbb{R}^{k},$ $\sigma$ is a ReLU function applied coordinate-wise, $y_i\in\\{-1,+1\\}$ is a label and $x_i\in\mathbb{R}^d$ is a feature vector. Assume that the interpolation does not hold, i.e. $\min_{(z,Z)\in\mathcal{S}}f_i(z,Z)>f_i^*$. Let $\mathcal{X}$ be any bounded set that contains $\mathcal{S}$. Then the $\alpha$-$\beta$-condition holds in $\mathcal{X}$ for some $\alpha\ge1$ and $\beta=\alpha-1.$
>
> **Proof sketch.** Due to space limit we only mention the main difference in comparison with the proof of Example 4 in the submission.
>
> - First, we have additional $\lambda_1v$ and $\lambda_2W$ in the calculations of the gradients $\nabla_vf_i(v,W)$ and $\nabla_Wf_i(v,W)$ correspondingly.
> - The optimal value of $f_i^*>0$ because of the L2 regularization.
> - Because of L2 regularization $\mathcal{S}$ is bounded and from example statement it fully lies in $\mathcal{X}.$
> - From non-interpolation assumption we make, there exists $c:=\min_{i\in[n]}\min\_{(Z,z)\in\mathcal{S}}f_i(Z,z)>f_i^*.$
> - In the next derivations, we mainly follow the derivations of example 4. After using convexity of $\phi$, $\alpha$-$\beta$-condition is verified if we have\begin{align}&2\phi(y\_i(v\circ e_w)^\top Wx_i)+2\lambda_1\left<v,v-z\right>+2\lambda_2\left<W,W-Z\right>_F-\phi(y_i(v\circ e_w)^\top Zx_i)-\phi(y_i(z\circ e_w)^\top Wx_i)\\\\&\quad\ge\alpha\left[\phi(y_i(v\circ e_w)^\top Wx_i)+\lambda_1||v||^2+\lambda_2||W||_F^2-\phi(y_i (z\circ e_z)^\top Zx_i)-\lambda_1||z||^2-\lambda_2||Z||^2_F\right]\\\\&\quad-\beta(\phi(y_i(v\circ e_w)^\top Wx_i)+\lambda_1||v||^2+\lambda_2||W||^2_F-f_i^*).\end{align}
> - Rearranging terms and choosing $\alpha-\beta=1$ we get\begin{align}&\phi(y_i(v\circ e_w)^\top Wx_i)+\lambda_1||v-z||^2+ \lambda_2||W-Z||^2_F+\alpha\phi(y_i(z\circ e_z)^\top Zx_i)+(\alpha-1)\left[\lambda_1||z||^2+\lambda_2||Z||^2_F\right]\\\\&\quad\ge\phi(y_i(v\circ e_w)^\top Zx_i)+\phi(y_i(z\circ e_w)^\top Wx_i)+(\alpha-1)f_i^*.\end{align}
> - Since $(W,v),(z,Z)\in\mathcal{X}$, there exist constants $R,r\ge0$ such that $||z||,||v||\le r$ and $||Z||_F,||W||_F\le R$,and RHS in the above is bounded by\begin{align}2\max_i\log(1+\exp(Rr||x_i||))+(\alpha-1)f_i^*\ge\phi(y_i(v\circ e_w)^\top Zx_i)+\phi(y_i(z\circ e_w)^\top Wx_i)+(\alpha-1)f_i^*.\end{align}
> - From the assumption we have $\phi(y_i(z\circ e_z)^\top Zx_i)\ge c$.
> - Then we can take $\alpha\ge\max\left\\{\frac{2\max_i\log(1+\exp(Rr||x_i||))}{c-f_i^*},1\right\\}$ and $\beta=\alpha-1.$
> We will modify Example 4 in the paper to the one with L2 regularization.
>
> **A to W2:** We refer to the general rebuttal.
>
> **A to W3:** We agree that a theoretical verification of the proposed condition for neural networks is an important question, albeit a difficult one. Most existing theoretical results rely on unpractical amounts of over-parameterization, whereas our aim was to relax this requirement. Notably, Example 4 demonstrates that the $\alpha$-$\beta$ condition holds for 2-layer NNs with L2 regularization, albeit under somewhat restrictive assumptions, underscoring the complexity of the problem. Empirically, we have conducted an extensive verification of the $\alpha$-$\beta$ condition across a wide range of architectures, indicating that this proposed condition is a promising direction for the analysis of neural networks.
>
> **A to W4:** We agree with the reviewer that for harder tasks $x^K$ might be a worse approximation of $x^*$ than for easier ones. We did mention in the limitations section that the empirical verification of the $\alpha$-$\beta$ condition is a difficult task. However, we use the best-known optimizer to get as close as possible to $x^*$ (we choose the optimizer and last iterate $x^K$ such that the performance of networks is close to state-of-the-art). We emphasize that this type of verification has not been conducted in previous studies to the best of our knowledge, making it a significant advancement in the literature.
> Furthermore, we have made a significant effort to provide empirical verification across a wide range of neural network architectures, including those that are widely used in practice, such as ResNet and Transformers.
>
> **A to Q:** Yes, it is possible to derive the convergence of GD under $\alpha$-$\beta$-condition. Note that averaging the $\alpha$-$\beta$-condition across all $i\in[n]$ gives $$\left<\nabla f(x),x-x_p\right>\ge\alpha(f(x)-f(x_p))-\beta\left(f(x)-n^{-1}\sum_if_i^*\right).$$
> Using this inequality and following the standard proof of GD we get
> \begin{align}\mathrm{dist}(x^{k+1},\mathcal{S})^2 &\le\mathrm{dist}(x^{k},\mathcal{S})^2-2\gamma\left<\nabla f(x^k),x^k-x_p^k\right>+\gamma^2||\nabla f(x^k)||^2\\\\&\le\mathrm{dist}(x^{k},\mathcal{S})^2-2\alpha \gamma(f(x^k)-f^*)+2\beta\gamma\left(f(x^k)-n^{-1}\sum_if_i^*\right)+2L\gamma^2(f(x^k)-f^*)\\\\&\le\mathrm{dist}(x^{k},\mathcal{S})^2-2(\alpha-\beta-L\gamma)\gamma(f(x^k) - f^*)+2\beta\gamma\left(f^*-n^{-1}\sum_if_i^*\right).\end{align}
>
> Choosing $\gamma\le \frac{\alpha-\beta}{2L}$ gives$$\mathrm{dist}(x^{k+1},\mathcal{S})^2\le\mathrm{dist}(x^{k},\mathcal{S})^2-(\alpha-\beta)\gamma(f(x^k)-f^*)+2\beta\gamma\left(f^*-n^{-1}\sum_if_i^*\right).$$Finally, unrolling this recursion we derive the following rate$$\min\limits_{0\le k<K}[f(x^k)-f^*]\le\frac{\mathrm{dist}(x^0,\mathcal{S})^2}{\gamma(\alpha-\beta)K}+\frac{2\beta}{\alpha-\beta}\sigma^2_{\mathrm{int}}.$$

---

> > ### Comment · Reviewer_W37Q · 2024-08-09
> > **Response to the Author's Rebuttal**
> >
> > Thank you so much for providing the detailed explanation and the proof sketch.
> >
> > For W1, I agree with the new proof sketch which turns the set $S$ bounded by adding the regularization term. For W3, I agree that verifying the condition theoretically on NN with more complicated architectures can be a significant next step.
> >
> > For W2 however, my concern remains about the non-vanishing term. Based on the author's argument, I agree that the non-vanishing term is necessary. However, this does poses the question of whether the $\alpha$-$\beta$ is a condition powerful enough to guarantee a good convergence property. In short, I am concerned that this condition might be too relaxed, as one can choose super large $\beta$ and $\alpha$ while maintaining the difference between the two.
> >
> > For W4, I understand the difficulty of verifying $\alpha$-$\beta$ condition, and I am truly aware of the fact that finding minimizers of NN training is basically not possible. However, my real concern is that the values of $\alpha$ and $\beta$ are large for simpler networks, but smaller for more complicated networks. I still suspect that the large values of $\alpha$ and $\beta$ will hold across all neural network training.
> >
> > Combining W2 with W4, I still have concern that the $\alpha$-$\beta$ condition may not be a good condition for NN training.
> >
> > I have raised the score based on the new proof sketch, but I believe that the paper still falls below the acceptance borderline given the above concern.

---

> ### Author Response · Authors · 2024-08-12
> **Response**
>
> We would like to highlight that the value of $\beta$ itself is not involved in the convergence rate, but rather $\alpha-\beta$ and $\beta\sigma^2_{\mathrm{int}}$. While $\alpha-\beta$ is typically a constant of order $0.1$ in our experiments, we also observe that $\beta\sigma^2\_{\mathrm{int}}$ decreases with increasing over-parameterization (see figures in additional pdf). Moreover, we decided to conduct experiments on Resnet18 and Resnet34 on CIFAR100 to support our claims on larger models. Since the rules regarding posting external links are unclear, we provide approximate values of the smallest $\beta$ found in the experiments over 3 runs:
> |model| batch size | $\beta$|
> |-------|:-------------:|:----------:|
> |Resnet18| 64 | 578|
> |Resnet18| 128| 160|
> |Resnet18| 256 | 49|
> |Resnet34| 64| 797|
> |Resnet34|128 | 494|
> |Resnet34| 256| 70|
>
> We will include the plots of this set of experiments in the revised version of the paper. We observe that the value of $\beta$ tends to decrease when we increase the depth of the Resnet model (note that we have results for Resnet9 in the main paper). These results are consistent with all our previous comments, but in this case for larger models as well.
>
> Based on all the results, we do not agree with the reviewer that $\alpha$-$\beta$-condition might be too relaxed. The experimental and theoretical results show all expected practical trade-offs.

---

> ### Author Response · Authors · 2024-08-14
> **Response**
>
> Dear reviewer,
>
> As the deadline is approaching, we would like to know if the above response answers your concern regarding the values of $\beta$. We highlight that Resnet architecture can be accurately trained using SGD with CycleOneLR learning rate schedule reaching small loss. Therefore the issue of finding a bad approximation of $x^*$ is not the same as for large models. Based on Resnet experiments the values of $\beta$ tend to decrease with the number of layers as we expect (i.e., increasing the complexity of the model). Hence, in our opinion, the proposed $\alpha$-$\beta$-condition captures all expected trends in the training (convergence to the non-vanishing neighborhood, neighborhood size vanishes as a model becomes more over-parameterized).

---

### Official Review · Reviewer_mWKG · 2024-07-15

**Soundness:** 4
**Presentation:** 4
**Contribution:** 3
**Rating:** 7
**Confidence:** 4

**Summary:**

This paper proposes a novel class of functions and proves convergence of gradient descent (and some other optimizers).
Contrary to some previous classes, in relation to deep neural networks, this new class does not require extreme overparameterization.
In addition to theoretical convergence results, experiments are provided showing that some example neural networks seem to belong to the function class.

**Strengths:**

The paper is written very clearly. It provides a new theoretical tool for studying convergence in deep neural networks, and has the potential of high impact.

**Weaknesses:**

Some of the highlighted differences with respect to previous work are unfair, in particular previous work in overparameterized models.
For example, figure 1c and 1d show very small values of the PL constant in practical problems, implying slow theoretical convergence. However, the same is true for the new proposed class of functions, where the PL constant is replaced by alpha-beta, which seems very small in most experiments (alpha and beta have very similar values) thus implying very slow convergence. It would be easy to plots alpha-beta values in experiments to verify how large it can get, and thus show how fast is the convergence predicted by theory.

Also, all other function classes considered in previous work guarantee convergence to a global minimum, here instead convergence theorems have an extra term since different data points have different optima, and it's not clear how big are those terms. It may be that these terms are so large that the bounds become trivial, for example when those terms are much larger than the decaying 1/K terms. At least some discussion comparing the magnitude of such terms would make the work more valuable.

**Questions:**

In my understanding, if the interpolation condition holds, then the new proposed class is equivalent to PL. Is that true and why is this not explained in the paper?

---

> ### Author Rebuttal · Authors · 2024-08-06
>
> We sincerely thank the reviewer for their thorough reviews, insightful comments, and valuable
> questions regarding our paper.
>
> ***W1.*** "Some of the highlighted differences..."
>
> ***A to W1:*** This is a good comment and we will provide further detail on it in a revision. In Figures 1c-1d we observe that the empirical PL constant can be of order $10^{-7}.$ Besides, in each figure where we plot the empirical values of $\alpha$ and $\beta$, we include only those values that satisfy $\alpha \ge \beta + 0.1$. Let's now compare the optimization terms decreasing with $K$ in the rates under both conditions: $\frac{L}{K(\alpha-\beta)}$ under the $\alpha$-$\beta$-condition vs $(1-\frac{\mu}{L})^K$ under the PL condition (see Theorem 3.9 in [1] for the explicit rate in the deterministic setting. Note that the contraction factor in the rate in the stochastic setting is even worse; see Theorem 5.10). We refer the reviewer to Figure 3 in the separate pdf file where we provide plots for several values of $L$ and $\mu=10^{-7}, \alpha-\beta=0.1$. We observe that the theoretical convergence under the $\alpha$-$\beta$-condition with empirical values of $\alpha$ and $\beta$ is always more favorable than that under the PL condition with empirical value of $\mu.$ In addition, from a theoretical point of view (see Theorem 1) we can choose the stepsize of order $\mathcal{O}(\frac{\alpha-\beta}{L})$ which implies that the first term in the convergence bound is $\mathcal{O}\left(\frac{L\mathrm{dist}(x^0,\mathcal{S})^2}{K}\right),$ i.e. it is not affected by $\alpha-\beta$ at all which shows a clear advantage of $\alpha$-$\beta$-condition over PL condition.
>
> ***W2:*** "Also, all other function classes considered in previous work..."
>
> ***A to W2:*** We refer to the general rebuttal.
>
> ***Q:*** "
> In my understanding, if the interpolation condition holds..."
>
> ***A to Q:*** Good question! In the interpolation regime, we have $f_i^* = f^*.$ This means that $\sigma_{\mathrm{int}}^2 = 0.$ Therefore, $\alpha$-$\beta$-condition reduces to
> $$
> \left<\nabla f_i(x), x-x_p\right> \ge \alpha(f_i(x) - f_i(x_p)), \quad x_p = \mathrm{proj}_{\mathcal{S}}(x).
> $$
> If we replace $x_p$ by some fixed point $x^*\in\mathcal{S},$ then the functions satisfying the above are called quasi-convex. However, to the best of our knowledge, there is no known relation between quasi-convex and PL functions. Therefore, the interpolation regime does not imply that the $\alpha$-$\beta$-condition reduces to PL. We will add a comment on this in the revised version of the paper.

---

> > ### Comment · Reviewer_mWKG · 2024-08-09
> >
> > Thank you for the detailed response, my opinion is still that the paper should be accepted.

---

### Author Rebuttal · Authors · 2024-08-06

We thank all reviewers for their valuable comments and questions that allowed us to improve our paper.

**Non-vanishing term in the convergence rate:**
Three of the reviewers raised the question on the convergence of optimizers under the $\alpha$-$\beta$-condition. The main comment is about the presence of the non-vanishing $\mathcal{O}(\beta\sigma^2_{\mathrm{int}})$ term in the rate. Below we provide a detailed discussion of this term which we believe can not be removed since $\alpha-\beta$ functions can have local minima.

1) This term directly appears because of the use of $\alpha$-$\beta$-condition in the analysis, without relying on additional upper bounds or approximations, and it captures the potential presence of local minima as explained below

2) We recall that one can find examples that satisfy the $\alpha$-$\beta$-condition and have local minima. For instance, let $$f_1(x,y)=\frac{1+x^2+y^2}{2+x^2+y^2},f_2(x,y)=\frac{(x-2)^2+(y-2)^2}{1+(x-2)^2+(y-2)^2},f=1/2(f_1+f_2).$$ For this problem, $f_1^*=\frac{1}{2}, w_1^*=(0,0),f_2^*=0,w_2^*=(2,2),f^*\approx 0.45,w^*=(1.97,1.97)$ (we will add the example in the revised version with a proof). We attach a surface plot in the separate pdf) that satisfies the condition with $\alpha\gtrsim1250,\beta=\alpha-1.$ Moreover, in example 4 we provide an example of a 2-layer neural network with ReLU activation that is known to have spurious local minima [4]. In contrast, previously proposed conditions such as PL and quasar-convexity do not allow to have local minima/saddle points.

3) Note that on the losses that have local minima, SGD does not converge to $f^*$, regardless of the values of stepsize $\gamma$. Indeed, as the rates must hold for any initialization $x^0$, in the worst-case scenario (i.e. $x^0$ is close to spurious minima), annealing the stepsize does not recover convergence to the global minimizer. Near local minima, the gradient dynamics resemble those on a quadratic function (Hartman–Grobman theorem), and one can assume the noise strength is bounded (in norm) near the basin (which holds for ERM problem that we consider). So if we initialize close to a local minimizer and let the stepsize $\gamma$ converge to zero, we get the convergence to the spurious local minima. This suboptimality is modeled by the stepsize-independent quantity $\mathcal{O}(\beta\sigma\_{\mathrm{int}}^2)$ since we provide convergence guarantees for function suboptimality value and not for squared gradient norm as usual in the non-convex setting. Because of the previous considerations, the loss is not necessarily minimized making it necessary to include a worst-case correction term $\mathcal{O}(\beta\sigma\_{\mathrm{int}}^2)$.


**Additional evidence from prior works:** Other prior works lead us to believe this term indeed correctly appears in the convergence bound:

- We emphasize that the stochastic term $\mathcal{O}(L\gamma\sigma^2_{\mathrm{int}})$ also appears in the standard rate of SGD with constant learning rate; see Theorem 5.5 in [1]. Although it can be annealed with a decreasing learning rate, this only guarantees convergence to a critical point for non-convex functions.
- A non-vanishing term is frequently observed when training neural networks. We for instance refer the reviewer to Figure 8.1 in the seminal reference [2]. This is also observed during the training of language models where the loss is significantly larger than $0$; see plots in Figures 16 and 17 corresponding to language modeling in the appendix. This phenomenon suggests that reaching a critical point that is a global minimizer is not always observed practically. Therefore, the presence of a non-vanishing error term in the rate with the $\alpha$-$\beta$ condition is consistent with empirical observation.
- Finally, prior works that propose other conditions to describe the loss landscape of deep neural networks (e.g. gradient confusion [3]) also obtain a non-vanishing term in the convergence rate (see Theorem 3.2 in [3]). Interestingly, the non-vanishing terms from both analyses capture some sort of discrepancy between datapoints, although they do not seem to be directly relatable.
- The value of $\beta\sigma^2_{\mathrm{int}}$ decreases with over-parametrization which is the expected trend; see rough estimations of $\beta\sigma^2_{\mathrm{int}}$ in the table in the separate pdf.

Nevertheless, we will investigate this aspect further in the next revision and provide an extended discussion regarding this limitation.

[1] Garrigos, Guillaume and Gower, Robert M, Handbook of convergence theorems for (stochastic) gradient methods, arXiv preprint arXiv:2301.11235, 2023.

[2] Ian Goodfellow, Yoshua Bengio, Aaron Courville, Deep Learning, MIT press, 2016.

[3] Sankararaman, Karthik Abinav and De, Soham and Xu, Zheng and Huang, W Ronny and Goldstein, Tom, The impact of neural network overparameterization on gradient confusion and stochastic gradient descent ICML 2020.

[4] Safran, Itay and Shamir, Ohad, Spurious local minima are common in two-layer relu neural networks, ICML 2018.

---

### Comment · Area_Chair_81NN · 2024-08-08

Dear reviewers,

could you have a look at the authors response and comment on them if you have done so, yet.

thanks in advance


your area chair

---

### Decision · Program_Chairs · 2024-09-25

**Decision:**

Accept (poster)

**Comment:**

This paper proposed a new a novel class of functions (alpha-beta condition) and proves convergence of gradient descent. The authors showed that several non-convex landscape with saddle point e.g. matrix completion satisfies the alpha-beta condition. Authors also empirical veryfied the condition on several NN training landscape.  Contrary to some previous classes, in relation to deep neural networks, this new class does not require extreme overparameterization. Relationship between the alpha-beta condition with pl condition and quasi-convex should be included. The framework of the paper is nice, but the applicability of the alpha-beta condition to (modern, current) architectures needs/inspires further study.  Overall this is quite nice paper for optimization studying in the area of machine learning.